# ASYNCHRONOUS ADVANTAGE ACTOR CRITIC: NON-ASYMPTOTIC ANALYSIS AND LINEAR SPEEDUP

## ABSTRACT

Asynchronous and parallel implementation of standard reinforcement learning (RL) algorithms is a key enabler of the tremendous success of modern RL. Among many asynchronous RL algorithms, arguably the most popular and effective one is the asynchronous advantage actor-critic (A3C) algorithm. Although A3C is becoming the workhorse of RL, its theoretical properties are still not well-understood, including the non-asymptotic analysis and the performance gain of parallelism (a.k.a. speedup). This paper revisits the A3C algorithm with TD(0) for the critic update, termed A3C-TD(0), with provable convergence guarantees. With linear value function approximation for the TD update, the convergence of A3C-TD(0) is established under both i.i.d. and Markovian sampling. Under i.i.d. sampling, A3C-TD(0) obtains sample complexity of $\mathcal{O}(\epsilon^{-2.5}/N)$ per worker to achieve $\epsilon$ accuracy, where $N$ is the number of workers. Compared to the best-known sample complexity of $\mathcal{O}(\epsilon^{-2.5})$ for two-timescale AC, A3C-TD(0) achieves *linear speedup*, which justifies the advantage of parallelism and asynchrony in AC algorithms theoretically for the first time. Numerical tests on synthetically generated instances and OpenAI Gym environments have been provided to verify our theoretical analysis.

## 1   INTRODUCTION

Reinforcement learning (RL) has achieved impressive performance in many domains such as robotics [1, 2] and video games [3]. However, these empirical successes are often at the expense of significant computation. To unlock high computation capabilities, the state-of-the-art RL approaches rely on sampling data from massive parallel simulators on multiple machines [3, 4, 5]. Empirically, these approaches can stabilize the learning processes and *reduce training time* when they are implemented in an asynchronous manner. One popular RL method that often achieves the best empirical performance is the asynchronous variant of the actor-critic (AC) algorithm, which is referred to as A3C [3].

A3C builds on the original AC algorithm [6]. At a high level, AC simultaneously performs policy optimization (a.k.a. the actor step) using the policy gradient method [7] and policy evaluation (a.k.a. the critic step) using the temporal difference learning (TD) algorithm [8]. To ensure scalability, both actor and critic steps can combine with various function approximation techniques. To ensure stability, AC is often implemented in a two time-scale fashion, where the actor step runs in the slow timescale and the critic step runs in the fast timescale. Similar to other on-policy RL algorithms, AC uses samples generated from the target policy. Thus, data sampling is entangled with the learning procedure, which generates significant *overhead*. To speed up the sampling process of AC, A3C introduces multiple workers with a shared policy, and each learner has its own simulator to perform data sampling. The shared policy can be then updated using samples collected from multiple learners.

Despite the tremendous empirical success achieved by A3C, to the best of our knowledge, its theoretical property is not well-understood. The following *theoretical* questions remain unclear: **Q1)** Under what assumption does A3C converge? **Q2)** What is its convergence rate? **Q3)** Can A3C obtain benefit (or speedup) using parallelism and asynchrony?

For **Q3**), we are interested in the *training time linear speedup* with $N$ workers, which is the ratio between the training time using a single worker and that using $N$ workers. Since asynchronous parallelism mitigates the effect of stragglers and keeps all workers busy, the training time speedup

can be measured roughly by the sample (i.e., computational) complexity linear speedup [9], given by

$$\text{Speedup}(N) = \frac{\text{sample complexity when using one worker}}{\text{average sample complexity per worker when using } N \text{ workers}}. \tag{1}$$

If $\text{Speedup}(N) = \Theta(N)$, the speedup is linear, and the training time roughly reduces linearly as the number of workers increases. This paper aims to answer these questions, towards the goal of providing theoretical justification for the empirical successes of parallel and asynchronous RL.

## 1.1 RELATED WORKS

**Analysis of actor critic algorithms.** AC method was first proposed by [6, 10], with asymptotic convergence guarantees provided in [6, 10, 11]. It was not until recently that the *non-asymptotic* analyses of AC have been established. The finite-sample guarantee for the batch AC algorithm has been established in [12, 13] with i.i.d. sampling. Later, in [14], the finite-sample analysis was established for the double-loop nested AC algorithm under the Markovian setting. An improved analysis for the Markovian setting with minibatch updates has been presented in [15] for the nested AC method. More recently, [16, 17] have provided the first finite-time analyses for the two-timescale AC algorithms under Markov sampling, with both $\tilde{O}(\epsilon^{-2.5})$ sample complexity, which is the best-known sample complexity for two-timescale AC. Through the lens of bi-level optimization, [18] has also provided finite-sample guarantees for this two-timescale Markov sampling setting, with global optimality guarantees when a *natural* policy gradient step is used in the actor. However, none of the existing works has analyzed the effect of the asynchronous and parallel updates in AC.

**Empirical parallel and distributed AC.** In [3], the original A3C method was proposed and became the workhorse in empirical RL. Later, [19] has provided a GPU-version of A3C which significantly decreases training time. Recently, the A3C algorithm is further optimized in modern computers by [20], where a large batch variant of A3C with improved efficiency is also proposed. In [21], an importance weighted distributed AC algorithm IMPALA has been developed to solve a collection of problems with one single set of parameters. Recently, a gossip-based distributed yet synchronous AC algorithm has been proposed in [5], which has achieved the performance competitive to A3C.

**Asynchronous stochastic optimization.** For solving general optimization problems, asynchronous stochastic methods have received much attention recently. The study of asynchronous stochastic methods can be traced back to 1980s [22]. With the batch size $M$, [23] analyzed asynchronous SGD (async-SGD) for convex functions, and derived a convergence rate of $\mathcal{O}(K^{-\frac{1}{2}}M^{-\frac{1}{2}})$ if delay $K_0$ is bounded by $\mathcal{O}(K^{\frac{1}{4}}M^{-\frac{3}{4}})$. This result implies linear speedup. [24] extended the analysis of [23] to smooth convex with nonsmooth regularization and derived a similar rate. Recent studies by [25] improved upper bound of $K_0$ to $\mathcal{O}(K^{\frac{1}{2}}M^{-\frac{1}{2}})$. However, all these works have focused on the single-timescale SGD with a single variable, which cannot capture the stochastic recursion of the AC and A3C algorithms. To best of our knowledge, non-asymptotic analysis of asynchronous two-timescale SGD has remained unaddressed, and its speedup analysis is even an uncharted territory.

## 1.2 THIS WORK

In this context, we revisit A3C with TD(0) for the critic update, termed A3C-TD(0). The hope is to provide *non-asymptotic* guarantee and *linear speedup* justification for this popular algorithm.

**Our contributions.** Compared to the existing literature on both the AC algorithms and the async-SGD, our contributions can be summarized as follows.

**c1)** We revisit two-timescale A3C-TD(0) and establish its convergence rates with both i.i.d. and Markovian sampling. To the best of our knowledge, this is the first non-asymptotic convergence result for *asynchronous parallel* AC algorithms.

**c2)** We characterize the sample complexity of A3C-TD(0). In i.i.d. setting, A3C-TD(0) achieves a sample complexity of $\mathcal{O}(\epsilon^{-2.5}/N)$ per worker, where $N$ is the number of workers. Compared to the best-known complexity of $\mathcal{O}(\epsilon^{-2.5})$ for i.i.d. two-timescale AC [18], A3C-TD(0) achieves *linear speedup*, thanks to the parallelism and asynchrony. In the Markovian setting, if delay is bounded, the sample complexity of A3C-TD(0) matches the order of the non-parallel AC algorithm [17].

**c3)** We test A3C-TD(0) on the synthetically generated environment to verify our theoretical guarantees with both i.i.d. and Markovian sampling. We also test A3C-TD(0) on the classic control tasks and Atari Games from OpenAI Gym. **Code is available in the supplementary material.**

**Technical challenges.** Compared to the recent convergence analysis of nonparallel two-timescale AC in [16, 17, 18], several new challenges arise due to the parallelism and asynchrony.

*Markovian noise coupled with asynchrony and delay.* The analysis of two-timescale AC algorithm is non-trivial because of the Markovian noise coupled with both the actor and critic steps. Different from the nonparallel AC that only involves a single Markov chain, asynchronous parallel AC introduces multiple Markov chains (one per worker) that mix at different speed. This is because at a given iteration, workers collect different number of samples and thus their chains mix to different degrees. As we will show later, the worker with the slowest mixing chain will determine the convergence.

*Linear speedup for SGD with two coupled sequences.* Parallel async-SGD has been shown to achieve linear speedup recently [9, 26]. Different from async-SGD, asynchronous AC is a two-timescale stochastic *semi-gradient* algorithm for solving the more challenging *bilevel* optimization problem (see [18]). The errors induced by asynchrony and delay are intertwined with both actor and critic updates via a nested structure, which makes the sharp analysis more challenging. Our linear speedup analysis should be also distinguished from that of mini-batch async-SGD [27], where the speedup is a result of *variance reduction* thanks to the larger batch size generated by parallel workers.

## 2 PRELIMINARIES

### 2.1 MARKOV DECISION PROCESS AND POLICY GRADIENT THEOREM

RL problems are often modeled as an MDP described by $\mathcal{M} = \{\mathcal{S}, \mathcal{A}, \mathcal{P}, r, \gamma\}$, where $\mathcal{S}$ is the state space, $\mathcal{A}$ is the action space, $\mathcal{P}(s'|s,a)$ is the probability of transitioning to $s' \in \mathcal{S}$ given current state $s \in \mathcal{S}$ and action $a \in \mathcal{A}$, and $r(s,a,s')$ is the reward associated with the transition $(s,a,s')$, and $\gamma \in [0,1)$ is a discount factor. Throughout the paper, we assume the reward $r$ is upper-bounded by a constant $r_{\max}$. A policy $\pi : \mathcal{S} \to \Delta(\mathcal{A})$ is defined as a mapping from the state space $\mathcal{S}$ to the probability distribution over the action space $\mathcal{A}$.

Considering discrete time $t$ in an infinite horizon, a policy $\pi$ can generate a trajectory of state-action pairs $(s_0, a_0, s_1, a_1, \ldots)$ with $a_t \sim \pi(\cdot|s_t)$ and $s_{t+1} \sim \mathcal{P}(\cdot|s_t, a_t)$. Given a policy $\pi$, we define the state and state action value functions as

$$V_\pi(s) := \mathbb{E}\left[\sum_{t=0}^{\infty} \gamma^t r(s_t, a_t, s_{t+1}) \mid s_0 = s\right], \; Q_\pi(s,a) := \mathbb{E}\left[\sum_{t=0}^{\infty} \gamma^t r(s_t, a_t, s_{t+1}) \mid s_0 = s, a_0 = a\right] \quad (2)$$

where $\mathbb{E}$ is taken over the trajectory $(s_0, a_0, s_1, a_1, \ldots)$ generated under policy $\pi$. With the above definitions, the advantage function is $A_\pi(s,a) := Q_\pi(s,a) - V_\pi(s)$. With $\eta$ denoting the initial state distribution, the discounted state visitation measure induced by policy $\pi$ is defined as $d_\pi(s) := (1 - \gamma) \sum_{t=0}^{\infty} \gamma^t \mathbb{P}(s_t = s \mid s_0 \sim \eta, \pi)$, and the discounted state action visitation measure is $d'_\pi(s,a) = (1 - \gamma) \sum_{t=0}^{\infty} \gamma^t \mathbb{P}(s_t = s \mid s_0 \sim \eta, \pi)\pi(a|s)$.

The goal of RL is to find a policy that maximizes the expected accumulative reward $J(\pi) := \mathbb{E}_{s \sim \eta}[V_\pi(s)]$. When the state and action spaces are large, finding the optimal policy $\pi$ becomes computationally intractable. To overcome the inherent difficulty of learning a function, the policy gradient methods search the best performing policy over a class of parameterized policies. We parameterize the policy with parameter $\theta \in \mathbb{R}^d$, and solve the optimization problem as

$$\max_{\theta \in \mathbb{R}^d} J(\theta) \quad \text{with} \quad J(\theta) := \mathbb{E}_{s \sim \eta}[V_{\pi_\theta}(s)]. \quad (3)$$

To maximize $J(\theta)$ with respect to $\theta$, one can update $\theta$ using the policy gradient direction given by [7]

$$\nabla J(\theta) = \mathbb{E}_{s,a \sim d'_\theta}[A_{\pi_\theta}(s,a)\psi_\theta(s,a)], \quad (4)$$

where $\psi_\theta(s,a) := \nabla \log \pi_\theta(a|s)$, and $d'_\theta := (1 - \gamma) \sum_{t=0}^{\infty} \gamma^t \mathbb{P}(s_t = s \mid s_0, \pi_\theta)\pi_\theta(a|s)$. Since computing $\mathbb{E}$ in (4) is expensive if not impossible, popular policy gradient-based algorithms iteratively update $\theta$ using stochastic estimate of (4) such as REINFORCE [28] and G(PO)MDP [29].

## 2.2 ACTOR-CRITIC ALGORITHM WITH VALUE FUNCTION APPROXIMATION

Both REINFORCE and G(PO)MDP-based policy gradient algorithms rely on a Monte-Carlo estimate of the value function $V_{\pi_\theta}(s)$ and thus $\nabla J(\theta)$ by generating a trajectory per iteration. However, policy gradient methods based on Monte-Carlo estimate typically suffer from high variance and large sampling cost. An alternative way is to recursively refine the estimate of $V_{\pi_\theta}(s)$. For a policy $\pi_\theta$, it is known that $V_{\pi_\theta}(s)$ satisfies the Bellman equation [30], that is

$$V_{\pi_\theta}(s) = \mathbb{E}_{a\sim\pi_\theta(\cdot|s),\, s'\sim\mathcal{P}(\cdot|s,a)} \left[ r(s,a,s') + \gamma V_{\pi_\theta}(s') \right], \quad \forall s \in \mathcal{S}. \tag{5}$$

In practice, when the state space $\mathcal{S}$ is prohibitively large, one cannot afford the computational and memory complexity of computing $V_{\pi_\theta}(s)$ and $A_{\pi_\theta}(s,a)$. To overcome this curse-of-dimensionality, a popular method is to approximate the value function using function approximation techniques. Given the state feature mapping $\phi(\cdot) : \mathcal{S} \to \mathbb{R}^{d'}$ for some $d' > 0$, we approximate the value function linearly as $V_{\pi_\theta}(s) \approx \hat{V}_\omega(s) := \phi(s)^\top \omega$, where $\omega \in \mathbb{R}^{d'}$ is the critic parameter.

Given a policy $\pi_\theta$, the task of finding the best $\omega$ such that $V_{\pi_\theta}(s) \approx \hat{V}_\omega(s)$ is usually addressed by TD learning [8]. Defining the $k$th transition as $x_k := (s_k, a_k, s_{k+1})$ and the corresponding TD-error as $\hat{\delta}(x_k, \omega_k) := r(s_k, a_k, s_{k+1}) + \gamma\phi(s_{k+1})^\top\omega_k - \phi(s_k)^\top\omega_k$, the parameter $\omega$ is updated via

$$\omega_{k+1} = \Pi_{R_\omega} \big( \omega_k + \beta_k g(x_k, \omega_k) \big) \quad \text{with} \quad g(x_k, \omega_k) := \hat{\delta}(x_k, \omega_k)\nabla_{\omega_k}\hat{V}_{\omega_k}(s_k) \tag{6}$$

where $\beta_k$ is the critic stepsize, and $\Pi_{R_\omega}$ is the projection with $R_\omega$ being a pre-defined constant. The projection step is often used to control the norm of the gradient. In AC, it prevents the actor and critic updates from going a too large step in the 'wrong' direction; see e.g., [6, 16, 17].

Using the definition of advantage function $A_{\pi_\theta}(s,a) = \mathbb{E}_{s'\sim\mathcal{P}}[r(s,a,s') + \gamma V_{\pi_\theta}(s')] - V_{\pi_\theta}(s)$, we can also rewrite (4) as $\nabla J(\theta) = \mathbb{E}_{s,a\sim d'_\theta, s'\sim\mathcal{P}} \left[ (r(s,a,s') + \gamma V_{\pi_\theta}(s') - V_{\pi_\theta}(s))\psi_\theta(s,a) \right]$. Leveraging the value function approximation, we can then approximate the policy gradient as

$$\widehat{\nabla} J(\theta) = \left( r(s,a,s') + \gamma\hat{V}_\omega(s') - \hat{V}_\omega(s) \right) \psi_\theta(s,a) = \hat{\delta}(x,\omega)\psi_\theta(s,a) \tag{7}$$

which gives rise to the policy update

$$\theta_{k+1} = \theta_k + \alpha_k v(x_k, \theta_k, \omega_k) \quad \text{with} \quad v(x_k, \theta_k, \omega_k) := \hat{\delta}(x_k, \omega_k)\psi_{\theta_k}(s_k, a_k) \tag{8}$$

where $\alpha_k$ is the stepsize for the actor update.

To ensure convergence when simultaneously performing critic and actor updates, the stepsizes $\alpha_k$ and $\beta_k$ often decay at two different rates, which is referred to the two-timescale AC [17, 18].

## 3 ASYNCHRONOUS ADVANTAGE ACTOR CRITIC WITH TD(0)

To speed up the training process, we implement AC over $N$ workers in a shared memory setting without coordinating among workers — a setting similar to that in A3C [3]. Each worker has its own simulator to perform sampling, and then collaboratively updates the shared policy $\pi_\theta$ using AC updates. As there is no synchronization after each update, the policy used by workers to generate samples may be outdated, which introduces staleness.

**Notations on transition** $(s, a, s')$**.** Since each worker will maintain a separate Markov chain, we thereafter use subscription $t$ in $(s_t, a_t, s_{t+1})$ to indicate the $t$th transition on a Markov chain. We use $k$ to denote the global counter (or iteration), which increases by one whenever a worker finishes the actor and critic updates in the shared memory. We use subscription $(k)$ in $(s_{(k)}, a_{(k)}, s'_{(k)})$ to indicate the transition used in the $k$th update.

Specifically, we initialize $\theta_0, \omega_0$ in the shared memory. Each worker will initialize the simulator with initial state $s_0$. Without coordination, workers will read $\theta, \omega$ in the shared memory. From each worker's view, it then generates sample $(s_t, a_t, s_{t+1})$ by either sampling $s_t$ from $\mu_\theta(\cdot)$, where $\mu_\theta(\cdot)$ is the stationary distribution of an artificial MDP with transition probability measure $\widetilde{\mathcal{P}}(\cdot|s_t, a_t) := \gamma\mathcal{P}(\cdot|s_t, a_t) + (1-\gamma)\eta(\cdot)$, or, sampling $s_t$ from a Markov chain under policy $\pi_\theta$. In both cases, each worker obtains $a_t \sim \pi_\theta(\cdot|s_t)$ and $s_{t+1} \sim \widetilde{\mathcal{P}}(\cdot|s_t, a_t)$. Sampling $s_{t+1}$ from $\widetilde{\mathcal{P}}(\cdot|s_t, a_t)$ can be achieved by sampling $s_{t+1}$ from $\eta(\cdot)$ with probability $1 - \gamma$ and from $\mathcal{P}(\cdot|s_t, a_t)$ otherwise. Once

---

**Algorithm 1** Asynchronous advantage AC with TD(0): each worker's view.

1: **Global initialize:** Global counter $k = 0$, initial $\theta_0, \omega_0$ in the shared memory.
2: **Worker initialize:** Local counter $t = 0$. Obtain initial state $s_0$.
3: **for** $t = 0, 1, 2, \cdots$ **do**
4:       Read $\theta, \omega$ in the shared memory.
5:       **Option 1 (i.i.d. sampling):**
6:           Sample $s_t \sim \mu_\theta(\cdot)$, $a_t \sim \pi_\theta(\cdot|s)$, $s_{t+1} \sim \widetilde{\mathcal{P}}(\cdot|s_t, a_t)$.
7:       **Option 2 (Markovian sampling):**
8:           Sample $a_t \sim \pi_\theta(\cdot|s_t)$, $s_{t+1} \sim \widetilde{\mathcal{P}}(\cdot|s_t, a_t)$.
9:       Compute $\hat{\delta}(x_t, \omega) = r(s_t, a_t, s_{t+1}) + \gamma \hat{V}_\omega(s_{t+1}) - \hat{V}_\omega(s_t)$.
10:      Compute $g(x_t, \omega) = \hat{\delta}(x_t, \omega) \nabla_\omega \hat{V}_\omega(s_t)$.
11:      Compute $v(x_t, \theta, \omega) = \hat{\delta}(x_t, \omega) \psi_\theta(s_t, a_t)$.
12:      In the shared memory, perform update (9).
13: **end for**

---

obtaining $x_t := (s_t, a_t, s_{t+1})$, each worker locally computes the policy gradient $v(x_t, \theta, \omega)$ and the TD(0) update $g(x_t, \omega)$, and then updates the parameters in shared memory asynchronously by

$$\omega_{k+1} = \Pi_{R_\omega} \left( \omega_k + \beta_k g(x_{(k)}, \omega_{k-\tau_k}) \right), \tag{9a}$$

$$\theta_{k+1} = \theta_k + \alpha_k v(x_{(k)}, \theta_{k-\tau_k}, \omega_{k-\tau_k}), \tag{9b}$$

where $\tau_k$ is the delay in the $k$th actor and critic updates. See the A3C with TD(0) in Algorithm 1.

**Sampling distributions.** Since the transition kernel required by the actor and critic updates are different in the *discounted* MDP, it is difficult to design a two-timescale AC algorithm. To address this issue, we adopt the sampling method introduced in the seminal work [6, 31] and the recent work [15, 16], which inevitably introduces bias by sampling from the artificial transition $\widetilde{\mathcal{P}}$ instead of $\mathcal{P}$. However, as we will mention later, this extra bias is small when the discount factor $\gamma$ is close to 1.

**Parallel sampling.** The AC update (6) and (8) uses samples generated "on-the-fly" from the target policy $\pi_\theta$, which brings overhead. Compared with (6) and (8), the A3C-TD(0) update (9) allows parallel sampling from $N$ workers, which is the key to linear speedup. We consider the case where only one worker can update parameters in the shared memory at the same time and the update cannot be interrupted. In practice, (9) can also be performed in a mini-batch fashion.

**Minor differences from A3C [3].** The A3C-TD(0) algorithm resembles the popular A3C method [3]. With $n_{\max}$ denoting the horizon of steps, for $n \in \{1, ..., n_{\max}\}$, A3C iteratively uses $n$-step TD errors to compute actor and critic gradients. In A3C-TD(0), we use the TD(0) method which is the 1-step TD method for actor and critic update. When $n_{\max} = 1$, A3C method reduces to A3C-TD(0). The $n$-step TD method is a hybrid version of the TD(0) method and the Monte-Carlo method. The A3C method with Monte-Carlo sampling is essentially the delayed policy gradient method, and thus its convergence follows directly from the delayed SGD. Therefore, we believe that the convergence of the A3C method based on TD(0) in this paper can be easily extended to the convergence of the A3C method with $n$-step TD. We here focus on A3C with TD(0) just for ease of exposition.

## 4   Convergence Analysis of Two-Timescale A3C-TD(0)

In this section, we analyze the convergence of A3C-TD(0) in both i.i.d. and Markovian settings. Throughout this section, the notation $\mathcal{O}(\cdot)$ contains constants that are independent of $N$ and $K_0$.

To analyze the performance of A3C-TD(0), we make the following assumptions.

**Assumption 1.** *There exists $K_0$ such that the delay at each iteration is bounded by $\tau_k \leq K_0, \forall k$.*

Assumption 1 ensures the viability of analyzing the asynchronous update; see the same assumption in e.g., [5, 25]. In practice, the delay usually scales as the number of workers, that is $K_0 = \Theta(N)$.

With $\widetilde{\mathcal{P}}_{\pi_\theta}(s'|s) = \sum_{a \in \mathcal{A}} \widetilde{\mathcal{P}}(s'|s, a) \pi_\theta(a|s)$, we define that:

$$A_{\theta,\phi} := \underset{s \sim \mu_\theta, s' \sim \widetilde{\mathcal{P}}_{\pi_\theta}}{\mathbb{E}} [\phi(s)(\gamma \phi(s') - \phi(s))^\top], \quad b_{\theta,\phi} := \underset{s \sim \mu_\theta, a \sim \pi_\theta, s' \sim \widetilde{\mathcal{P}}}{\mathbb{E}} [r(s, a, s') \phi(s)]. \tag{10}$$

It is known that for a given $\theta$, the stationary point $\omega_\theta^*$ of the TD(0) update in Algorithm 1 satisfies

$$A_{\theta,\phi}\omega_\theta^* + b_{\theta,\phi} = 0. \tag{11}$$

**Assumption 2.** *For all $s \in \mathcal{S}$, the feature vector $\phi(s)$ is normalized so that $\|\phi(s)\|_2 \leq 1$. For all $\gamma \in [0,1]$ and $\theta \in \mathbb{R}^d$, $A_{\theta,\phi}$ is negative definite and its max eigenvalue is upper bounded by $-\lambda$.*

Assumption 2 is common in analyzing TD with linear function approximation; see e.g., [17, 32, 33]. With this assumption, $A_{\theta,\phi}$ is invertible, so we have $\omega_\theta^* = -A_{\theta,\phi}^{-1}b_{\theta,\phi}$. Define $R_\omega := r_{\max}/\lambda$, then we have $\|\omega_\theta^*\|_2 \leq R_\omega$. It justifies the projection introduced in Algorithm 1. In practice, the projection radius $R_\omega$ can be estimated online by methods proposed in [32, Section 8.2] or [34, Lemma 1].

**Assumption 3.** *For any $\theta, \theta' \in \mathbb{R}^d$, $s \in \mathcal{S}$ and $a \in \mathcal{A}$, there exist constants such that: i) $\|\psi_\theta(s,a)\|_2 \leq C_\psi$; ii) $\|\psi_\theta(s,a) - \psi_{\theta'}(s,a)\|_2 \leq L_\psi\|\theta - \theta'\|_2$; iii) $|\pi_\theta(a|s) - \pi_{\theta'}(a|s)| \leq L_\pi\|\theta - \theta'\|_2$.*

Assumption 3 is common in analyzing policy gradient-type algorithms which has also been made by e.g., [34, 35, 36]. This assumption holds for many policy parameterization methods such as tabular softmax policy [36], Gaussian policy [37] and Boltzmann policy [31].

**Assumption 4.** *For any $\theta \in \mathbb{R}^d$, the Markov chain under policy $\pi_\theta$ and transition kernel $\mathcal{P}(\cdot|s,a)$ or $\widetilde{\mathcal{P}}(\cdot|s,a)$ is irreducible and aperiodic. Then there exist constants $\kappa > 0$ and $\rho \in (0,1)$ such that*

$$\sup_{s \in \mathcal{S}} \ d_{TV}\left(\mathbb{P}(s_t \in \cdot|s_0 = s, \pi_\theta), \mu_\theta\right) \leq \kappa\rho^t, \quad \forall t \tag{12}$$

*where $\mu_\theta$ is the stationary state distribution under $\pi_\theta$, and $s_t$ is the state of Markov chain at time $t$.*

Assumption 4 assumes the Markov chain mixes at a geometric rate; see also [32, 33]. The stationary distribution $\mu_\theta$ of an artificial Markov chain with transition $\widetilde{\mathcal{P}}$ is the same as the discounted visitation measure $d_\theta$ of the Markov chain with transition $\mathcal{P}$ [6]. This means that if we sample according to $a_t \sim \pi_\theta(\cdot|s_t), s_{t+1} \sim \widetilde{\mathcal{P}}(\cdot|s_t, a_t)$, the marginal distribution of $(s_t, a_t)$ will converge to the discounted state-action visitation measure $d'_\theta(s,a)$, which allows us to control the gradient bias.

### 4.1 LINEAR SPEEDUP RESULT WITH I.I.D. SAMPLING

In this section, we consider A3C-TD(0) under the i.i.d. sampling setting, which is widely used for analyzing RL algorithms; see e.g., [13, 18, 38].

We first give the convergence result of critic update as follows.

**Theorem 1** (Critic convergence). *Suppose Assumptions 1–4 hold. Consider Algorithm 1 with i.i.d. sampling and $\hat{V}_\omega(s) = \phi(s)^\top\omega$. Select step size $\alpha_k = \frac{c_1}{(1+k)^{\sigma_1}}$, $\beta_k = \frac{c_2}{(1+k)^{\sigma_2}}$, where $0 < \sigma_2 < \sigma_1 < 1$ and $c_1, c_2$ are positive constants. Then it holds that*

$$\frac{1}{K}\sum_{k=1}^{K}\mathbb{E}\left\|\omega_k - \omega_{\theta_k}^*\right\|_2^2 = \mathcal{O}\left(\frac{1}{K^{1-\sigma_2}}\right) + \mathcal{O}\left(\frac{1}{K^{2(\sigma_1-\sigma_2)}}\right) + \mathcal{O}\left(\frac{K_0^2}{K^{2\sigma_2}}\right) + \mathcal{O}\left(\frac{K_0}{K^{\sigma_1}}\right) + \mathcal{O}\left(\frac{1}{K^{\sigma_2}}\right). \tag{13}$$

Different from async-SGD (e.g., [9]), the optimal critic parameter $\omega_\theta^*$ is constantly drifting as $\theta$ changes at each iteration. This necessitates setting $\sigma_1 > \sigma_2$ to make the policy change slower than the critic, which can be observed in the second term in (13). If $\sigma_1 > \sigma_2$, then the policy is static relative to the critic in an asymptotic sense.

To introduce the convergence of actor update, we first define the critic approximation error as

$$\epsilon_{\mathrm{app}} := \max_{\theta \in \mathbb{R}^d}\sqrt{\mathop{\mathbb{E}}_{s \sim \mu_\theta}|V_{\pi_\theta}(s) - \hat{V}_{\omega_\theta^*}(s)|^2} \leq \epsilon_{\mathrm{fa}} + \epsilon_{\mathrm{sp}}, \tag{14}$$

where $\mu_\theta$ is the stationary distribution under $\pi_\theta$ and $\widetilde{\mathcal{P}}$. The error $\epsilon_{\mathrm{app}}$ captures the quality of the critic approximation under Algorithm 1. It can be further decomposed into the function approximation error $\epsilon_{\mathrm{fa}}$, which is common in analyzing AC with function approximation [14, 15, 17], and the sampling error $\epsilon_{\mathrm{sp}} = \mathcal{O}(1 - \gamma)$, which is unique in analyzing two-timescale AC for a discounted MDP. The error $\epsilon_{\mathrm{app}}$ is small when the value function approximation is accurate and the discounting factor $\gamma$ is close to 1; see the detailed derivations in Lemma 7 of supplementary material. Now we are ready to give the actor convergence.

**Theorem 2** (Actor convergence). *Under the same assumptions of Theorem 1, select step size $\alpha_k = \frac{c_1}{(1+k)^{\sigma_1}}$, $\beta_k = \frac{c_2}{(1+k)^{\sigma_2}}$, where $0 < \sigma_2 < \sigma_1 < 1$ and $c_1, c_2$ are positive constants. Then it holds that*

$$\frac{1}{K}\sum_{k=1}^{K}\mathbb{E}\left\|\nabla J(\theta_k)\right\|_2^2 = \mathcal{O}\left(\frac{1}{K^{1-\sigma_1}}\right) + \mathcal{O}\left(\frac{K_0}{K^{\sigma_1}}\right) + \mathcal{O}\left(\frac{K_0^2}{K^{2\sigma_2}}\right) + \mathcal{O}\left(\frac{1}{K}\sum_{k=1}^{K}\mathbb{E}\left\|\omega_k - \omega_{\theta_k}^*\right\|_2^2\right) + \mathcal{O}(\epsilon_{\text{app}}).$$
(15)

Different from the analysis of async-SGD, in actor update, the stochastic gradient $v(x, \theta, \omega)$ is biased because of inexact value function approximation. The bias introduced by the critic optimality gap and the function approximation error correspond to the last two terms in (15).

In Theorem 1 and Theorem 2, optimizing $\sigma_1$ and $\sigma_2$ gives the following convergence rate.

**Corollary 1** (Linear speedup). *Given Theorem 1 and Theorem 2, select $\sigma_1 = \frac{3}{5}$ and $\sigma_2 = \frac{2}{5}$. If we further assume $K_0 = \mathcal{O}(K^{\frac{1}{5}})$, then it holds that*

$$\frac{1}{K}\sum_{k=1}^{K}\mathbb{E}\left\|\nabla J(\theta_k)\right\|_2^2 = \mathcal{O}\left(K^{-\frac{2}{5}}\right) + \mathcal{O}(\epsilon_{\text{app}})$$
(16)

*where $\mathcal{O}(\cdot)$ contains constants that are independent of $N$ and $K_0$.*

By setting the first term in (16) to $\epsilon$, we get the total iteration complexity to reach $\epsilon$-accuracy is $\mathcal{O}(\epsilon^{-2.5})$. Since each iteration only uses one sample (one transition), it also implies a total sample complexity of $\mathcal{O}(\epsilon^{-2.5})$. Then the average sample complexity per worker is $\mathcal{O}(\epsilon^{-2.5}/N)$ which indicates linear speedup in (1). Intuitively, the negative effect of parameter staleness introduced by parallel asynchrony vanishes asymptotically, which implies linear speedup in terms of convergence.

## 4.2 CONVERGENCE RESULT WITH MARKOVIAN SAMPLING

**Theorem 3** (Critic convergence). *Suppose Assumptions 1–4 hold. Consider Algorithm 1 with Markovian sampling and $\hat{V}_\omega(s) = \phi(s)^\top \omega$. Select step size $\alpha_k = \frac{c_1}{(1+k)^{\sigma_1}}$, $\beta_k = \frac{c_2}{(1+k)^{\sigma_2}}$, where $0 < \sigma_2 < \sigma_1 < 1$ and $c_1, c_2$ are positive constants. Then it holds that*

$$\frac{1}{K}\sum_{k=1}^{K}\mathbb{E}\left\|\omega_k - \omega_{\theta_k}^*\right\|_2^2 = \mathcal{O}\left(\frac{1}{K^{1-\sigma_2}}\right) + \mathcal{O}\left(\frac{1}{K^{2(\sigma_1-\sigma_2)}}\right) + \mathcal{O}\left(\frac{K_0^2}{K^{2\sigma_2}}\right) + \mathcal{O}\left(\frac{K_0^2\log^2 K}{K^{\sigma_1}}\right) + \mathcal{O}\left(\frac{K_0\log K}{K^{\sigma_2}}\right).$$
(17)

The following theorem gives the convergence rate of actor update in Algorithm 1.

**Theorem 4** (Actor convergence). *Under the same assumptions of Theorem 3, select step size $\alpha_k = \frac{c_1}{(1+k)^{\sigma_1}}$, $\beta_k = \frac{c_2}{(1+k)^{\sigma_2}}$, where $0 < \sigma_2 < \sigma_1 < 1$ and $c_1, c_2$ are positive constants. Then it holds that*

$$\frac{1}{K}\sum_{k=1}^{K}\mathbb{E}\left\|\nabla J(\theta_k)\right\|_2^2 = \mathcal{O}\left(\frac{1}{K^{1-\sigma_1}}\right) + \mathcal{O}\left(\frac{K_0^2\log^2 K}{K^{\sigma_1}}\right) + \mathcal{O}\left(\frac{K_0^2}{K^{2\sigma_2}}\right) + \mathcal{O}\left(\frac{1}{K}\sum_{k=1}^{K}\mathbb{E}\left\|\omega_k - \omega_{\theta_k}^*\right\|_2^2\right) + \mathcal{O}(\epsilon_{\text{app}}).$$
(18)

*Assume $K_0 = \mathcal{O}(K^{\frac{1}{5}})$. Given Theorem 3, select $\sigma_1 = \frac{3}{5}$ and $\sigma_2 = \frac{2}{5}$, then it holds that*

$$\frac{1}{K}\sum_{k=1}^{K}\mathbb{E}\left\|\nabla J(\theta_k)\right\|_2^2 = \widetilde{\mathcal{O}}\left(K_0 K^{-\frac{2}{5}}\right) + \mathcal{O}(\epsilon_{\text{app}}),$$
(19)

*where $\widetilde{\mathcal{O}}(\cdot)$ hides constants and the logarithmic order of $K$.*

With Markovian sampling, the stochastic gradients $g(x, \omega)$ and $v(x, \theta, \omega)$ are biased, and the bias decreases as the Markov chain mixes. The mixing time corresponds to the logarithmic term $\log K$ in (17) and (18). Because of asynchrony, at a given iteration, workers collect different number of samples and their chains mix to different degrees. The worker with the slowest mixing chain will determine the rate of convergence. The product of $K_0$ and $\log K$ in (17) and (18) appears due to the slowest mixing chain. As the last term in (17) dominates other terms asymptotically, the convergence rate reduces as the number of workers increases. While the theoretical linear speedup is difficult to establish in the Markovian setting, we will empirically demonstrate it in Section 5.2.

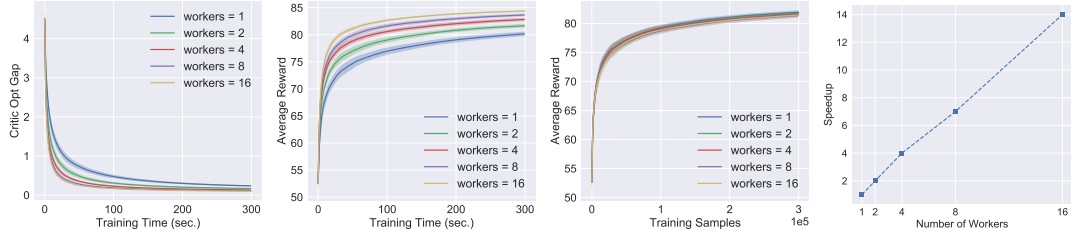

Figure 1: Convergence results of A3C-TD(0) with i.i.d. sampling in synthetic environment.

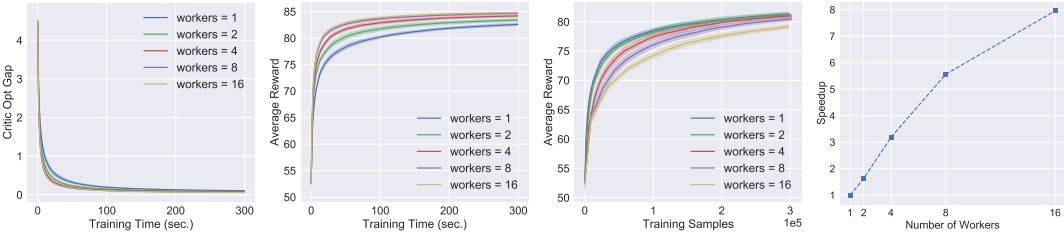

Figure 2: Convergence results of A3C-TD(0) with Markovian sampling in synthetic environment.

## 5 NUMERICAL EXPERIMENTS

We test the speedup performance of A3C-TD(0) on both synthetically generated and OpenAI Gym environments. **The settings, parameters, and codes are provided in supplementary material**.

### 5.1 A3C-TD(0) IN SYNTHETIC ENVIRONMENT

To verify the theoretical result, we tested A3C-TD(0) with linear value function approximation in a synthetic environment. We use tabular softmax policy parameterization [36], which satisfies Assumption 3. The MDP has a state space $|\mathcal{S}| = 100$, an discrete action space of $|\mathcal{A}| = 5$. Each state feature has a dimension of 10. Elements of the transition matrix, the reward and the state features are randomly sampled from a uniform distribution over $(0, 1)$. We evaluate the convergence of actor and critic respectively with the running average of test reward and critic optimality gap $\|\omega_k - \omega^*_{\theta_k}\|_2$.

Figures 1 and 2 show the training time and sample complexity of running A3C-TD(0) with i.i.d. sampling and Markovian sampling respectively. The speedup plot is measured by the number of samples needed to achieve a target running average reward under different number of workers. All the results are average over 10 Monte-Carlo runs. Figure 1 shows that the sample complexity of A3C-TD(0) stays about the same with different number of workers under i.i.d. sampling. Also, it can be observed from the speedup plot of Figure 1 that the A3C-TD(0) achieves roughly linear speedup with i.i.d. sampling, which is consistent with Corollary 1. The speedup of A3C-TD(0) with Markovian sampling shown in Figure 2 is roughly linear when number of workers is small.

### 5.2 A3C-TD(0) IN OPENAI GYM ENVIRONMENTS

We have also tested A3C-TD(0) with neural network parametrization in the classic control (Carpole) environment and the Atari game (Seaquest and Beamrider) environments. In Figures 3-5, each curve is generated by averaging over 5 Monte-Carlo runs with $95\%$ confidence interval. Figures 3–5 show the speedup of A3C-TD(0) under different number of workers, where the average reward is computed by taking the running average of test rewards. The speedup and runtime speedup plots are respectively measured by the number of samples and training time needed to achieve a target running average reward under different number of workers. Although not justified theoretically, Figures 3–5 suggest that the sample complexity speedup is roughly linear, and the runtime speedup slightly degrades when the number of workers increases. This is partially due to our hardware limit. Similar observation has also been obtained in async-SGD [9].

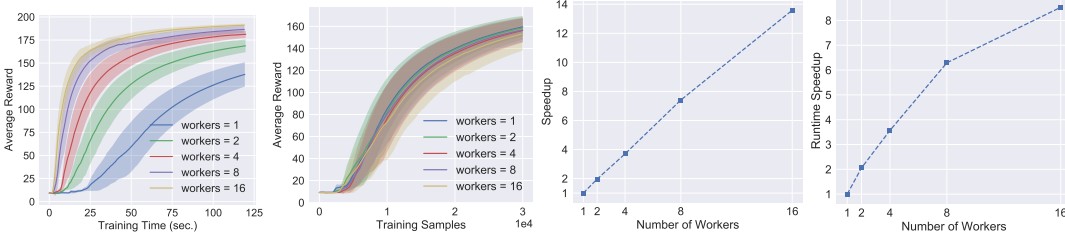

Figure 3: Speedup of A3C-TD(0) in OpenAI gym classic control task (Carpole).

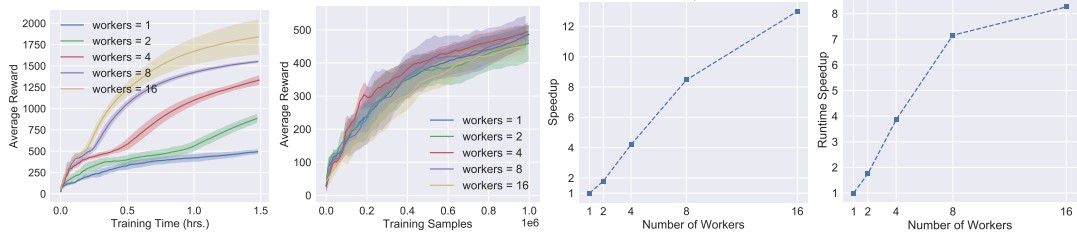

Figure 4: Speedup of A3C-TD(0) in OpenAI Gym Atari game (Seaquest).

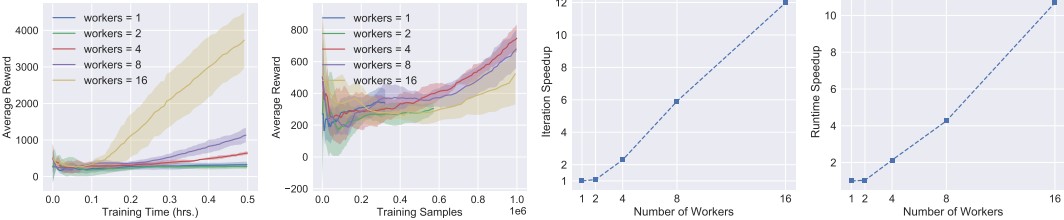

Figure 5: Speedup of A3C-TD(0) in OpenAI Gym Atari game (Beamrider).

## 6 CONCLUSIONS

This paper revisits the A3C algorithm with TD(0) for the critic update, termed A3C-TD(0). With linear value function approximation, the convergence of the A3C-TD(0) algorithm has been established under both i.i.d. and Markovian sampling settings. Under i.i.d. sampling, A3C-TD(0) achieves linear speedup compared to the best-known sample complexity of two-timescale AC, theoretically justifying the benefit of parallelism and asynchrony for the first time. Under Markov sampling, such a linear speedup can be observed in most classic benchmark tasks.

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

# Supplementary Material

## A  PRELIMINARY LEMMAS

### A.1  GEOMETRIC MIXING

The operation $p \otimes q$ denotes the tensor product between two distributions $p(x)$ and $q(y)$, i.e. $(p \otimes q)(x, y) = p(x) \cdot q(y)$.

**Lemma 1.** *Suppose Assumption 4 holds for a Markov chain generated by the rule $a_t \sim \pi_\theta(\cdot|s_t)$, $s_{t+1} \sim \widetilde{\mathcal{P}}(\cdot|s_t, a_t)$. For any $\theta \in \mathbb{R}^d$, we have*

$$\sup_{s_0 \in \mathcal{S}} d_{TV}\left(\mathbb{P}((s_t, a_t, s_{t+1}) \in \cdot|s_0, \pi_\theta), \mu_\theta \otimes \pi_\theta \otimes \widetilde{\mathcal{P}}\right) \leq \kappa\rho^t. \tag{20}$$

*where $\mu_\theta(\cdot)$ is the stationary distribution with policy $\pi_\theta$ and transition kernel $\widetilde{\mathcal{P}}(\cdot|s, a)$.*

*Proof.* We start with

$$\sup_{s_0 \in \mathcal{S}} d_{TV}\left(\mathbb{P}((s_t, a_t, s_{t+1}) = \cdot|s_0, \pi_\theta), \mu_\theta \otimes \pi_\theta \otimes \widetilde{\mathcal{P}}\right)$$

$$= \sup_{s_0 \in \mathcal{S}} d_{TV}\left(\mathbb{P}(s_t = \cdot|s_0, \pi_\theta) \otimes \pi_\theta \otimes \widetilde{\mathcal{P}}, \mu_\theta \otimes \pi_\theta \otimes \widetilde{\mathcal{P}}\right)$$

$$= \sup_{s_0 \in \mathcal{S}} \frac{1}{2} \int_{s \in \mathcal{S}} \sum_{a \in \mathcal{A}} \int_{s' \in \mathcal{S}} |\mathbb{P}(s_t = ds|s_0, \pi_\theta)\pi_\theta(a|s)\widetilde{\mathcal{P}}(ds'|s, a) - \mu_\theta(ds)\pi_\theta(a|s)\widetilde{\mathcal{P}}(ds'|s, a)|$$

$$= \sup_{s_0 \in \mathcal{S}} \frac{1}{2} \int_{s \in \mathcal{S}} |\mathbb{P}(s_t = ds|s_0, \pi_\theta) - \mu_\theta(ds)| \sum_{a \in \mathcal{A}} \pi_\theta(a|s) \int_{s' \in \mathcal{S}} \widetilde{\mathcal{P}}(ds'|s, a)$$

$$= \sup_{s_0 \in \mathcal{S}} d_{TV}\left(\mathbb{P}(s_t \in \cdot|s_0, \pi_\theta), \mu_\theta\right)$$

$$\leq \kappa\rho^t,$$

which completes the proof. □

For the use in the later proof, given $K > 0$, we first define $m_K$ as:

$$m_K := \min\left\{m \in \mathbb{N}^+ \,|\, \kappa\rho^{m-1} \leq \min\{\alpha_k, \beta_k\}\right\}, \tag{21}$$

where $\kappa$ and $\rho$ are constants defined in (4). $m_K$ is the minimum number of samples needed for the Markov chain to approach the stationary distribution so that the bias incurred by the Markovian sampling is small enough.

### A.2  AUXILIARY MARKOV CHAIN

The auxiliary Markov chain is a virtual Markov chain with no policy drifting — a technique developed in [34] to analyze stochastic approximation algorithms in non-stationary settings.

**Lemma 2.** *Under Assumption 1 and Assumption 3, consider the update (9) in Algorithm 1 with Markovian sampling. For a given number of samples $m$, consider the Markov chain of the worker that contributes to the $k$th update:*

$$s_{t-m} \xrightarrow{\theta_{k-d_m}} a_{t-m} \xrightarrow{\widetilde{\mathcal{P}}} s_{t-m+1} \xrightarrow{\theta_{k-d_{m-1}}} a_{t-m+1} \cdots s_{t-1} \xrightarrow{\theta_{k-d_1}} a_{t-1} \xrightarrow{\widetilde{\mathcal{P}}} s_t \xrightarrow{\theta_{k-d_0}} a_t \xrightarrow{\widetilde{\mathcal{P}}} s_{t+1},$$

*where $(s_t, a_t, s_{t+1}) = (s_{(k)}, a_{(k)}, s'_{(k)})$, and $\{d_j\}_{j=0}^m$ is some increasing sequence with $d_0 := \tau_k$.*

*Given $(s_{t-m}, a_{t-m}, s_{t-m+1})$ and $\theta_{k-d_m}$, we construct its auxiliary Markov chain by repeatedly applying $\pi_{\theta_{k-d_m}}$:*

$$s_{t-m} \xrightarrow{\theta_{k-d_m}} a_{t-m} \xrightarrow{\widetilde{\mathcal{P}}} s_{t-m+1} \xrightarrow{\theta_{k-d_m}} \widetilde{a}_{t-m+1} \cdots \widetilde{s}_{t-1} \xrightarrow{\theta_{k-d_m}} \widetilde{a}_{t-1} \xrightarrow{\widetilde{\mathcal{P}}} \widetilde{s}_t \xrightarrow{\theta_{k-d_m}} \widetilde{a}_t \xrightarrow{\widetilde{\mathcal{P}}} \widetilde{s}_{t+1}.$$

*Define $x_t := (s_t, a_t, s_{t+1})$, then we have:*

$$d_{TV}\left(\mathbb{P}(x_t \in \cdot|\theta_{k-d_m}, s_{t-m+1}), \mathbb{P}(\widetilde{x}_t \in \cdot|\theta_{k-d_m}, s_{t-m+1})\right)$$

$$\leq \frac{1}{2}|\mathcal{A}|L_\pi \sum_{i=\tau_k}^{d_m} \mathbb{E}\left[\|\theta_{k-i} - \theta_{k-d_m}\|_2|\theta_{k-d_m}, s_{t-m+1}\right]. \tag{22}$$

*Proof.* Throughout the lemma, all expectations and probabilities are conditioned on $\theta_{k-d_m}$ and $s_{t-m+1}$. We omit this condition for convenience.

First we have

$$
\begin{aligned}
&d_{TV}\left(\mathbb{P}(s_{t+1}\in\cdot),\mathbb{P}(\widetilde{s}_{t+1}\in\cdot)\right)\\
&=\frac{1}{2}\int_{s'\in\mathcal{S}}|\mathbb{P}(s_{t+1}=ds')-\mathbb{P}(\widetilde{s}_{t+1}=ds')|\\
&=\frac{1}{2}\int_{s'\in\mathcal{S}}\left|\int_{s\in\mathcal{S}}\sum_{a\in\mathcal{A}}\mathbb{P}(s_t=ds,a_t=a,s_{t+1}=ds')-\mathbb{P}(\widetilde{s}_t=ds,\widetilde{a}_t=a,\widetilde{s}_{t+1}=ds')\right|\\
&\le\frac{1}{2}\int_{s'\in\mathcal{S}}\int_{s\in\mathcal{S}}\sum_{a\in\mathcal{A}}|\mathbb{P}(s_t=ds,a_t=a,s_{t+1}=ds')-\mathbb{P}(\widetilde{s}_t=ds,\widetilde{a}_t=a,\widetilde{s}_{t+1}=ds')|\\
&=\frac{1}{2}\int_{s\in\mathcal{S}}\sum_{a\in\mathcal{A}}\int_{s'\in\mathcal{S}}|\mathbb{P}(s_t=ds,a_t=a,s_{t+1}=ds')-\mathbb{P}(\widetilde{s}_t=ds,\widetilde{a}_t=a,\widetilde{s}_{t+1}=ds')|\\
&=d_{TV}\left(\mathbb{P}(x_t\in\cdot),\mathbb{P}(\widetilde{x}_t\in\cdot)\right),
\end{aligned}
\tag{23}
$$

where the last second equality is due to Tonelli's theorem. Next we have

$$
\begin{aligned}
&d_{TV}\left(\mathbb{P}(x_t\in\cdot),\mathbb{P}(\widetilde{x}_t\in\cdot)\right)\\
&=\frac{1}{2}\int_{s\in\mathcal{S}}\sum_{a\in\mathcal{A}}\int_{s'\in\mathcal{S}}|\mathbb{P}(s_t=ds,a_t=a,s_{t+1}=ds')-\mathbb{P}(\widetilde{s}_t=ds,\widetilde{a}_t=a,\widetilde{s}_{t+1}=ds')|\\
&=\frac{1}{2}\int_{s\in\mathcal{S}}\sum_{a\in\mathcal{A}}|\mathbb{P}(s_t=ds,a_t=a)-\mathbb{P}(\widetilde{s}_t=ds,\widetilde{a}_t=a)|\int_{s'\in\mathcal{S}}\widetilde{\mathcal{P}}(s_{t+1}=ds'|s_t=ds,a_t=a)\\
&=\frac{1}{2}\int_{s\in\mathcal{S}}\sum_{a\in\mathcal{A}}|\mathbb{P}(s_t=ds,a_t=a)-\mathbb{P}(\widetilde{s}_t=ds,\widetilde{a}_t=a)|\\
&=d_{TV}\left(\mathbb{P}\left((s_t,a_t)\in\cdot\right),\mathbb{P}\left((\widetilde{s}_t,\widetilde{a}_t)\in\cdot\right)\right).
\end{aligned}
\tag{24}
$$

Due to the fact that $\theta_{k-\tau_k}$ is dependent on $s_t$, we need to write $\mathbb{P}(s_t,a_t)$ as

$$
\begin{aligned}
\mathbb{P}(s_t,a_t)&=\int_{\theta_{k-\tau_k}\in\mathbb{R}^d}\mathbb{P}(s_t,\theta_{k-\tau_k},a_t)\\
&=\int_{\theta\in\mathbb{R}^d}\mathbb{P}(s_t)\mathbb{P}(\theta_{k-\tau_k}=d\theta|s_t)\pi_{\theta_{k-\tau_k}}(a_t|s_t)\\
&=\mathbb{P}(s_t)\int_{\theta\in\mathbb{R}^d}\mathbb{P}(\theta_{k-\tau_k}=d\theta|s_t)\pi_{\theta_{k-\tau_k}}(a_t|s_t)\\
&=\mathbb{P}(s_t)\,\mathbb{E}[\pi_{\theta_{k-\tau_k}}(a_t|s_t)|s_t].
\end{aligned}
$$

Then we have

$$
\begin{aligned}
&d_{TV}\left(\mathbb{P}\left((s_t,a_t)\in\cdot\right),\mathbb{P}\left((\widetilde{s}_t,\widetilde{a}_t)\in\cdot\right)\right)\\
&=\frac{1}{2}\int_{s\in\mathcal{S}}\sum_{a\in\mathcal{A}}\left|\mathbb{P}(s_t=ds)\,\mathbb{E}[\pi_{\theta_{k-\tau_k}}(a_t=a|s_t=ds)|s_t=ds]-\mathbb{P}(\widetilde{s}_t=ds)\pi_{\theta_{k-d_m}}(\widetilde{a}_t=a|\widetilde{s}_t=ds)\right|\\
&\le\frac{1}{2}\int_{s\in\mathcal{S}}\sum_{a\in\mathcal{A}}\left|\mathbb{P}(s_t=ds)\,\mathbb{E}[\pi_{\theta_{k-\tau_k}}(a_t=a|s_t=ds)|s_t=ds]-\mathbb{P}(s_t=ds)\pi_{\theta_{k-d_m}}(a_t=a|s_t=ds)\right|\\
&\quad+\frac{1}{2}\int_{s\in\mathcal{S}}\sum_{a\in\mathcal{A}}\left|\mathbb{P}(s_t=ds)\pi_{\theta_{k-d_m}}(\widetilde{a}_t=a|\widetilde{s}_t=ds)-\mathbb{P}(\widetilde{s}_t=ds)\pi_{\theta_{k-d_m}}(\widetilde{a}_t=a|\widetilde{s}_t=ds)\right|\\
&=\frac{1}{2}\int_{s\in\mathcal{S}}\mathbb{P}(s_t=ds)\sum_{a\in\mathcal{A}}\left|\mathbb{E}[\pi_{\theta_{k-\tau_k}}(a_t=a|s_t=ds)|s_t=ds]-\pi_{\theta_{k-d_m}}(a_t=a|s_t=ds)\right|\\
&\quad+\frac{1}{2}\int_{s\in\mathcal{S}}|\mathbb{P}(s_t=ds)-\mathbb{P}(\widetilde{s}_t=ds)|.
\end{aligned}
\tag{25}
$$

Using Jensen's inequality, we have

$$
\begin{aligned}
d_{TV} &\left( \mathbb{P}\left( (s_t, a_t) \in \cdot \right), \mathbb{P}\left( (\widetilde{s}_t, \widetilde{a}_t) \in \cdot \right) \right) \\
&\leq \frac{1}{2} \int_{s \in \mathcal{S}} \mathbb{P}(s_t = ds) \sum_{a \in \mathcal{A}} \mathbb{E}\left[ \left| \pi_{\theta_{k-\tau_k}}(a_t = a | s_t = ds) - \pi_{\theta_{k-d_m}}(a_t = a | s_t = ds) \right| \middle| s_t = ds \right] \\
&\quad + \frac{1}{2} \int_{s \in \mathcal{S}} \left| \mathbb{P}(s_t = ds) - \mathbb{P}(\widetilde{s}_t = ds) \right| \\
&\leq \frac{1}{2} \int_{s \in \mathcal{S}} \mathbb{P}(s_t = ds) \sum_{a \in \mathcal{A}} \mathbb{E}\left[ \| \theta_{k-\tau_k} - \theta_{k-d_m} \|_2 \middle| s_t = ds \right] + \frac{1}{2} \int_{s \in \mathcal{S}} \left| \mathbb{P}(s_t = ds) - \mathbb{P}(\widetilde{s}_t = ds) \right| \\
&= \frac{1}{2} |\mathcal{A}| L_\pi \, \mathbb{E} \, \| \theta_{k-\tau_k} - \theta_{k-d_m} \|_2 + d_{TV}\left( \mathbb{P}(s_t \in \cdot), \mathbb{P}(\widetilde{s}_t \in \cdot) \right)
\end{aligned}
\tag{26}
$$

where the last inequality follows Assumption 3.

Now we start to prove (22).

$$
\begin{aligned}
d_{TV}\left( \mathbb{P}(x_t \in \cdot), \mathbb{P}(\widetilde{x}_t \in \cdot) \right) &\overset{(24)}{=} d_{TV}\left( \mathbb{P}((s_t, a_t) \in \cdot), \mathbb{P}((\widetilde{s}_t, \widetilde{a}_t) \in \cdot) \right) \\
&\overset{(25)}{\leq} d_{TV}\left( \mathbb{P}(s_t \in \cdot), \mathbb{P}(\widetilde{s}_t \in \cdot) \right) + \frac{1}{2} |\mathcal{A}| L_\pi \, \mathbb{E} \, \| \theta_{k-\tau_k} - \theta_{k-d_m} \|_2 \\
&\overset{(23)}{\leq} d_{TV}\left( \mathbb{P}(x_{t-1} \in \cdot), \mathbb{P}(\widetilde{x}_{t-1} \in \cdot) \right) + \frac{1}{2} |\mathcal{A}| L_\pi \, \mathbb{E} \, \| \theta_{k-\tau_k} - \theta_{k-d_m} \|_2 .
\end{aligned}
$$

Now we have

$$
d_{TV}\left( \mathbb{P}(x_t \in \cdot), \mathbb{P}(\widetilde{x}_t \in \cdot) \right) \leq d_{TV}\left( \mathbb{P}(x_{t-1} \in \cdot), \mathbb{P}(\widetilde{x}_{t-1} \in \cdot) \right) + \frac{1}{2} |\mathcal{A}| L_\pi \, \mathbb{E} \, \| \theta_{k-\tau_k} - \theta_{k-d_m} \|_2 .
\tag{27}
$$

Since $d_{TV}\left( \mathbb{P}(x_{t-m} \in \cdot), \mathbb{P}(x_{t-m} \in \cdot) \right) = 0$, recursively applying (27) for $\{t-1, ..., t-m\}$ gives

$$
\begin{aligned}
d_{TV}\left( \mathbb{P}(x_t \in \cdot), \mathbb{P}(\widetilde{x}_t \in \cdot) \right) &\leq \frac{1}{2} |\mathcal{A}| L_\pi \sum_{j=0}^{m} \mathbb{E} \, \| \theta_{k-d_j} - \theta_{k-d_m} \|_2 \\
&\leq \frac{1}{2} |\mathcal{A}| L_\pi \sum_{i=\tau_k}^{d_m} \mathbb{E} \, \| \theta_{k-i} - \theta_{k-d_m} \|_2 ,
\end{aligned}
$$

which completes the proof. $\qquad\qquad\square$

### A.3 LIPSCHITZ CONTINUITY OF VALUE FUNCTION

**Lemma 3.** *Suppose Assumption 3 holds. For any $\theta_1, \theta_2 \in \mathbb{R}^d$ and $s \in \mathcal{S}$, we have*

$$
\| \nabla V_{\pi_{\theta_1}}(s) \|_2 \leq L_V,
\tag{28a}
$$
$$
| V_{\pi_{\theta_1}}(s) - V_{\pi_{\theta_2}}(s) | \leq L_V \| \theta_1 - \theta_2 \|_2,
\tag{28b}
$$

*where the constant is $L_V := C_\psi r_{\max} / (1 - \gamma)$ with $C_\psi$ defined as in Assumption 3.*

*Proof.* First we have

$$
\begin{aligned}
Q_\pi(s, a) &= \mathbb{E}\left[ \sum_{t=0}^{\infty} \gamma^t r(s_t, a_t, s_{t+1}) | s_0 = s, a_0 = a \right] \\
&\leq \sum_{t=0}^{\infty} \gamma^t r_{\max} = \frac{r_{\max}}{1 - \gamma}.
\end{aligned}
$$

By the policy gradient theorem [7], we have

$$
\begin{aligned}
\|\nabla V_{\pi_{\theta_1}}(s)\|_2 &= \left\| \mathbb{E}\left[ Q_{\pi_{\theta_1}}(s,a)\psi_{\theta_1}(s,a) \right] \right\|_2 \\
&\leq \mathbb{E}\left\| Q_{\pi_{\theta_1}}(s,a)\psi_{\theta_1}(s,a) \right\|_2 \\
&\leq \mathbb{E}\left[ |Q_{\pi_{\theta_1}}(s,a)| \|\psi_{\theta_1}(s,a)\|_2 \right] \\
&\leq \frac{r_{\max}}{1-\gamma} C_\psi,
\end{aligned}
$$

where the first inequality is due to Jensen's inequality, and the last inequality follows Assumption 3 and the fact that $Q_\pi(s,a) \leq \frac{r_{\max}}{1-\gamma}$. By the mean value theorem, we immediately have

$$
|V_{\pi_{\theta_1}}(s) - V_{\pi_{\theta_2}}(s)| \leq \sup_{\theta_1 \in \mathbb{R}^d} \left\| \nabla V_{\pi_{\theta_1}}(s) \right\|_2 \|\theta_1 - \theta_2\|_2 = L_V \|\theta_1 - \theta_2\|_2,
$$

which completes the proof. $\qquad\square$

### A.4 LIPSCHITZ CONTINUITY OF POLICY GRADIENT

We give a proposition regarding the $L_J$-Lipschitz of the policy gradient under proper assumptions, which has been shown by [35].

**Proposition 1.** *Suppose Assumption 3 and 4 hold. For any $\theta, \theta' \in \mathbb{R}^d$, we have $\|\nabla J(\theta) - \nabla J(\theta')\|_2 \leq L_J \|\theta - \theta'\|_2$, where $L_J$ is a positive constant.*

### A.5 LIPSCHITZ CONTINUITY OF OPTIMAL CRITIC PARAMETER

We provide a justification for Lipschitz continuity of $\omega_\theta^*$ in the next proposition.

**Proposition 2.** *Suppose Assumption 3 and 4 hold. For any $\theta_1, \theta_2 \in \mathbb{R}^d$, we have*

$$
\|\omega_{\theta_1}^* - \omega_{\theta_2}^*\|_2 \leq L_\omega \|\theta_1 - \theta_2\|_2,
$$

*where $L_\omega := 2r_{\max}|\mathcal{A}|L_\pi(\lambda^{-1} + \lambda^{-2}(1+\gamma))(1 + \log_\rho \kappa^{-1} + (1-\rho)^{-1})$.*

*Proof.* We use $A_1$, $A_2$, $b_1$ and $b_2$ as shorthand notations of $A_{\pi_{\theta_1}}$, $A_{\pi_{\theta_2}}$, $b_{\pi_{\theta_1}}$ and $b_{\pi_{\theta_2}}$ respectively. By Assumption 2, $A_{\theta,\phi}$ is invertible for any $\theta \in \mathbb{R}^d$, so we can write $\omega_\theta^* = -A_{\theta,\phi}^{-1}b_{\theta,\phi}$. Then we have

$$
\begin{aligned}
\|\omega_1^* - \omega_2^*\|_2 &= \| - A_1^{-1}b_1 + A_2^{-1}b_2 \|_2 \\
&= \| - A_1^{-1}b_1 - A_1^{-1}b_2 + A_1^{-1}b_2 + A_2^{-1}b_2 \|_2 \\
&= \| - A_1^{-1}(b_1 - b_2) - (A_1^{-1} - A_2^{-1})b_2 \|_2 \\
&\leq \| A_1^{-1}(b_1 - b_2) \|_2 + \| (A_1^{-1} - A_2^{-1})b_2 \|_2 \\
&\leq \| A_1^{-1} \|_2 \| b_1 - b_2 \|_2 + \| A_1^{-1} - A_2^{-1} \|_2 \| b_2 \|_2 \\
&= \| A_1^{-1} \|_2 \| b_1 - b_2 \|_2 + \| A_1^{-1}(A_2 - A_1)A_2^{-1} \|_2 \| b_2 \|_2 \\
&\leq \| A_1^{-1} \|_2 \| b_1 - b_2 \|_2 + \| A_1^{-1} \|_2 \| A_2^{-1} \|_2 \| b_2 \|_2 \| (A_2 - A_1) \|_2 \\
&\leq \lambda^{-1} \| b_1 - b_2 \|_2 + \lambda^{-2} r_{\max} \| A_1 - A_2 \|_2, \qquad (29)
\end{aligned}
$$

where the last inequality follows Assumption 2, and the fact that

$$
\| b_2 \|_2 = \| \mathbb{E}[r(s,a,s')\phi(s)] \|_2 \leq \mathbb{E}\| r(s,a,s')\phi(s) \|_2 \leq \mathbb{E}[|r(s,a,s')|\|\phi(s)\|_2] \leq r_{\max}.
$$

Denote $(s^1, a^1, s'^1)$ and $(s^2, a^2, s'^2)$ as samples drawn with $\theta_1$ and $\theta_2$ respectively, i.e. $s^1 \sim \mu_{\theta_1}$, $a^1 \sim \pi_{\theta_1}$, $s'^1 \sim \widetilde{\mathcal{P}}$ and $s^2 \sim \mu_{\theta_2}$, $a^2 \sim \pi_{\theta_2}$, $s'^2 \sim \widetilde{\mathcal{P}}$. Then we have

$$
\begin{aligned}
\| b_1 - b_2 \|_2 &= \left\| \mathbb{E}\left[ r(s^1, a^1, s'^1)\phi(s^1) \right] - \mathbb{E}\left[ r(s^2, a^2, s'^2)\phi(s^2) \right] \right\|_2 \\
&\leq \sup_{s,a,s'} \| r(s,a,s')\phi(s) \|_2 \| \mathbb{P}((s^1,a^1,s'^1) \in \cdot) - \mathbb{P}((s^2,a^2,s'^2) \in \cdot) \|_{TV} \\
&\leq r_{\max} \| \mathbb{P}((s^1,a^1,s'^1) \in \cdot) - \mathbb{P}((s^2,a^2,s'^2) \in \cdot) \|_{TV} \\
&= 2r_{\max}d_{TV}\left( \mu_{\theta_1} \otimes \pi_{\theta_1} \otimes \widetilde{\mathcal{P}}, \mu_{\theta_2} \otimes \pi_{\theta_2} \otimes \widetilde{\mathcal{P}} \right) \\
&\leq 2r_{\max}|\mathcal{A}|L_\pi(1 + \log_\rho \kappa^{-1} + (1-\rho)^{-1})\|\theta_1 - \theta_2\|_2, \qquad (30)
\end{aligned}
$$

where the first inequality follows the definition of total variation (TV) norm, and the last inequality follows Lemma A.1. in [17]. Similarly we have:

$$\|A_1 - A_2\|_2 \leq 2(1 + \gamma)d_{TV}\left(\mu_{\theta_1} \otimes \pi_{\theta_1}, \mu_{\theta_2} \otimes \pi_{\theta_2}\right)$$
$$= (1 + \gamma)|\mathcal{A}|L_\pi(1 + \log_\rho \kappa^{-1} + (1 - \rho)^{-1})\|\theta_1 - \theta_2\|_2. \tag{31}$$

Substituting (30) and (31) into (29) completes the proof. $\qquad \square$

## B  PROOF OF MAIN THEOREMS

### B.1  PROOF OF THEOREM 1

For brevity, we first define the following notations:

$$x := (s, a, s'),$$
$$\hat{\delta}(x, \omega) := r(s, a, s') + \gamma\phi(s')^\top \omega - \phi(s)^\top \omega,$$
$$g(x, \omega) := \hat{\delta}(x, \omega)\phi(s),$$
$$\overline{g}(\theta, \omega) := \mathop{\mathbb{E}}_{s \sim \mu_\theta, a \sim \pi_\theta, s' \sim \widetilde{\mathcal{P}}} [g(x, \omega)].$$

We also define constant $C_\delta := r_{\max} + (1 + \gamma)\max\{\frac{r_{\max}}{1-\gamma}, R_\omega\}$, and we immediately have

$$\|g(x, \omega)\|_2 \leq |r(x) + \gamma\phi(s')^\top \omega - \phi(s)^\top \omega| \leq r_{\max} + (1 + \gamma)R_\omega \leq C_\delta \tag{32}$$

and likewise, we have $\|\overline{g}(x, \omega)\|_2 \leq C_\delta$.

The critic update in Algorithm 1 can be written compactly as:

$$\omega_{k+1} = \Pi_{R_\omega}\left(\omega_k + \beta_k g(x_{(k)}, \omega_{k-\tau_k})\right), \tag{33}$$

where $\tau_k$ is the delay of the parameters used in evaluating the $k$th stochastic gradient, and $x_{(k)} := (s_{(k)}, a_{(k)}, s'_{(k)})$ is the sample used to evaluate the stochastic gradient at $k$th update.

*Proof.* Using $\omega_k^*$ as shorthand notation of $\omega_{\theta_k}^*$, we start with the optimality gap

$$\|\omega_{k+1} - \omega_{k+1}^*\|_2^2$$
$$= \|\Pi_{R_\omega}\left(\omega_k + \beta_k g(x_{(k)}, \omega_{k-\tau_k})\right) - \omega_{k+1}^*\|_2^2$$
$$\leq \|\omega_k + \beta_k g(x_{(k)}, \omega_{k-\tau_k}) - \omega_{k+1}^*\|_2^2$$
$$= \|\omega_k - \omega_k^*\|_2^2 + 2\beta_k \left\langle \omega_k - \omega_k^*, g(x_{(k)}, \omega_{k-\tau_k})\right\rangle + 2\left\langle \omega_k - \omega_k^*, \omega_k^* - \omega_{k+1}^*\right\rangle + \left\|\omega_k^* - \omega_{k+1}^* + \beta_k g(x_{(k)}, \omega_{k-\tau_k})\right\|_2^2$$
$$= \|\omega_k - \omega_k^*\|_2^2 + 2\beta_k \left\langle \omega_k - \omega_k^*, g(x_{(k)}, \omega_{k-\tau_k}) - g(x_{(k)}, \omega_k)\right\rangle + 2\beta_k \left\langle \omega_k - \omega_k^*, g(x_{(k)}, \omega_k) - \overline{g}(\theta_k, \omega_k)\right\rangle$$
$$\quad + 2\beta_k \left\langle \omega_k - \omega_k^*, \overline{g}(\theta_k, \omega_k)\right\rangle + 2\left\langle \omega_k - \omega_k^*, \omega_k^* - \omega_{k+1}^*\right\rangle + \left\|\omega_k^* - \omega_{k+1}^* + \beta_k g(x_{(k)}, \omega_{k-\tau_k})\right\|_2^2$$
$$\leq \|\omega_k - \omega_k^*\|_2^2 + 2\beta_k \left\langle \omega_k - \omega_k^*, g(x_{(k)}, \omega_{k-\tau_k}) - g(x_{(k)}, \omega_k)\right\rangle + 2\beta_k \left\langle \omega_k - \omega_k^*, g(x_{(k)}, \omega_k) - \overline{g}(\theta_k, \omega_k)\right\rangle$$
$$\quad + 2\beta_k \left\langle \omega_k - \omega_k^*, \overline{g}(\theta_k, \omega_k)\right\rangle + 2\left\langle \omega_k - \omega_k^*, \omega_k^* - \omega_{k+1}^*\right\rangle + 2\left\|\omega_k^* - \omega_{k+1}^*\right\|_2^2 + 2C_\delta^2 \beta_k^2. \tag{34}$$

We first bound $\left\langle \omega_k - \omega_k^*, \overline{g}(\theta_k, \omega_k)\right\rangle$ in (34) as

$$\left\langle \omega_k - \omega_k^*, \overline{g}(\theta_k, \omega_k)\right\rangle = \left\langle \omega_k - \omega_k^*, \overline{g}(\theta_k, \omega_k) - \overline{g}(\theta_k, \omega_k^*)\right\rangle$$
$$= \left\langle \omega_k - \omega_k^*, \mathbb{E}\left[(\gamma\phi(s') - \phi(s))^\top (\omega_k - \omega_k^*)\phi(s)\right]\right\rangle$$
$$= \left\langle \omega_k - \omega_k^*, \mathbb{E}\left[\phi(s)(\gamma\phi(s') - \phi(s))^\top\right](\omega_k - \omega_k^*)\right\rangle$$
$$= \left\langle \omega_k - \omega_k^*, A_{\pi_{\theta_k}}(\omega_k - \omega_k^*)\right\rangle$$
$$\leq -\lambda\|\omega_k - \omega_k^*\|_2^2, \tag{35}$$

where the first equality is due to $\overline{g}(\theta, \omega_\theta^*) = A_{\theta,\phi}\omega_\theta^* + b = 0$, and the last inequality follows Assumption 2. Substituting (35) into (34), then taking expectation on both sides of (34) yield

$$\mathbb{E}\|\omega_{k+1} - \omega_{k+1}^*\|_2^2 \leq (1 - 2\lambda\beta_k)\mathbb{E}\|\omega_k - \omega_k^*\|_2^2 + 2\beta_k \mathbb{E}\left\langle \omega_k - \omega_k^*, g(x_{(k)}, \omega_{k-\tau_k}) - g(x_{(k)}, \omega_k)\right\rangle$$
$$\quad + 2\beta_k \mathbb{E}\left\langle \omega_k - \omega_k^*, g(x_{(k)}, \omega_k) - \overline{g}(\theta_k, \omega_k)\right\rangle + 2\mathbb{E}\left\langle \omega_k - \omega_k^*, \omega_k^* - \omega_{k+1}^*\right\rangle$$
$$\quad + 2\mathbb{E}\left\|\omega_k^* - \omega_{k+1}^*\right\|_2^2 + 2C_\delta^2 \beta_k^2. \tag{36}$$

We then bound the term $\mathbb{E}\left\langle \omega_k - \omega_k^*, g(x_{(k)}, \omega_{k-\tau_k}) - g(x_{(k)}, \omega_k) \right\rangle$ in (36) as

$$
\begin{aligned}
\mathbb{E}\left\langle \omega_k - \omega_k^*, g(x_{(k)}, \omega_{k-\tau_k}) - g(x_{(k)}, \omega_k) \right\rangle &= \mathbb{E}\left\langle \omega_k - \omega_k^*, \left(\gamma\phi(s'_{(k)}) - \phi(s_{(k)})\right)^\top (\omega_{k-\tau_k} - \omega_k)\phi(s_{(k)}) \right\rangle \\
&\leq (1+\gamma)\,\mathbb{E}\left[\|\omega_k - \omega_k^*\|_2 \|\omega_{k-\tau_k} - \omega_k\|_2\right] \\
&\leq (1+\gamma)\,\mathbb{E}\left[\|\omega_k - \omega_k^*\|_2 \left\|\sum_{i=k-\tau_k}^{k-1} (\omega_{i+1} - \omega_i)\right\|_2\right] \\
&\leq (1+\gamma)\,\mathbb{E}\left[\|\omega_k - \omega_k^*\|_2 \sum_{i=k-\tau_k}^{k-1} \beta_i \|g(x_i, \omega_{i-\tau_i})\|_2\right] \\
&\leq (1+\gamma)\,\mathbb{E}\left[\|\omega_k - \omega_k^*\|_2 \sum_{i=k-\tau_k}^{k-1} \beta_{k-K_0} \|g(x_i, \omega_{i-\tau_i})\|_2\right] \\
&\leq C_\delta(1+\gamma)K_0\beta_{k-K_0}\,\mathbb{E}\,\|\omega_k - \omega_k^*\|_2, \qquad (37)
\end{aligned}
$$

where the second last inequality is due to the monotonicity of step size, and the last inequality follows the definition of $C_\delta$ in (32).

Next we jointly bound the fourth and fifth term in (36) as

$$
\begin{aligned}
&2\,\mathbb{E}\left\langle \omega_k - \omega_k^*, \omega_k^* - \omega_{k+1}^* \right\rangle + 2\,\mathbb{E}\left\|\omega_k^* - \omega_{k+1}^*\right\|_2^2 \\
&\leq 2\,\mathbb{E}\left[\|\omega_k - \omega_k^*\|_2 \left\|\omega_k^* - \omega_{k+1}^*\right\|_2\right] + 2\,\mathbb{E}\left\|\omega_k^* - \omega_{k+1}^*\right\|_2^2 \\
&\leq 2L_\omega\,\mathbb{E}\left[\|\omega_k - \omega_k^*\|_2 \|\theta_k - \theta_{k+1}\|_2\right] + 2L_\omega^2\,\mathbb{E}\|\theta_k - \theta_{k+1}\|_2^2 \\
&= 2L_\omega\alpha_k\,\mathbb{E}\left[\|\omega_k - \omega_k^*\|_2 \left\|\hat{\delta}(x_{(k)}, \omega_{k-\tau_k})\psi_{\theta_{k-\tau_k}}(s_{(k)}, a_{(k)})\right\|_2\right] + 2L_\omega^2\alpha_k^2\,\mathbb{E}\left\|\hat{\delta}(x_{(k)}, \omega_{k-\tau_k})\psi_{\theta_{k-\tau_k}}(s_{(k)}, a_{(k)})\right\|_2^2 \\
&\leq 2L_\omega C_p\alpha_k\,\mathbb{E}\,\|\omega_k - \omega_k^*\|_2 + 2L_\omega^2 C_p^2\alpha_k^2, \qquad (38)
\end{aligned}
$$

where constant $C_p := C_\delta C_\psi$. The second inequality is due to the $L_\omega$-Lipschitz of $\omega_\theta^*$ shown in Proposition 2, and the last inequality follows the fact that

$$
\|\hat{\delta}(x_{(k)}, \omega_{k-\tau_k})\psi_{\theta_{k-\tau_k}}(s_{(k)}, a_{(k)})\|_2 \leq C_\delta C_\psi = C_p. \qquad (39)
$$

Substituting (37) and (38) into (36) yields

$$
\begin{aligned}
\mathbb{E}\,\|\omega_{k+1} - \omega_{k+1}^*\|_2^2 &\leq (1 - 2\lambda\beta_k)\,\mathbb{E}\,\|\omega_k - \omega_k^*\|_2^2 + 2\beta_k(C_1\frac{\alpha_k}{\beta_k} + C_2 K_0\beta_{k-K_0})\,\mathbb{E}\,\|\omega_k - \omega_k^*\|_2 \\
&\quad + 2\beta_k\,\mathbb{E}\left\langle \omega_k - \omega_k^*, g(x_{(k)}, \omega_k) - \overline{g}(\theta_k, \omega_k) \right\rangle + C_q\beta_k^2, \qquad (40)
\end{aligned}
$$

where $C_1 := L_\omega C_p$, $C_2 := C_\delta(1+\gamma)$ and $C_q := 2C_\delta^2 + 2L_\omega^2 C_p^2 \max_{(k)} \frac{\alpha_k^2}{\beta_k^2} = 2C_\delta^2 + 2L_\omega^2 C_p^2 \frac{c_1^2}{c_2^2}$.

For brevity, we use $x \sim \theta$ to denote $s \sim \mu_\theta$, $a \sim \pi_\theta$ and $s' \sim \widetilde{\mathcal{P}}$ in this proof. Consider the third term in (40) conditioned on $\theta_k, \omega_k, \theta_{k-\tau_k}$. We bound it as

$$
\begin{aligned}
&\mathbb{E}\left[\left\langle \omega_k - \omega_k^*, g(x_{(k)}, \omega_k) - \overline{g}(\theta_k, \omega_k) \right\rangle | \theta_k, \omega_k, \theta_{k-\tau_k}\right] \\
&= \left\langle \omega_k - \omega_k^*, \mathop{\mathbb{E}}_{x_{(k)} \sim \theta_{k-\tau_k}}\left[g(x_{(k)}, \omega_k)|\omega_k\right] - \overline{g}(\theta_k, \omega_k) \right\rangle \\
&= \left\langle \omega_k - \omega_k^*, \overline{g}(\theta_{k-\tau_k}, \omega_k) - \overline{g}(\theta_k, \omega_k) \right\rangle \\
&\leq \|\omega_k - \omega_k^*\|_2 \|\overline{g}(\theta_{k-\tau_k}, \omega_k) - \overline{g}(\theta_k, \omega_k)\|_2 \\
&\leq 2R_\omega \left\|\mathop{\mathbb{E}}_{x \sim \theta_{k-\tau_k}}[g(x, \omega_k)] - \mathop{\mathbb{E}}_{x \sim \theta_k}[g(x, \omega_k)]\right\|_2 \\
&\leq 2R_\omega \sup_x \|g(x, \omega_k)\|_2 \left\|\mu_{\theta_{k-\tau_k}} \otimes \pi_{\theta_{k-\tau_k}} \otimes \widetilde{\mathcal{P}} - \mu_{\theta_k} \otimes \pi_{\theta_k} \otimes \widetilde{\mathcal{P}}\right\|_{TV} \\
&\leq 4R_\omega C_\delta d_{TV}(\mu_{\theta_{k-\tau_k}} \otimes \pi_{\theta_{k-\tau_k}} \otimes \widetilde{\mathcal{P}}, \mu_{\theta_k} \otimes \pi_{\theta_k} \otimes \widetilde{\mathcal{P}}), \qquad (41)
\end{aligned}
$$

where second last inequality follows the definition of TV norm and the last inequality uses the definition of $C_\delta$ in (32).

Define constant $C_3 := 2R_\omega C_\delta |\mathcal{A}| L_\pi (1 + \log_\rho \kappa^{-1} + (1 - \rho)^{-1})$. Then by following the third item in Lemma A.1 shown by [17], we can write (41) as

$$
\begin{aligned}
&\mathbb{E}\left[\langle \omega_k - \omega_k^*, g(x_{(k)}, \omega_k) - \overline{g}(\theta_k, \omega_k)\rangle \,|\, \theta_k, \omega_k, \theta_{k-\tau_k}\right] \\
&\leq 4R_\omega C_\delta d_{TV}(\mu_{\theta_{k-\tau_k}} \otimes \pi_{\theta_{k-\tau_k}} \otimes \widetilde{\mathcal{P}}, \mu_{\theta_k} \otimes \pi_{\theta_k} \otimes \widetilde{\mathcal{P}}) \\
&\leq C_3 \|\theta_{k-\tau_k} - \theta_k\|_2 \\
&\leq C_3 \sum_{i=k-\tau_k}^{k-1} \alpha_i \|g(x_i, \omega_{i-\tau_i})\|_2 \\
&\leq C_3 C_\delta K_0 \alpha_{k-K_0},
\end{aligned}
\tag{42}
$$

where we used the monotonicity of $\alpha_k$ and Assumption 1.

Taking total expectation on both sides of (42) and substituting it into (40) yield

$$
\begin{aligned}
\mathbb{E}\|\omega_{k+1} - \omega_{k+1}^*\|_2^2 &\leq (1 - 2\lambda\beta_k)\,\mathbb{E}\|\omega_k - \omega_k^*\|_2^2 + 2\beta_k\left(C_1 \frac{\alpha_k}{\beta_k} + C_2 K_0 \beta_{k-K_0}\right)\mathbb{E}\|\omega_k - \omega_k^*\|_2 \\
&\quad + 2C_3 C_\delta K_0 \beta_k \alpha_{k-K_0} + C_q \beta_k^2.
\end{aligned}
\tag{43}
$$

Taking summation on both sides of (43) and rearranging yield

$$
2\lambda \sum_{k=K_0}^{K} \mathbb{E}\|\omega_k - \omega_k^*\|_2^2 \leq \underbrace{\sum_{k=K_0}^{K} \frac{1}{\beta_k}\left(\mathbb{E}\|\omega_k - \omega_k^*\|_2^2 - \mathbb{E}\|\omega_{k+1} - \omega_{k+1}^*\|_2^2\right)}_{I_1} + C_q \underbrace{\sum_{k=K_0}^{K} \beta_k}_{I_2}
$$

$$
+ 2\underbrace{\sum_{k=K_0}^{K} 2C_3 C_\delta K_0 \alpha_{k-K_0}}_{I_3} + 2\underbrace{\sum_{k=K_0}^{K}\left(C_1 \frac{\alpha_k}{\beta_k} + C_2 K_0 \beta_{k-K_0}\right)\mathbb{E}\|\omega_k - \omega_k^*\|_2}_{I_4}.
\tag{44}
$$

We bound $I_1$ as

$$
\begin{aligned}
I_1 &= \sum_{k=M_K}^{K} \frac{1}{\beta_k}\left(\mathbb{E}\|\omega_k - \omega_k^*\|_2^2 - \mathbb{E}\|\omega_{k+1} - \omega_{k+1}^*\|_2^2\right) \\
&= \sum_{k=M_K}^{K}\left(\frac{1}{\beta_k} - \frac{1}{\beta_{k-1}}\right)\mathbb{E}\|\omega_k - \omega_k^*\|_2^2 + \frac{1}{\beta_{M_K-1}}\mathbb{E}\|\omega_{M_K} - \omega_{M_K}^*\|_2^2 - \frac{1}{\beta_k}\mathbb{E}\|\omega_{K+1} - \omega_{K+1}^*\|_2^2 \\
&\leq \sum_{k=M_K}^{K}\left(\frac{1}{\beta_k} - \frac{1}{\beta_{k-1}}\right)\mathbb{E}\|\omega_k - \omega_k^*\|_2^2 + \frac{1}{\beta_{M_K-1}}\mathbb{E}\|\omega_{M_K} - \omega_{M_K}^*\|_2^2 \\
&\leq 4R_\omega^2 \left(\sum_{k=M_K}^{K}\left(\frac{1}{\beta_k} - \frac{1}{\beta_{k-1}}\right) + \frac{1}{\beta_{M_K-1}}\right) = \frac{4R_\omega^2}{\beta_k} = \mathcal{O}(K^{\sigma_2}),
\end{aligned}
\tag{45}
$$

where the last inequality is due to the fact that

$$
\|\omega_k - \omega_\theta^*\|_2 \leq \|\omega_k\|_2 + \|\omega_\theta^*\|_2 \leq 2R_\omega.
$$

We bound $I_2$ as

$$
\sum_{k=M_K}^{K} \beta_k = \sum_{k=M_K}^{K} \frac{c_2}{(1+k)^{\sigma_2}} = \mathcal{O}(K^{1-\sigma_2})
\tag{46}
$$

where the inequality follows from the integration rule $\sum_{k=a}^{b} k^{-\sigma} \leq \frac{b^{1-\sigma}}{1-\sigma}$.

We bound $I_3$ as

$$I_3 = \sum_{k=K_0}^{K} 2C_3 C_\delta K_0 \alpha_{k-K_0} = 2C_3 C_\delta c_1 K_0 \sum_{k=0}^{K-K_0} (1+k)^{-\sigma_1} = \mathcal{O}(K_0 K^{1-\sigma_1}). \quad (47)$$

For the last term $I_4$, we have

$$
\begin{aligned}
I_4 &= \sum_{k=K_0}^{K} \left( C_1 \frac{\alpha_k}{\beta_k} + C_2 K_0 \beta_{k-K_0} \right) \mathbb{E} \left\| \omega_k - \omega_k^* \right\|_2 \\
&\leq \sqrt{ \sum_{k=K_0}^{K} \left( C_1 \frac{\alpha_k}{\beta_k} + C_2 K_0 \beta_{k-K_0} \right)^2 } \sqrt{ \sum_{k=K_0}^{K} \left( \mathbb{E} \left\| \omega_k - \omega_k^* \right\|_2 \right)^2 } \\
&\leq \sqrt{ \sum_{k=K_0}^{K} \left( C_1 \frac{\alpha_k}{\beta_k} + C_2 K_0 \beta_{k-K_0} \right)^2 } \sqrt{ \sum_{k=K_0}^{K} \mathbb{E} \left\| \omega_k - \omega_k^* \right\|_2^2 }, \quad (48)
\end{aligned}
$$

where the first inequality follows Cauchy–Schwartz inequality, and the second inequality follows Jensen's inequality. In (48), we have

$$
\begin{aligned}
\sum_{k=K_0}^{K} \left( C_1 \frac{\alpha_k}{\beta_k} + C_2 K_0 \beta_{k-K_0} \right)^2 &\leq \sum_{k=0}^{K-K_0} \left( C_1 \frac{\alpha_k}{\beta_k} + C_2 K_0 \beta_k \right)^2 \\
&= C_1^2 \sum_{k=0}^{K-K_0} \frac{\alpha_k^2}{\beta_k^2} + 2C_1 C_2 K_0 \sum_{k=0}^{K-K_0} \alpha_k + C_2^2 K_0^2 \sum_{k=0}^{K-K_0} \beta_k^2 \\
&= \mathcal{O}\left( K^{2(\sigma_2-\sigma_1)+1} \right) + \mathcal{O}\left( K_0 K^{-\sigma_1+1} \right) + \mathcal{O}\left( K_0^2 K^{1-2\sigma_2} \right) \quad (49)
\end{aligned}
$$

where the first inequality is due to the fact that $\frac{\alpha_k}{\beta_k}$ and $\beta_{k-K_0}$ are monotonically decreasing.

Substituting (49) into (48) gives

$$I_4 \leq \sqrt{ \mathcal{O}\left( K^{2(\sigma_2-\sigma_1)+1} \right) + \mathcal{O}\left( K_0 K^{-\sigma_1+1} \right) + \mathcal{O}\left( K_0^2 K^{1-2\sigma_2} \right) } \sqrt{ \sum_{k=M_K}^{K} \mathbb{E} \left\| \omega_k - \omega_k^* \right\|_2^2 }. \quad (50)$$

Substituting (45), (46), (47) and (50) into (44), and dividing both sides of (44) by $K - K_0 + 1$ give

$$
\begin{aligned}
&2\lambda \frac{1}{K - K_0 + 1} \sum_{k=K_0}^{K} \mathbb{E} \left\| \omega_k - \omega_k^* \right\|_2^2 \\
&\leq \frac{ \sqrt{ \mathcal{O}\left( K^{2(\sigma_2-\sigma_1)+1} \right) + \mathcal{O}\left( K_0 K^{-\sigma_1+1} \right) + \mathcal{O}\left( K_0^2 K^{1-2\sigma_2} \right) } }{ K - K_0 + 1 } \sqrt{ \sum_{k=K_0}^{K} \mathbb{E} \left\| \omega_k - \omega_k^* \right\|_2^2 } \\
&\quad + \mathcal{O}\left( \frac{1}{K^{1-\sigma_2}} \right) + \mathcal{O}\left( \frac{1}{K^{\sigma_2}} \right) + \mathcal{O}\left( \frac{K_0}{K^{\sigma_1}} \right). \quad (51)
\end{aligned}
$$

We define the following functions:

$$
\begin{aligned}
T_1(K) &:= \frac{1}{K - K_0 + 1} \sum_{k=K_0}^{K} \mathbb{E} \left\| \omega_k - \omega_k^* \right\|_2^2, \\
T_2(K) &:= \mathcal{O}\left( \frac{1}{K^{1-\sigma_2}} \right) + \mathcal{O}\left( \frac{1}{K^{\sigma_2}} \right) + \mathcal{O}\left( \frac{K_0}{K^{\sigma_1}} \right), \\
T_3(K) &:= \frac{ \mathcal{O}\left( K^{2(\sigma_2-\sigma_1)+1} \right) + \mathcal{O}\left( K_0 K^{-\sigma_1+1} \right) + \mathcal{O}\left( K_0^2 K^{1-2\sigma_2} \right) }{ K - K_0 + 1 }.
\end{aligned}
$$

Then (51) can be written as:

$$T_1(K) - \frac{1}{2\lambda}\sqrt{T_1(K)}\sqrt{T_3(K)} \le \frac{1}{2\lambda}T_2(K).$$

Solving this quadratic inequality in terms of $T_1(K)$, we obtain

$$T_1(K) \le \frac{1}{\lambda}T_2(K) + \frac{1}{2\lambda^2}T_3(K), \tag{52}$$

which implies

$$\frac{1}{K - K_0 + 1}\sum_{k=K_0}^{K}\mathbb{E}\,\|\omega_k - \omega_k^*\|_2^2$$

$$= \mathcal{O}\left(\frac{1}{K^{1-\sigma_2}}\right) + \mathcal{O}\left(\frac{1}{K^{2(\sigma_1-\sigma_2)}}\right) + \mathcal{O}\left(\frac{K_0^2}{K^{2\sigma_2}}\right) + \mathcal{O}\left(\frac{K_0}{K^{\sigma_1}}\right) + \mathcal{O}\left(\frac{1}{K^{\sigma_2}}\right).$$

We further have

$$\frac{1}{K}\sum_{k=1}^{K}\mathbb{E}\,\|\omega_k - \omega_k^*\|_2^2 \le \frac{1}{K}\left(\sum_{k=1}^{K_0-1}4R_\omega^2 + \sum_{k=K_0}^{K}\mathbb{E}\,\|\omega_k - \omega_k^*\|_2^2\right)$$

$$= \frac{K_0 - 1}{K}4R_\omega^2 + \frac{K - K_0 + 1}{K}\frac{1}{K - K_0 + 1}\sum_{k=K_0}^{K}\mathbb{E}\,\|\omega_k - \omega_k^*\|_2^2$$

$$= \mathcal{O}\left(\frac{K_0}{K}\right) + \mathcal{O}\left(\frac{1}{K - K_0 + 1}\sum_{k=K_0}^{K}\mathbb{E}\,\|\omega_k - \omega_k^*\|_2^2\right)$$

$$= \mathcal{O}\left(\frac{1}{K - K_0 + 1}\sum_{k=K_0}^{K}\mathbb{E}\,\|\omega_k - \omega_k^*\|_2^2\right) \tag{53}$$

which completes the proof. $\qquad\square$

### B.2 PROOF OF THEOREM 2

We first clarify the notations:

$$x := (s, a, s'),$$
$$\hat{\delta}(x, \omega) := r(s, a, s') + \gamma\phi(s')^\top\omega - \phi(s)^\top\omega,$$
$$\delta(x, \theta) := r(s, a, s') + \gamma V_{\pi_\theta}(s') - V_{\pi_\theta}(s).$$

The update in Algorithm 1 can be written compactly as:

$$\theta_{k+1} = \theta_k + \alpha_k\hat{\delta}(x_{(k)}, \omega_{k-\tau_k})\psi_{\theta_{k-\tau_k}}(s_{(k)}, a_{(k)}). \tag{54}$$

For brevity, we use $\omega_k^*$ as shorthand notation of $\omega_{\theta_k}^*$. Then we are ready to give the proof.

*Proof.* From $L_J$-Lipschitz of policy gradient shown in Proposition 1, we have:

$$J(\theta_{k+1}) \ge J(\theta_k) + \langle\nabla J(\theta_k), \theta_{k+1} - \theta_k\rangle - \frac{L_J}{2}\|\theta_{k+1} - \theta_k\|_2^2$$

$$= J(\theta_k) + \alpha_k\left\langle\nabla J(\theta_k), \left(\hat{\delta}(x_{(k)}, \omega_{k-\tau_k}) - \hat{\delta}(x_{(k)}, \omega_k^*)\right)\psi_{\theta_{k-\tau_k}}(s_{(k)}, a_{(k)})\right\rangle$$

$$+ \alpha_k\left\langle\nabla J(\theta_k), \hat{\delta}(x_{(k)}, \omega_k^*)\psi_{\theta_{k-\tau_k}}(s_{(k)}, a_{(k)})\right\rangle - \frac{L_J}{2}\alpha_k^2\|\hat{\delta}(x_{(k)}, \omega_{k-\tau_k})\psi_{\theta_{k-\tau_k}}(s_{(k)}, a_{(k)})\|_2^2$$

$$\ge J(\theta_k) + \alpha_k\left\langle\nabla J(\theta_k), \left(\hat{\delta}(x_{(k)}, \omega_{k-\tau_k}) - \hat{\delta}(x_{(k)}, \omega_k^*)\right)\psi_{\theta_{k-\tau_k}}(s_{(k)}, a_{(k)})\right\rangle$$

$$+ \alpha_k\left\langle\nabla J(\theta_k), \hat{\delta}(x_{(k)}, \omega_k^*)\psi_{\theta_{k-\tau_k}}(s_{(k)}, a_{(k)})\right\rangle - \frac{L_J}{2}C_p^2\alpha_k^2,$$

where the last inequality follows the definition of $C_p$ in (39).

Taking expectation on both sides of the last inequality yields

$$\mathbb{E}[J(\theta_{k+1})] \geq \mathbb{E}[J(\theta_k)] + \alpha_k \underbrace{\mathbb{E}\left\langle \nabla J(\theta_k), \left(\hat{\delta}(x_{(k)}, \omega_{k-\tau_k}) - \hat{\delta}(x_{(k)}, \omega_k^*)\right) \psi_{\theta_{k-\tau_k}}(s_{(k)}, a_{(k)})\right\rangle}_{I_1}$$

$$+ \alpha_k \underbrace{\mathbb{E}\left\langle \nabla J(\theta_k), \hat{\delta}(x_{(k)}, \omega_k^*) \psi_{\theta_{k-\tau_k}}(s_{(k)}, a_{(k)})\right\rangle}_{I_2} - \frac{L_J}{2} C_p^2 \alpha_k^2. \tag{55}$$

We first decompose $I_1$ as

$$I_1 = \mathbb{E}\left\langle \nabla J(\theta_k), \left(\hat{\delta}(x_{(k)}, \omega_{k-\tau_k}) - \hat{\delta}(x_{(k)}, \omega_k^*)\right) \psi_{\theta_{k-\tau_k}}(s_{(k)}, a_{(k)})\right\rangle$$

$$= \underbrace{\mathbb{E}\left\langle \nabla J(\theta_k), \left(\hat{\delta}(x_{(k)}, \omega_{k-\tau_k}) - \hat{\delta}(x_{(k)}, \omega_k)\right) \psi_{\theta_{k-\tau_k}}(s_{(k)}, a_{(k)})\right\rangle}_{I_1^{(1)}}$$

$$+ \underbrace{\mathbb{E}\left\langle \nabla J(\theta_k), \left(\hat{\delta}(x_{(k)}, \omega_k) - \hat{\delta}(x_{(k)}, \omega_k^*)\right) \psi_{\theta_{k-\tau_k}}(s_{(k)}, a_{(k)})\right\rangle}_{I_1^{(2)}}.$$

We bound $I_1^{(1)}$ as

$$I_1^{(1)} = \mathbb{E}\left\langle \nabla J(\theta_k), \left(\gamma\phi(s'_{(k)}) - \phi(s_{(k)})\right)^\top (\omega_{k-\tau_k} - \omega_k)\psi_{\theta_{k-\tau_k}}(s_{(k)}, a_{(k)})\right\rangle$$

$$\geq -\mathbb{E}\left[\|\nabla J(\theta_k)\|_2 \|\gamma\phi(s'_{(k)}) - \phi(s_{(k)})\|_2 \|\omega_k - \omega_{k-\tau_k}\|_2 \|\psi_{\theta_{k-\tau_k}}(s_{(k)}, a_{(k)})\|_2\right]$$

$$\geq -2C_\psi \, \mathbb{E}\left[\|\nabla J(\theta_k)\|_2 \|\omega_k - \omega_{k-\tau_k}\|_2\right]$$

$$\geq -2C_\psi C_\delta K_0 \beta_{k-1} \, \mathbb{E}\,\|\nabla J(\theta_k)\|_2,$$

where the last inequality follows

$$\|\omega_k - \omega_{k-\tau_k}\|_2 = \left\|\sum_{i=k-\tau_k}^{k-1} (\omega_{i+1} - \omega_i)\right\|_2$$

$$\leq \sum_{i=k-\tau_k}^{k-1} \|\beta_i g(x_i, \omega_{i-\tau_i})\|_2$$

$$\leq \beta_{k-1} \sum_{i=k-\tau_k}^{k-1} \|g(x_i, \omega_{i-\tau_i})\|_2$$

$$\leq \beta_{k-1} K_0 C_\delta,$$

where the second inequality is due to the monotonicity of step size, and the third one follows (32).

Then we bound $I_1^{(2)}$ as

$$I_1^{(2)} = \mathbb{E}\left\langle \nabla J(\theta_k), \left(\hat{\delta}(x_{(k)}, \omega_k) - \hat{\delta}(x_{(k)}, \omega_k^*)\right) \psi_{\theta_{k-\tau_k}}(s_{(k)}, a_{(k)})\right\rangle$$

$$= -\mathbb{E}\left\langle \nabla J(\theta_k), \left(\gamma\phi(s'_{(k)}) - \phi(s_{(k)})\right)^\top (\omega_k^* - \omega_k)\psi_{\theta_{k-\tau_k}}(s_{(k)}, a_{(k)})\right\rangle$$

$$\geq -\mathbb{E}\left[\|\nabla J(\theta_k)\|_2 \|\gamma\phi(s'_{(k)}) - \phi(s_{(k)})\|_2 \|\omega_k - \omega_k^*\|_2 \|\psi_{\theta_{k-\tau_k}}(s_{(k)}, a_{(k)})\|_2\right]$$

$$\geq -2C_\psi \, \mathbb{E}\left[\|\nabla J(\theta_k)\|_2 \|\omega_k - \omega_k^*\|_2\right].$$

Collecting the lower bounds of $I_1^{(1)}$ and $I_1^{(2)}$ gives

$$I_1 \geq -2C_\psi \, \mathbb{E}\left[\|\nabla J(\theta_k)\|_2 \left(C_\delta K_0 \beta_{k-1} + \|\omega_k - \omega_k^*\|_2\right)\right]. \tag{56}$$

Now we consider $I_2$. We first decompose $I_2$ as

$$
\begin{aligned}
I_2 &= \mathbb{E}\left\langle \nabla J(\theta_k), \hat{\delta}(x_{(k)}, \omega_k^*)\psi_{\theta_{k-\tau_k}}(s_{(k)}, a_{(k)})\right\rangle \\
&= \underbrace{\mathbb{E}\left\langle \nabla J(\theta_k), \left(\hat{\delta}(x_{(k)}, \omega_k^*) - \hat{\delta}(x_{(k)}, \omega_{k-\tau_k}^*)\right)\psi_{\theta_{k-\tau_k}}(s_{(k)}, a_{(k)})\right\rangle}_{I_2^{(1)}} \\
&\quad + \underbrace{\mathbb{E}\left\langle \nabla J(\theta_k), \left(\hat{\delta}(x_{(k)}, \omega_{k-\tau_k}^*) - \delta(x_{(k)}, \theta_{k-\tau_k})\right)\psi_{\theta_{k-\tau_k}}(s_{(k)}, a_{(k)})\right\rangle}_{I_2^{(2)}} \\
&\quad + \underbrace{\mathbb{E}\left\langle \nabla J(\theta_k), \delta(x_{(k)}, \theta_{k-\tau_k})\psi_{\theta_{k-\tau_k}}(s_{(k)}, a_{(k)}) - \nabla J(\theta_k)\right\rangle}_{I_2^{(3)}} + \|\nabla J(\theta_k)\|_2^2.
\end{aligned}
$$

We bound $I_2^{(1)}$ as

$$
\begin{aligned}
I_2^{(1)} &= \mathbb{E}\left\langle \nabla J(\theta_k), \left(\hat{\delta}(x_{(k)}, \omega_k^*) - \hat{\delta}(x_{(k)}, \omega_{k-\tau_k}^*)\right)\psi_{\theta_{k-\tau_k}}(s_{(k)}, a_{(k)})\right\rangle \\
&= \mathbb{E}\left\langle \nabla J(\theta_k), \left(\gamma\phi(s'_{(k)}) - \phi(s_{(k)})\right)^\top (\omega_k^* - \omega_{k-\tau_k}^*)\psi_{\theta_{k-\tau_k}}(s_{(k)}, a_{(k)})\right\rangle \\
&\geq -\mathbb{E}\left[\|\nabla J(\theta_k)\|_2\|\left(\gamma\phi(s'_{(k)}) - \phi(s_{(k)})\right)^\top\|_2 \left\|\omega_k^* - \omega_{k-\tau_k}^*\right\|_2 \|\psi_{\theta_{k-\tau_k}}(s_{(k)}, a_{(k)})\|_2\right] \\
&\geq -L_V C_\psi(1+\gamma)\mathbb{E}\left\|\omega_k^* - \omega_{k-\tau_k}^*\right\|_2 \\
&\geq -L_V L_\omega C_\psi(1+\gamma)\mathbb{E}\left\|\theta_k - \theta_{k-\tau_k}\right\|_2 \\
&\geq -L_V L_\omega C_\psi C_p(1+\gamma)K_0\alpha_{k-K_0},
\end{aligned}
$$

where the second last inequality follows from Proposition 2 and the last inequality uses (39) as

$$
\begin{aligned}
\|\theta_k - \theta_{k-\tau_k}\|_2 &\leq \sum_{i=k-\tau_k}^{k-1}\|\theta_{i+1} - \theta_i\|_2 \\
&= \sum_{i=k-\tau_k}^{k-1}\alpha_i\|\hat{\delta}(x_i, \omega_{i-\tau_i})\psi_{\theta_{i-\tau_i}}(s_i, a_i)\|_2 \\
&\leq \sum_{i=k-\tau_k}^{k-1}\alpha_{k-\tau_k}C_p \\
&\leq C_p K_0\alpha_{k-K_0}. \tag{57}
\end{aligned}
$$

We bound $I_2^{(2)}$ as

$$
\begin{aligned}
I_2^{(2)} &= \mathbb{E}\left\langle \nabla J(\theta_k), \left(\hat{\delta}(x_{(k)}, \omega_{k-\tau_k}^*) - \delta(x_{(k)}, \theta_{k-\tau_k})\right)\psi_{\theta_{k-\tau_k}}(s_{(k)}, a_{(k)})\right\rangle \\
&\geq -\mathbb{E}\left[\|\nabla J(\theta_k)\|_2 \left|\hat{\delta}(x_{(k)}, \omega_{k-\tau_k}^*) - \delta(x_{(k)}, \theta_{k-\tau_k})\right| \|\psi_{\theta_{k-\tau_k}}(s_{(k)}, a_{(k)})\|_2\right] \\
&\geq -C_\psi\mathbb{E}\left[\|\nabla J(\theta_k)\|_2 \left|\hat{\delta}(x_{(k)}, \omega_{k-\tau_k}^*) - \delta(x_{(k)}, \theta_{k-\tau_k})\right|\right] \\
&= -C_\psi\mathbb{E}\left[\|\nabla J(\theta_k)\|_2 \left|\gamma\left(\phi(s'_{(k)})^\top\omega_{k-\tau_k}^* - V_{\pi_{\theta_{k-\tau_k}}}(s'_{(k)})\right) + V_{\pi_{\theta_{k-\tau_k}}}(s_{(k)}) - \phi(s_{(k)})^\top\omega_{k-\tau_k}^*\right|\right] \\
&\geq -C_\psi\mathbb{E}\left[\|\nabla J(\theta_k)\|_2 \left(\gamma\left|\phi(s'_{(k)})^\top\omega_{k-\tau_k}^* - V_{\pi_{\theta_{k-\tau_k}}}(s'_{(k)})\right| + \left|V_{\pi_{\theta_{k-\tau_k}}}(s_{(k)}) - \phi(s_{(k)})^\top\omega_{k-\tau_k}^*\right|\right)\right] \\
&= -C_\psi\mathbb{E}\left[\|\nabla J(\theta_k)\|_2 \mathbb{E}\left[\gamma\left|\phi(s'_{(k)})^\top\omega_{k-\tau_k}^* - V_{\pi_{\theta_{k-\tau_k}}}(s'_{(k)})\right| + \left|V_{\pi_{\theta_{k-\tau_k}}}(s_{(k)}) - \phi(s_{(k)})^\top\omega_{k-\tau_k}^*\right| \Big| \theta_k, \theta_{k-\tau_k}\right]\right] \\
&\geq -2C_\psi\epsilon_{\text{app}}\mathbb{E}\|\nabla J(\theta_k)\|_2 \\
&\geq -2C_\psi L_V\epsilon_{\text{fa}} - 2C_\psi\epsilon_{\text{sp}}\mathbb{E}\|\nabla J(\theta_k)\|_2 \tag{58}
\end{aligned}
$$

where the second last inequality follows from the fact that

$$\mathbb{E}\left[\gamma\left|\phi(s'_{(k)})^\top\omega^*_{k-\tau_k} - V_{\pi_{\theta_{k-\tau_k}}}(s'_{(k)})\right| + \left|V_{\pi_{\theta_{k-\tau_k}}}(s_{(k)}) - \phi(s_{(k)})^\top\omega^*_{k-\tau_k}\right|\right]$$

$$\leq \gamma\sqrt{\mathbb{E}\left|\phi(s'_{(k)})^\top\omega^*_{k-\tau_k} - V_{\pi_{\theta_{k-\tau_k}}}(s'_{(k)})\right|^2} + \sqrt{\mathbb{E}\left|V_{\pi_{\theta_{k-\tau_k}}}(s_{(k)}) - \phi(s_{(k)})^\top\omega^*_{k-\tau_k}\right|^2}$$

$$\leq 2\epsilon_{\text{app}}.$$

Define artificial transition $\bar{x}_{(k)} := (s_{(k)}, a_{(k)}, \bar{s}'_{(k)} \sim \mathcal{P})$, then $I_2^{(3)}$ can be bounded as

$$I_2^{(3)} = \mathbb{E}\left\langle\nabla J(\theta_k), \delta(x_{(k)},\theta_{k-\tau_k})\psi_{\theta_{k-\tau_k}}(s_{(k)},a_{(k)}) - \nabla J(\theta_k)\right\rangle$$

$$= \mathbb{E}\left[\mathbb{E}\left[\left\langle\nabla J(\theta_k), \delta(x_{(k)},\theta_{k-\tau_k})\psi_{\theta_{k-\tau_k}}(s_{(k)},a_{(k)}) - \nabla J(\theta_k)\right\rangle\middle|\theta_{k-\tau_k},\theta_k\right]\right]$$

$$= \mathbb{E}\left\langle\nabla J(\theta_k), \mathbb{E}\left[\left(\delta(x_{(k)},\theta_{k-\tau_k}) - \delta(\bar{x}_{(k)},\theta_{k-\tau_k})\right)\psi_{\theta_{k-\tau_k}}(s_{(k)},a_{(k)})\middle|\theta_{k-\tau_k},\theta_k\right]\right\rangle$$

$$+ \mathbb{E}\left\langle\nabla J(\theta_k), \mathbb{E}\left[\delta(\bar{x}_{(k)},\theta_{k-\tau_k})\psi_{\theta_{k-\tau_k}}(s_{(k)},a_{(k)})\middle|\theta_{k-\tau_k},\theta_k\right] - \nabla J(\theta_k)\right\rangle$$

$$\geq -\mathbb{E}\left[\|\nabla J(\theta_k)\|_2\left\|\mathbb{E}\left[\left(\delta(x_{(k)},\theta_{k-\tau_k}) - \delta(\bar{x}_{(k)},\theta_{k-\tau_k})\right)\psi_{\theta_{k-\tau_k}}(s_{(k)},a_{(k)})\middle|\theta_{k-\tau_k},\theta_k\right]\right\|_2\right]$$

$$-\mathbb{E}\left[\|\nabla J(\theta_k)\|_2\left\|\mathbb{E}\left[\delta(\bar{x}_{(k)},\theta_{k-\tau_k})\psi_{\theta_{k-\tau_k}}(s_{(k)},a_{(k)})\middle|\theta_{k-\tau_k},\theta_k\right] - \nabla J(\theta_k)\right\|_2\right]. \quad (59)$$

The first term in the last inequality can be bounded as

$$\mathbb{E}\left[\left(\delta(x_{(k)},\theta_{k-\tau_k}) - \delta(\bar{x}_{(k)},\theta_{k-\tau_k})\right)\psi_{\theta_{k-\tau_k}}(s_{(k)},a_{(k)})\middle|\theta_{k-\tau_k},\theta_k\right]$$

$$= \mathbb{E}\left[\left(\delta(x_{(k)},\theta_{k-\tau_k}) - \delta(\bar{x}_{(k)},\theta_{k-\tau_k})\right)\psi_{\theta_{k-\tau_k}}(s_{(k)},a_{(k)})\middle|\theta_{k-\tau_k},\theta_k\right]$$

$$= \mathbb{E}\left[\left(r(x_{(k)}) + \gamma\,\mathbb{E}[r(s'_k,a',s'')] - \left(r(\bar{x}_{(k)}) + \gamma\,\mathbb{E}[r(\bar{s}'_k,a',s'')]\right)\right)\psi_{\theta_{k-\tau_k}}(s_{(k)},a_{(k)})\middle|\theta_{k-\tau_k},\theta_k\right]$$

$$\leq 2C_\psi r_{\max}\|\widetilde{\mathcal{P}} - \mathcal{P}\|_{TV}$$

$$\leq 8C_\psi r_{\max}(1-\gamma), \quad (60)$$

where the last inequality follows

$$\|\widetilde{\mathcal{P}} - \mathcal{P}\|_{TV} = 2\int_{s'\in\mathcal{S}}\left|\widetilde{\mathcal{P}}(s'|s,a) - \mathcal{P}(s'|s,a)\right| = 2(1-\gamma)\int_{s'\in\mathcal{S}}|\mathcal{P}(s'|s,a) - \eta(s')| \leq 4(1-\gamma). \quad (61)$$

The second term in (59) can be rewritten as

$$\mathbb{E}\left[\delta(\bar{x}_{(k)},\theta_{k-\tau_k})\psi_{\theta_{k-\tau_k}}(s_{(k)},a_{(k)})\middle|\theta_{k-\tau_k},\theta_k\right]$$

$$= \mathop{\mathbb{E}}_{\substack{s_{(k)}\sim\mu_{\theta_{k-\tau_k}}\\ a_{(k)}\sim\pi_{\theta_{k-\tau_k}}\\ \bar{s}'_{(k)}\sim\mathcal{P}}}\left[\left(r(\bar{x}_{(k)}) + \gamma V_{\pi_{\theta_{k-\tau_k}}}(\bar{s}'_{(k)}) - V_{\pi_{\theta_{k-\tau_k}}}(s_{(k)})\right)\psi_{\theta_{k-\tau_k}}(s_{(k)},a_{(k)})\middle|\theta_{k-\tau_k},\theta_k\right]$$

$$= \mathop{\mathbb{E}}_{\substack{s_{(k)}\sim\mu_{\theta_{k-\tau_k}}\\ a_{(k)}\sim\pi_{\theta_{k-\tau_k}}}}\left[\left(Q_{\pi_{\theta_{k-\tau_k}}}(s_{(k)},a_{(k)}) - V_{\pi_{\theta_{k-\tau_k}}}(s_{(k)})\right)\psi_{\theta_{k-\tau_k}}(s_{(k)},a_{(k)})\middle|\theta_{k-\tau_k},\theta_k\right]$$

$$= \mathop{\mathbb{E}}_{\substack{s_{(k)}\sim\mu_{\theta_{k-\tau_k}}\\ a_{(k)}\sim\pi_{\theta_{k-\tau_k}}}}\left[A_{\pi_{\theta_{k-\tau_k}}}(s_{(k)},a_{(k)})\psi_{\theta_{k-\tau_k}}(s_{(k)},a_{(k)})\middle|\theta_{k-\tau_k},\theta_k\right]$$

$$= \mathop{\mathbb{E}}_{\substack{s_{(k)}\sim d_{\theta_{k-\tau_k}}\\ a_{(k)}\sim\pi_{\theta_{k-\tau_k}}}}\left[A_{\pi_{\theta_{k-\tau_k}}}(s_{(k)},a_{(k)})\psi_{\theta_{k-\tau_k}}(s_{(k)},a_{(k)})\middle|\theta_{k-\tau_k},\theta_k\right] = \nabla J(\theta_{k-\tau_k}) \quad (62)$$

where the second last equality follows $\mu_\theta(\cdot) = d_\theta(\cdot)$ with $d_\theta$ being a shorthand notation of $d_{\pi_\theta}$ [6].

Substituting (60) and (62) into (59) yields

$$
\begin{aligned}
I_2^{(3)} &\geq -8C_\psi r_{\max}(1-\gamma)\,\mathbb{E}\,\|\nabla J(\theta_k)\|_2 - \mathbb{E}\left[\|\nabla J(\theta_k)\|_2\|\nabla J(\theta_{k-\tau_k}) - \nabla J(\theta_k)\|_2\right] \\
&\geq -8C_\psi r_{\max}(1-\gamma)\,\mathbb{E}\,\|\nabla J(\theta_k)\|_2 - L_V L_J\,\mathbb{E}\,\|\theta_{k-\tau_k} - \theta_k\|_2 \\
&\geq -8C_\psi r_{\max}(1-\gamma)\,\mathbb{E}\,\|\nabla J(\theta_k)\|_2 - L_V L_J C_p K_0 \alpha_{k-K_0},
\end{aligned}
\tag{63}
$$

where the second last inequality is due to $L_J$-Lipschitz of policy gradient shown in Proposition 1, and the last inequality follows (57).

Collecting lower bounds of $I_2^{(1)}$, $I_2^{(2)}$ and $I_2^{(3)}$ gives

$$
I_2 \geq -D_1 K_0 \alpha_{k-K_0} - (2C_\psi \epsilon_{\mathrm{sp}} + 8C_\psi r_{\max}(1-\gamma))\,\mathbb{E}\,\|\nabla J(\theta_k)\|_2 - 2C_\psi L_V \epsilon_{\mathrm{fa}} + \|\nabla J(\theta_k)\|_2^2,
\tag{64}
$$

where the constant is $D_1 := L_V L_\omega C_\psi C_p(1+\gamma) + L_V L_J C_p$.

Substituting (56) and (64) into (55) yields

$$
\begin{aligned}
\mathbb{E}[J(\theta_{k+1})] \geq{}& \mathbb{E}[J(\theta_k)] - 2\alpha_k C_\psi \left(\epsilon_{\mathrm{sp}} + 4r_{\max}(1-\gamma) + C_\delta K_0 \beta_{k-1} + \|\omega_k - \omega_k^*\|_2\right)\mathbb{E}\,\|\nabla J(\theta_k)\|_2 \\
&- \alpha_k D_1 K_0 \alpha_{k-K_0} - 2\alpha_k C_\psi L_V \epsilon_{\mathrm{fa}} + \alpha_k\|\nabla J(\theta_k)\|_2^2 - \frac{L_J}{2}C_p^2 \alpha_k^2.
\end{aligned}
\tag{65}
$$

By following Cauchy-Schwarz inequality, the second term in (65) can be bounded as

$$
\begin{aligned}
&\left(\epsilon_{\mathrm{sp}} + 4r_{\max}(1-\gamma) + C_\delta K_0 \beta_{k-1} + \|\omega_k - \omega_k^*\|_2\right)\mathbb{E}\,\|\nabla J(\theta_k)\|_2 \\
&\leq \sqrt{\mathbb{E}\,\|\nabla J(\theta_k)\|_2^2}\,\mathbb{E}\left[\left(\epsilon_{\mathrm{sp}} + 4r_{\max}(1-\gamma) + C_\delta K_0 \beta_{k-1} + \|\omega_k - \omega_k^*\|_2\right)^2\right] \\
&\leq \sqrt{\mathbb{E}\,\|\nabla J(\theta_k)\|_2^2}\sqrt{\mathbb{E}\left[4C_\delta^2 K_0^2 \beta_{k-1}^2 + 4\|\omega_k - \omega_k^*\|_2^2 + 4\epsilon_{\mathrm{sp}}^2 + 64r_{\max}^2(1-\gamma)^2\right]} \\
&= 2\sqrt{\mathbb{E}\,\|\nabla J(\theta_k)\|_2^2}\sqrt{C_\delta^2 K_0^2 \beta_{k-1}^2 + \mathbb{E}\,\|\omega_k - \omega_k^*\|_2^2 + \mathcal{O}(\epsilon_{\mathrm{sp}}^2)},
\end{aligned}
\tag{66}
$$

where the last inequality follows the order of $\epsilon_{\mathrm{sp}}$ in Lemma 7.

Collecting the upper bound gives

$$
\begin{aligned}
\mathbb{E}[J(\theta_{k+1})] \geq{}& \mathbb{E}[J(\theta_k)] - 4\alpha_k C_\psi \sqrt{\mathbb{E}\,\|\nabla J(\theta_k)\|_2^2}\sqrt{C_\delta^2 K_0^2 \beta_{k-1}^2 + \mathbb{E}\,\|\omega_k - \omega_k^*\|_2^2 + \mathcal{O}(\epsilon_{\mathrm{sp}}^2)} \\
&- \alpha_k D_1 K_0 \alpha_{k-K_0} - 2\alpha_k C_\psi L_V \epsilon_{\mathrm{fa}} + \alpha_k\|\nabla J(\theta_k)\|_2^2 - \frac{L_J}{2}C_p^2 \alpha_k^2.
\end{aligned}
\tag{67}
$$

Dividing both sides of (67) by $\alpha_k$, then rearranging and taking summation on both sides give

$$
\begin{aligned}
\sum_{k=K_0}^{K} \mathbb{E}\,\|\nabla J(\theta_k)\|_2^2 \leq{}& \underbrace{\sum_{k=K_0}^{K} \frac{1}{\alpha_k}\left(\mathbb{E}[J(\theta_{k+1})] - \mathbb{E}[J(\theta_k)]\right)}_{I_3} + \underbrace{\sum_{k=K_0}^{K}\left(D_1 K_0 \alpha_{k-K_0} + \frac{L_J}{2}C_p^2 \alpha_k\right)}_{I_4} \\
&+ 4C_\psi \underbrace{\sum_{k=K_0}^{K}\sqrt{\mathbb{E}\,\|\nabla J(\theta_k)\|_2^2}\sqrt{C_\delta^2 K_0^2 \beta_{k-1}^2 + \mathbb{E}\,\|\omega_k - \omega_k^*\|_2^2 + \mathcal{O}(\epsilon_{\mathrm{sp}}^2)}}_{I_5} \\
&+ 2C_\psi L_V(K - K_0 + 1)\epsilon_{\mathrm{fa}}.
\end{aligned}
\tag{68}
$$

We bound $I_3$ as

$$
\begin{aligned}
I_3 &= \sum_{k=K_0}^{K} \frac{1}{\alpha_k}\left(\mathbb{E}\left[J(\theta_{k+1})\right] - \mathbb{E}\left[J(\theta_k)\right]\right) \\
&= \sum_{k=K_0}^{K}\left(\frac{1}{\alpha_{k-1}} - \frac{1}{\alpha_k}\right)\mathbb{E}\left[J(\theta_k)\right] - \frac{1}{\alpha_{M_K-1}}\mathbb{E}\left[J(\theta_{M_K})\right] + \frac{1}{\alpha_K}\mathbb{E}\left[J(\theta_{K+1})\right] \\
&\leq \frac{1}{\alpha_K}\mathbb{E}\left[J(\theta_{K+1})\right] \\
&\leq \frac{r_{\max}}{1-\gamma}\frac{1}{\alpha_K} = \mathcal{O}(K^{\sigma_1}),
\end{aligned}
\tag{69}
$$

where the first inequality is due to the $\alpha_k$ is monotonic decreasing and positive, and last inequality is due to $V_{\pi_\theta}(s) \le \frac{r_{\max}}{1-\gamma}$ for any $s \in \mathcal{S}$ and $\pi_\theta$.

We bound $I_4$ as

$$I_4 = \sum_{k=K_0}^{K} \left( D_1 K_0 \alpha_{k-K_0} + \frac{L_J}{2} C_p^2 \alpha_k \right) \le \sum_{k=0}^{K-K_0} \left( D_1 K_0 \alpha_k + \frac{L_J}{2} C_p^2 \alpha_k \right) = \mathcal{O}(K_0 K^{1-\sigma_1}).$$

We bound $I_5$ as

$$
\begin{aligned}
I_5 &= \sum_{k=K_0}^{K} \sqrt{\mathbb{E}\|\nabla J(\theta_k)\|_2^2} \sqrt{C_\delta^2 K_0^2 \beta_{k-1}^2 + \mathbb{E}\|\omega_k - \omega_k^*\|_2^2 + \mathcal{O}(\epsilon_{\text{sp}}^2)} \\
&\le \sqrt{\sum_{k=K_0}^{K} \mathbb{E}\|\nabla J(\theta_k)\|_2^2} \sqrt{\sum_{k=K_0}^{K} \left( C_\delta^2 K_0^2 \beta_{k-1}^2 + \mathbb{E}\|\omega_k - \omega_k^*\|_2^2 + \mathcal{O}(\epsilon_{\text{sp}}^2) \right)} \\
&= \sqrt{\sum_{k=K_0}^{K} \mathbb{E}\|\nabla J(\theta_k)\|_2^2} \sqrt{C_\delta^2 K_0^2 \sum_{k=K_0}^{K} \beta_{k-1}^2 + \sum_{k=K_0}^{K} \mathbb{E}\|\omega_k - \omega_k^*\|_2^2 + \mathcal{O}(K\epsilon_{\text{sp}}^2)}, \quad (70)
\end{aligned}
$$

where the first inequality follows Cauchy-Schwartz inequality.

In (70), we have

$$\sum_{k=K_0}^{K} \beta_{k-1}^2 \le \sum_{k=0}^{K-K_0} \beta_k^2 = \sum_{k=0}^{K-K_0} c_2^2 (1+k)^{-2\sigma_2} = \mathcal{O}(K^{1-2\sigma_2}).$$

Substituting the last equality into (70) gives

$$I_5 \le \sqrt{\sum_{k=M_K}^{K} \mathbb{E}\|\nabla J(\theta_k)\|_2^2} \sqrt{\mathcal{O}(K_0^2 K^{1-2\sigma_2}) + \sum_{k=M_K}^{K} \mathbb{E}\|\omega_k - \omega_k^*\|_2^2 + \mathcal{O}(K\epsilon_{\text{sp}}^2)}. \quad (71)$$

Dividing both sides of (67) by $K - K_0 + 1$ and collecting upper bounds of $I_3$, $I_4$ and $I_5$ give

$$
\begin{aligned}
&\frac{1}{K-K_0+1} \sum_{k=K_0}^{K} \mathbb{E}\|\nabla J(\theta_k)\|_2^2 \\
&\le \frac{4C_\psi}{K-K_0+1} \sqrt{\sum_{k=K_0}^{K} \mathbb{E}\|\nabla J(\theta_k)\|_2^2} \sqrt{\mathcal{O}(K_0^2 K^{1-2\sigma_2}) + \sum_{k=K_0}^{K} \mathbb{E}\|\omega_k - \omega_k^*\|_2^2 + \mathcal{O}(K\epsilon_{\text{sp}}^2)} \\
&\quad + \mathcal{O}\left( \frac{1}{K^{1-\sigma_1}} \right) + \mathcal{O}\left( \frac{K_0}{K^{\sigma_1}} \right) + \mathcal{O}(\epsilon_{\text{fa}}). \quad (72)
\end{aligned}
$$

Define the following functions

$$T_4(K) := \frac{1}{K-K_0+1} \sum_{k=K_0}^{K} \mathbb{E}\|\nabla J(\theta_k)\|_2^2,$$

$$T_5(K) := \frac{1}{K-K_0+1} \left( \mathcal{O}(K_0^2 K^{1-2\sigma_2}) + \sum_{k=K_0}^{K} \mathbb{E}\|\omega_k - \omega_k^*\|_2^2 + \mathcal{O}(K\epsilon_{\text{sp}}^2) \right),$$

$$T_6(K) := \mathcal{O}\left( \frac{1}{K^{1-\sigma_1}} \right) + \mathcal{O}\left( \frac{K_0}{K^{\sigma_1}} \right) + \mathcal{O}(\epsilon_{\text{fa}}).$$

Then (72) can be rewritten as

$$T_4(K) \le T_6(K) + \sqrt{2}(1+\gamma)C_\psi \sqrt{T_4(K)} \sqrt{T_5(K)}.$$

Solving this quadratic inequality in terms of $T_4(K)$, we obtain

$$T_4(K) \leq 2T_6(K) + 4(1+\gamma)^2 C_\psi^2 T_5(K),\tag{73}$$

which implies

$$\frac{1}{K-K_0+1}\sum_{k=K_0}^{K}\mathbb{E}\,\|\nabla J(\theta_k)\|_2^2$$

$$=\mathcal{O}\left(\frac{1}{K^{1-\sigma_1}}\right)+\mathcal{O}\left(\frac{K_0}{K^{\sigma_1}}\right)+\mathcal{O}\left(\frac{K_0^2}{K^{2\sigma_2}}\right)+\mathcal{O}\left(\frac{1}{K-K_0+1}\sum_{k=K_0}^{K}\mathbb{E}\,\|\omega_k-\omega_k^*\|_2^2\right)+\mathcal{O}(\epsilon_{\mathrm{app}}).$$

We further have

$$\frac{1}{K}\sum_{k=1}^{K}\mathbb{E}\,\|\nabla J(\theta_k)\|_2^2 \leq \frac{1}{K}\left(\sum_{k=1}^{K_0-1}L_V^2+\sum_{k=K_0}^{K}\mathbb{E}\,\|\nabla J(\theta_k)\|_2^2\right)$$

$$=\frac{K_0-1}{K}L_V^2+\frac{K-K_0+1}{K}\frac{1}{K-K_0+1}\sum_{k=K_0}^{K}\mathbb{E}\,\|\nabla J(\theta_k)\|_2^2$$

$$=\mathcal{O}\left(\frac{K_0}{K}\right)+\mathcal{O}\left(\frac{1}{K-K_0+1}\sum_{k=K_0}^{K}\mathbb{E}\,\|\nabla J(\theta_k)\|_2^2\right)$$

$$=\mathcal{O}\left(\frac{1}{K-K_0+1}\sum_{k=K_0}^{K}\mathbb{E}\,\|\nabla J(\theta_k)\|_2^2\right)\tag{74}$$

which completes the proof. $\qquad\square$

### B.3 Proof of Theorem 3

Given the definition in Section B.1, we now give the convergence proof of critic update in Algorithm 1 with linear function approximation and Markovian sampling.

By following the derivation of (40), we have

$$\mathbb{E}\,\|\omega_{k+1}-\omega_{k+1}^*\|_2^2 \leq (1-2\lambda\beta_k)\,\mathbb{E}\,\|\omega_k-\omega_k^*\|_2^2+2\beta_k(C_1\frac{\alpha_k}{\beta_k}+C_2K_0\beta_{k-K_0})\,\mathbb{E}\,\|\omega_k-\omega_k^*\|_2$$

$$+2\beta_k\,\mathbb{E}\left\langle\omega_k-\omega_k^*,g(x_{(k)},\omega_k)-\overline{g}(\theta_k,\omega_k)\right\rangle+C_q\beta_k^2,\tag{75}$$

where $C_1 := C_pL_\omega$, $C_2 := C_\delta(1+\gamma)$ and $C_q := 2C_\delta^2+2L_\omega^2C_p^2\max_{(k)}\frac{\alpha_k^2}{\beta_k^2}=2C_\delta^2+2L_\omega^2C_p^2\frac{c_1^2}{c_2^2}$.

Now we consider the third item in the last inequality. For some $m\in\mathbb{N}^+$, we define $M := (K_0+1)m+K_0$. Following Lemma 4 (to be presented in Sec. C.1), for some $d_m\leq M$ and positive constants $C_4,C_5,C_6,C_7$, we have

$$\mathbb{E}\left\langle\omega_k-\omega_k^*,g(x_{(k)},\omega_k)-\overline{g}(\theta_k,\omega_k)\right\rangle$$

$$\leq C_4\,\mathbb{E}\,\|\theta_k-\theta_{k-d_m}\|_2+C_5\sum_{i=\tau_k}^{d_m}\mathbb{E}\,\|\theta_{k-i}-\theta_{k-d_m}\|_2+C_6\,\mathbb{E}\,\|\omega_k-\omega_{k-d_m}\|_2+C_7\kappa\rho^{m-1}$$

$$\leq C_4\sum_{i=k-d_m}^{k-1}\mathbb{E}\,\|\theta_{i+1}-\theta_i\|_2+C_5\sum_{i=\tau_k}^{d_m-1}\sum_{j=k-d_m}^{k-i-1}\mathbb{E}\,\|\theta_{j+1}-\theta_j\|_2+C_6\sum_{i=k-d_m}^{k-1}\mathbb{E}\,\|\omega_{i+1}-\omega_i\|_2+C_7\kappa\rho^{m-1}$$

$$\leq C_4\sum_{i=k-d_m}^{k-1}\alpha_iC_p+C_5\sum_{i=\tau_k}^{d_m-1}\sum_{j=k-d_m}^{k-i-1}\alpha_jC_p+C_6\sum_{i=k-d_m}^{k-1}\beta_iC_\delta+C_7\kappa\rho^{m-1}$$

$$\leq C_4\alpha_{k-d_m}\sum_{i=k-d_m}^{k-1}C_p+C_5\alpha_{k-d_m}\sum_{i=\tau_k}^{d_m-1}\sum_{j=k-d_m}^{k-i-1}C_p+C_6\beta_{k-d_m}\sum_{i=k-d_m}^{k-1}C_\delta+C_7\kappa\rho^{m-1}$$

$$\leq C_4d_mC_p\alpha_{k-d_m}+C_5(d_m-\tau_k)^2C_p\alpha_{k-d_m}+C_6d_mC_\delta\beta_{k-d_m}+C_7\kappa\rho^{m-1}$$

$$\leq \left(C_4M+C_5M^2\right)C_p\alpha_{k-M}+C_6MC_\delta\beta_{k-M}+C_7\kappa\rho^{m-1},\tag{76}$$

where the third last inequality is due to the monotonicity of step size, and the last inequality is due to $\tau_k \geq 0$ and $d_m \leq M$.

Further letting $m = m_K$ which is defined in (21) yields

$$
\begin{aligned}
&\mathbb{E}\left\langle \omega_k - \omega_k^*, g(x_{(k)}, \omega_k) - \overline{g}(\theta_k, \omega_k) \right\rangle \\
&= \left( C_4 M_K + C_5 M_K^2 \right) C_p \alpha_{k-M_K} + C_6 C_\delta M_K \beta_{k-M_K} + C_7 \kappa \rho^{m_K - 1} \\
&\leq \left( C_4 M_K + C_5 M_K^2 \right) C_p \alpha_{k-M_K} + C_6 C_\delta M_K \beta_{k-M_K} + C_7 \alpha_K,
\end{aligned}
\tag{77}
$$

where $M_K = (K_0 + 1)m_K + K_0$, and the last inequality follows the definition of $m_K$.

Substituting (77) into (75), then rearranging and summing up both sides over $k = M_K, ..., K$ yield

$$
2\lambda \sum_{k=M_K}^{K} \mathbb{E}\|\omega_k - \omega_k^*\|_2^2 \leq \underbrace{\sum_{k=M_K}^{K} \frac{1}{\beta_k} \left( \mathbb{E}\|\omega_k - \omega_k^*\|_2^2 - \mathbb{E}\left\|\omega_{k+1} - \omega_{k+1}^*\right\|_2^2 \right)}_{I_1} + \underbrace{C_q \sum_{k=M_K}^{K} \beta_k}_{I_2}
$$

$$
+ 2 \underbrace{\sum_{k=M_K}^{K} \left( \left( C_4 M_K + C_5 M_K^2 \right) C_p \alpha_{k-M_K} + C_6 C_\delta M_K \beta_{k-M_K} + C_7 \alpha_K \right)}_{I_3}
$$

$$
+ 2 \underbrace{\sum_{k=M_K}^{K} \left( C_1 \frac{\alpha_k}{\beta_k} + C_2 K_0 \beta_{k-K_0} \right) \mathbb{E}\|\omega_k - \omega_k^*\|_2}_{I_4}.
\tag{78}
$$

where the order of $I_1$, $I_2$ and $I_4$ have already been given by (45), (46) and (50) respectively.
We bound $I_3$ as

$$
\begin{aligned}
I_3 &= \left( C_4 M_K + C_5 M_K^2 \right) C_p \sum_{k=M_K}^{K} \alpha_k + C_6 C_\delta M_K \sum_{k=M_K}^{K} \beta_k + C_7 \alpha_K \sum_{k=M_K}^{K} 1 \\
&\leq \left( C_4 M_K + C_5 M_K^2 \right) C_p c_1 \frac{K^{1-\sigma_1}}{1-\sigma_1} + C_6 C_\delta M_K c_2 \frac{K^{1-\sigma_2}}{1-\sigma_2} + C_7 c_1 K (1+K)^{-\sigma_1} \\
&= \mathcal{O}\left( (K_0^2 \log^2 K) K^{1-\sigma_1} \right) + \mathcal{O}\left( (K_0 \log K) K^{1-\sigma_2} \right),
\end{aligned}
\tag{79}
$$

where the last inequality follows from the integration rule $\sum_{k=a}^{b} k^{-\sigma} \leq \frac{b^{1-\sigma}}{1-\sigma}$, and the last equality is due to $\mathcal{O}(M_K) = \mathcal{O}(K_0 m_K) = \mathcal{O}(K_0 \log K)$.

Collecting the bounds of $I_1$, $I_2$, $I_3$ and $I_4$, and dividing both sides of (78) by $K - M_K + 1$ yield

$$
2\lambda \frac{1}{K - M_K + 1} \sum_{k=M_K}^{K} \mathbb{E}\|\omega_k - \omega_k^*\|_2^2
$$

$$
\leq \frac{\sqrt{\mathcal{O}\left( K^{2(\sigma_2 - \sigma_1)+1} \right) + \mathcal{O}\left( K_0 K^{-\sigma_1 + 1} \right) + \mathcal{O}\left( K_0^2 K^{1-2\sigma_2} \right)}}{K - M_K + 1} \sqrt{\sum_{k=M_K}^{K} \mathbb{E}\|\omega_k - \omega_k^*\|_2^2}
$$

$$
+ \mathcal{O}\left( \frac{1}{K^{1-\sigma_2}} \right) + \mathcal{O}\left( \frac{K_0^2 \log^2 K}{K^{\sigma_1}} \right) + \mathcal{O}\left( \frac{K_0 \log K}{K^{\sigma_2}} \right).
\tag{80}
$$

Similar to the derivation of (52), (80) implies

$$
\frac{1}{K - M_K + 1} \sum_{k=M_K}^{K} \mathbb{E}\|\omega_k - \omega_k^*\|_2^2
$$

$$
= \mathcal{O}\left( \frac{1}{K^{1-\sigma_2}} \right) + \mathcal{O}\left( \frac{1}{K^{2(\sigma_1 - \sigma_2)}} \right) + \mathcal{O}\left( \frac{K_0^2}{K^{2\sigma_2}} \right) + \mathcal{O}\left( \frac{K_0^2 \log^2 K}{K^{\sigma_1}} \right) + \mathcal{O}\left( \frac{K_0 \log K}{K^{\sigma_2}} \right).
$$

Similar to (53), we have

$$\frac{1}{K}\sum_{k=1}^{K}\mathbb{E}\left\|\omega_k-\omega_k^*\right\|_2^2 = \mathcal{O}\left(\frac{K_0\log K}{K}\right) + \mathcal{O}\left(\frac{1}{K-M_K+1}\sum_{k=M_K}^{K}\mathbb{E}\left\|\omega_k-\omega_k^*\right\|_2^2\right)$$

$$= \mathcal{O}\left(\frac{1}{K-M_K+1}\sum_{k=M_K}^{K}\mathbb{E}\left\|\omega_k-\omega_k^*\right\|_2^2\right) \tag{81}$$

which completes the proof. □

## B.4 Proof of Theorem 4

Given the definition in section B.2, we now give the convergence proof of actor update in Algorithm 1 with linear value function approximation and Markovian sampling method.

By following the derivation of (55), we have

$$\mathbb{E}[J(\theta_{k+1})] \geq \mathbb{E}[J(\theta_k)] + \alpha_k \underbrace{\mathbb{E}\left\langle\nabla J(\theta_k), \left(\hat{\delta}(x_{(k)},\omega_{k-\tau_k}) - \hat{\delta}(x_{(k)},\omega_k^*)\right)\psi_{\theta_{k-\tau_k}}(s_{(k)},a_{(k)})\right\rangle}_{I_1}$$

$$+ \alpha_k \underbrace{\mathbb{E}\left\langle\nabla J(\theta_k), \hat{\delta}(x_{(k)},\omega_k^*)\psi_{\theta_{k-\tau_k}}(s_{(k)},a_{(k)})\right\rangle}_{I_2} - \frac{L_J}{2}C_p^2\alpha_k^2. \tag{82}$$

The item $I_1$ can be bounded by following (56) as

$$I_1 \geq -2C_\psi\,\mathbb{E}\left[\|\nabla J(\theta_k)\|_2\left(C_\delta K_0\beta_{k-1} + \|\omega_k-\omega_k^*\|_2\right)\right]. \tag{83}$$

Next we consider $I_2$. We first decompose it as

$$I_2 = \mathbb{E}\left\langle\nabla J(\theta_k), \hat{\delta}(x_{(k)},\omega_k^*)\psi_{\theta_{k-\tau_k}}(s_{(k)},a_{(k)})\right\rangle$$

$$= \underbrace{\mathbb{E}\left\langle\nabla J(\theta_k), \left(\hat{\delta}(x_{(k)},\omega_k^*) - \delta(x_{(k)},\theta_k)\right)\psi_{\theta_{k-\tau_k}}(s_{(k)},a_{(k)})\right\rangle}_{I_2^{(1)}}$$

$$+ \underbrace{\mathbb{E}\left\langle\nabla J(\theta_k), \delta(x_{(k)},\theta_k)\psi_{\theta_{k-\tau_k}}(s_{(k)},a_{(k)}) - \nabla J(\theta_k)\right\rangle}_{I_2^{(2)}} + \mathbb{E}\left\|\nabla J(\theta_k)\right\|_2^2. \tag{84}$$

For some $m\in\mathbb{N}^+$, define $M \coloneqq (K_0+1)m + K_0$. Following Lemma 5, for some $d_m \leq M$ and positive constants $D_2, D_3, D_4, D_5$, $I_2^{(1)}$ can be bounded as

$$I_2^{(1)} = \mathbb{E}\left\langle\nabla J(\theta_k), \left(\hat{\delta}(x_{(k)},\omega_k^*) - \delta(x_{(k)},\theta_k)\right)\psi_{\theta_{k-\tau_k}}(s_{(k)},a_{(k)})\right\rangle$$

$$\geq -D_2\,\mathbb{E}\left\|\theta_{k-\tau_k}-\theta_{k-d_m}\right\|_2 - D_3\,\mathbb{E}\left\|\theta_k-\theta_{k-d_m}\right\|_2 - D_4\sum_{i=k-d_m}^{k-\tau_k}\mathbb{E}\left\|\theta_i-\theta_{k-d_m}\right\|_2$$

$$- D_5\kappa\rho^{m-1} - 2C_\psi L_V\epsilon_{\mathrm{fa}} - 2C_\psi\epsilon_{\mathrm{sp}}\,\mathbb{E}\left\|\nabla J(\theta_k)\right\|_2$$

$$\geq -D_2(d_m-\tau_k)C_p\alpha_{k-d_m} - D_3 d_m C_p\alpha_{k-d_m} - D_4(d_m-\tau_k)^2 C_p\alpha_{k-d_m}$$

$$- D_5\kappa\rho^{m-1} - 2C_\psi L_V\epsilon_{\mathrm{fa}} - 2C_\psi\epsilon_{\mathrm{sp}}\,\mathbb{E}\left\|\nabla J(\theta_k)\right\|_2, \tag{85}$$

where the derivation of the last inequality is similar to that of (76).

By setting $m = m_K$ in (85), and following the fact that $d_{m_K} \leq M_K$ and $\tau_k \geq 0$, we have

$$I_2^{(1)} \geq -D_2 M_K C_p\alpha_{k-M_K} - D_3 M_K C_p\alpha_{k-M_K} - D_4 M_K^2 C_p\alpha_{k-M_K} - D_5\kappa\rho^{m_K-1}$$

$$- 2C_\psi L_V\epsilon_{\mathrm{fa}} - 2C_\psi\epsilon_{\mathrm{sp}}\,\mathbb{E}\left\|\nabla J(\theta)\right\|_2$$

$$= -\left((D_2+D_3)C_p M_K + D_4 C_p M_K^2\right)\alpha_{k-M_K} - D_5\kappa\rho^{m_K-1} - 2C_\psi L_V\epsilon_{\mathrm{fa}} - 2C_\psi\epsilon_{\mathrm{sp}}\,\mathbb{E}\left\|\nabla J(\theta_k)\right\|_2$$

$$\geq -\left((D_2+D_3)C_p M_K + D_4 C_p M_K^2\right)\alpha_{k-M_K} - D_5\alpha_K - 2C_\psi L_V\epsilon_{\mathrm{fa}} - 2C_\psi\epsilon_{\mathrm{sp}}\,\mathbb{E}\left\|\nabla J(\theta_k)\right\|_2, \tag{86}$$

where the last inequality is due to the definition of $m_K$.

Following Lemma 6, for some positive constants $D_6, D_7, D_8$ and $D_9$, we bound $I_2^{(2)}$ as

$$I_2^{(2)} = \mathbb{E}\left\langle \nabla J(\theta_k), \delta(x_{(k)}, \theta_k)\psi_{\theta_{k-\tau_k}}(s_{(k)}, a_{(k)}) - \nabla J(\theta_k)\right\rangle$$

$$\geq -D_6 \,\mathbb{E}\,\|\theta_{k-\tau_k} - \theta_{k-d_m}\|_2 - D_7\,\mathbb{E}\,\|\theta_k - \theta_{k-d_m}\|_2 - D_8 \sum_{i=\tau_k}^{d_m} \mathbb{E}\,\|\theta_{k-i} - \theta_{k-d_m}\|_2$$

$$- D_9 \kappa \rho^{m-1} - 8C_\psi r_{\max}(1-\gamma)\,\mathbb{E}\,\|\nabla J(\theta_k)\|_2 \,.$$

Similar to the derivation of (86), we have

$$I_2^{(2)} \geq -\left(D_6 + D_7 + D_8 M_K\right)C_p M_K \alpha_{k-M_K} - D_9 \alpha_K - 8C_\psi r_{\max}(1-\gamma)\,\mathbb{E}\,\|\nabla J(\theta_k)\|_2 \,. \quad (87)$$

Collecting the lower bounds of $I_2^{(1)}$ and $I_2^{(2)}$ yields

$$I_2 \geq -2C_\psi L_V \epsilon_{\text{fa}} - 2C_\psi\left(\epsilon_{\text{sp}} + 4r_{\max}(1-\gamma)\right)\mathbb{E}\,\|\nabla J(\theta_k)\|_2 + \mathbb{E}\,\|\nabla J(\theta_k)\|_2^2$$
$$- D_K \alpha_{k-M_K} - (D_5 + D_9)\alpha_K, \quad (88)$$

where we define $D_K := (D_4 + D_8)C_p M_K^2 + (D_2 + D_3 + D_6 + D_7)C_p M_K$ for brevity.

Substituting (83) and (88) into (82) yields

$$\mathbb{E}[J(\theta_{k+1})] \geq \mathbb{E}[J(\theta_k)] - 2\alpha_k C_\psi\,\mathbb{E}\left[\|\nabla J(\theta_k)\|_2\left(\epsilon_{\text{sp}} + 4r_{\max}(1-\gamma) + C_\delta K_0\beta_{k-1} + \|\omega_k - \omega_k^*\|_2\right)\right]$$

$$- \alpha_k\left(D_K\alpha_{k-M_K} + (D_5 + D_9)\alpha_K\right) - 2C_\psi L_V \epsilon_{\text{fa}}\alpha_k + \alpha_k\,\mathbb{E}\,\|\nabla J(\theta_k)\|_2^2 - \frac{L_J}{2}C_p^2\alpha_k^2.$$

Similar to the derivation of (67), the last inequality implies

$$\mathbb{E}[J(\theta_{k+1})] \geq \mathbb{E}[J(\theta_k)] - 4\alpha_k C_\psi\sqrt{\mathbb{E}\,\|\nabla J(\theta_k)\|_2^2}\sqrt{C_\delta^2 K_0^2\beta_{k-1}^2 + \mathbb{E}\,\|\omega_k - \omega_k^*\|_2^2 + \mathcal{O}(\epsilon_{\text{sp}}^2)}$$

$$- \alpha_k\left(D_K\alpha_{k-M_K} + (D_5 + D_9)\alpha_K\right) - 2C_\psi L_V \epsilon_{\text{fa}}\alpha_k + \alpha_k\,\mathbb{E}\,\|\nabla J(\theta_k)\|_2^2 - \frac{L_J}{2}C_p^2\alpha_k^2.$$

Rearranging and dividing both sides by $\alpha_k$ yield

$$\mathbb{E}\,\|\nabla J(\theta_k)\|_2^2 \leq \frac{1}{\alpha_k}\left(\mathbb{E}[J(\theta_{k+1})] - \mathbb{E}[J(\theta_k)]\right) + D_K\alpha_{k-M_K} + (D_5 + D_9)\alpha_K + \frac{L_J}{2}C_p^2\alpha_k$$

$$+ 4C_\psi\sqrt{\mathbb{E}\,\|\nabla J(\theta_k)\|_2^2}\sqrt{C_\delta^2 K_0^2\beta_{k-1}^2 + \mathbb{E}\,\|\omega_k - \omega_k^*\|_2^2 + \mathcal{O}(\epsilon_{\text{sp}}^2)} + 2C_\psi L_V \epsilon_{\text{fa}}.$$

Taking summation gives

$$\sum_{k=M_K}^K \mathbb{E}\,\|\nabla J(\theta_k)\|_2^2 \leq \underbrace{\sum_{k=M_K}^K \frac{1}{\alpha_k}\left(\mathbb{E}[J(\theta_{k+1})] - \mathbb{E}[J(\theta_k)]\right)}_{I_3}$$

$$+ \underbrace{\sum_{k=M_K}^K \left(D_K\alpha_{k-M_K} + \frac{L_J}{2}C_p^2\alpha_k + (D_5 + D_9)\alpha_K\right)}_{I_4}$$

$$+ \underbrace{4C_\psi \sum_{k=M_K}^K \sqrt{\mathbb{E}\,\|\nabla J(\theta_k)\|_2^2}\sqrt{C_\delta^2 K_0^2\beta_{k-1}^2 + \mathbb{E}\,\|\omega_k - \omega_k^*\|_2^2 + \mathcal{O}(\epsilon_{\text{sp}}^2)}}_{I_5}$$

$$+ 2C_\psi L_V(K - M_K + 1)\epsilon_{\text{fa}}. \quad (89)$$

in which the upper bounds of $I_3$ and $I_5$ have already been given by (69) and (71) respectively.

We bound $I_4$ as

$$
\begin{aligned}
I_4 &= \sum_{k=M_K}^{K} \left( D_K \alpha_{k-M_K} + \frac{L_J}{2} C_p^2 \alpha_k + (D_5 + D_9)\alpha_K \right) \\
&\leq \sum_{k=M_K}^{K} \left( D_K \alpha_{k-M_K} + \frac{L_J}{2} C_p^2 \alpha_{k-M_K} + (D_5 + D_9)\alpha_K \right) \\
&= \left( D_K + \frac{L_J}{2} C_p^2 \right) \sum_{k=M_K}^{K} \alpha_{k-M_K} + (D_5 + D_9)(K - M_K + 1)\alpha_K \\
&= \left( D_K + \frac{L_J}{2} C_p^2 \right) \sum_{k=0}^{K-M_K} \alpha_k + (D_5 + D_9)(K - M_K + 1)\alpha_K \\
&\leq \left( D_K + \frac{L_J}{2} C_p^2 \right) \frac{c_1}{1 - \sigma_1} K^{1-\sigma_1} + c_1(D_5 + D_9)(K + 1)^{1-\sigma_1} \\
&= \mathcal{O}\left( (K_0^2 \log^2 K) K^{1-\sigma_1} \right)
\end{aligned}
\tag{90}
$$

where the last inequality uses $\sum_{k=a}^{b} k^{-\sigma} \leq \frac{b^{1-\sigma}}{1-\sigma}$, and the last equality is due to the fact that

$$
\mathcal{O}(D_K) = \mathcal{O}(M_K^2 + M_K) = \mathcal{O}((K_0 m_K)^2 + K_0 m_K) = \mathcal{O}(K_0^2 \log^2 K).
$$

Substituting the upper bounds of $I_3$, $I_4$ and $I_5$ into (89), and dividing both sides by $K - M_K + 1$ give

$$
\begin{aligned}
&\frac{1}{K - M_K + 1} \sum_{k=M_K}^{K} \mathbb{E} \|\nabla J(\theta_k)\|_2^2 \\
&\leq \frac{4C_\psi}{K - M_K + 1} \sqrt{\sum_{k=M_K}^{K} \mathbb{E} \|\nabla J(\theta_k)\|_2^2} \sqrt{\mathcal{O}(K_0^2 K^{1-2\sigma_2}) + \sum_{k=M_K}^{K} \mathbb{E} \|\omega_k - \omega_k^*\|_2^2 + \mathcal{O}(K\epsilon_{\text{sp}}^2)} \\
&\quad + \mathcal{O}\left( \frac{1}{K^{1-\sigma_1}} \right) + \mathcal{O}\left( \frac{K_0^2 \log^2 K}{K^{\sigma_1}} \right) + \mathcal{O}(\epsilon_{\text{fa}}).
\end{aligned}
\tag{91}
$$

Following the similar steps of those in (73), (91) essentially implies

$$
\begin{aligned}
&\frac{1}{K - M_K + 1} \sum_{k=M_K}^{K} \mathbb{E} \|\nabla J(\theta_k)\|_2^2 \\
&= \mathcal{O}\left( \frac{1}{K^{1-\sigma_1}} \right) + \mathcal{O}\left( \frac{K_0^2 \log^2 K}{K^{\sigma_1}} \right) + \mathcal{O}\left( \frac{K_0^2}{K^{2\sigma_2}} \right) + \mathcal{O}\left( \frac{1}{K - M_K + 1} \sum_{k=M_K}^{K} \mathbb{E} \|\omega_k - \omega_{\theta_k}^*\|_2^2 \right) + \mathcal{O}(\epsilon_{\text{app}}).
\end{aligned}
$$

Similar to (74), we have

$$
\begin{aligned}
\frac{1}{K} \sum_{k=1}^{K} \mathbb{E} \|\nabla J(\theta_k)\|_2^2 &= \mathcal{O}\left( \frac{K_0 \log K}{K} \right) + \mathcal{O}\left( \frac{1}{K - M_K + 1} \sum_{k=M_K}^{K} \mathbb{E} \|\nabla J(\theta_k)\|_2^2 \right) \\
&= \mathcal{O}\left( \frac{1}{K - M_K + 1} \sum_{k=M_K}^{K} \mathbb{E} \|\nabla J(\theta_k)\|_2^2 \right)
\end{aligned}
$$

which completes the proof. □

## C SUPPORTING LEMMAS

### C.1 SUPPORTING LEMMAS FOR THEOREM 3

**Lemma 4.** *For any $m \geq 1$ and $k \geq (K_0 + 1)m + K_0 + 1$, we have*

$$\mathbb{E}\left\langle \omega_k - \omega_{\theta_k}^*, g(x_{(k)}, \omega_k) - \overline{g}(\theta_k, \omega_k) \right\rangle \leq C_4 \, \mathbb{E}\left\| \theta_k - \theta_{k-d_m} \right\|_2 + C_5 \sum_{i=\tau_k}^{d_m} \mathbb{E}\left\| \theta_{k-i} - \theta_{k-d_m} \right\|_2$$
$$+ C_6 \, \mathbb{E}\left\| \omega_k - \omega_{k-d_m} \right\|_2 + C_7 \kappa \rho^{m-1},$$

*where $d_m \leq (K_0 + 1)m + K_0$, and $C_4 := 2C_\delta L_\omega + 4R_\omega C_\delta |\mathcal{A}| L_\pi (1 + \log_\rho \kappa^{-1} + (1 - \rho)^{-1})$, $C_5 := 4R_\omega C_\delta |\mathcal{A}| L_\pi$ and $C_6 := 4(1 + \gamma) R_\omega + 2C_\delta$, $C_7 := 8R_\omega C_\delta$.*

*Proof.* Consider the collection of random samples $\{x_{(k-K_0-1)}, x_{(k-K_0)}, ..., x_{(k)}\}$. Suppose $x_{(k)}$ is sampled by worker $n$, then due to Assumption 1, $\{x_{(k-K_0-1)}, x_{(k-K_0)}, ..., x_{(k-1)}\}$ will contain at least another sample drawn by worker $n$. Therefore, $\{x_{(k-(K_0+1)m)}, x_{(k-(K_0+1)m+1)}, ..., x_{(k-1)}\}$ will contain at least $m$ samples from worker $n$.

Consider the Markov chain formed by $m + 1$ samples in $\{x_{(k-(K_0+1)m)}, x_{(k-(K_0+1)m+1)}, ..., x_{(k)}\}$:

$$s_{t-m} \xrightarrow{\theta_{k-d_m}} a_{t-m} \xrightarrow{\widetilde{\mathcal{P}}} s_{t-m+1} \xrightarrow{\theta_{k-d_{m-1}}} a_{t-m+1} \cdots s_{t-1} \xrightarrow{\theta_{k-d_1}} a_{t-1} \xrightarrow{\widetilde{\mathcal{P}}} s_t \xrightarrow{\theta_{k-d_0}} a_t \xrightarrow{\widetilde{\mathcal{P}}} s_{t+1},$$

where $(s_t, a_t, s_{t+1}) = (s_{(k)}, a_{(k)}, s'_{(k)})$, and $\{d_j\}_{j=0}^m$ is some increasing sequence with $d_0 := \tau_k$.

Suppose $\theta_{k-d_m}$ was used to do the $k_m$th update, then we have $x_{t-m} = x_{(k_m)}$. Following Assumption 1, we have $\tau_{k_m} = k_m - (k - d_m) \leq K_0$. Since $x_{(k_m)}$ is in $\{x_{(k-(K_0+1)m)}, ..., x_{(k)}\}$, we have $k_m \geq k - (K_0 + 1)m$. Combining these two inequalities, we have

$$d_m \leq (K_0 + 1)m + K_0. \tag{92}$$

Given $(s_{t-m}, a_{t-m}, s_{t-m+1})$ and $\theta_{k-d_m}$, we construct an auxiliary Markov chain as that in Lemma 2:

$$s_{t-m} \xrightarrow{\theta_{k-d_m}} a_{t-m} \xrightarrow{\widetilde{\mathcal{P}}} s_{t-m+1} \xrightarrow{\theta_{k-d_m}} \widetilde{a}_{t-m+1} \cdots \widetilde{s}_{t-1} \xrightarrow{\theta_{k-d_m}} \widetilde{a}_{t-1} \xrightarrow{\widetilde{\mathcal{P}}} \widetilde{s}_t \xrightarrow{\theta_{k-d_m}} \widetilde{a}_t \xrightarrow{\widetilde{\mathcal{P}}} \widetilde{s}_{t+1}.$$

For brevity, we define

$$\Delta_1(x, \theta, \omega) := \left\langle \omega - \omega_\theta^*, g(x, \omega) - \overline{g}(\theta, \omega) \right\rangle.$$

Throughout this proof, we use $\theta$, $\theta'$, $\omega$, $\omega'$, $x$ and $\widetilde{x}$ as shorthand notations of $\theta_k$, $\theta_{k-d_m}$, $\omega_k$, $\omega_{k-d_m}$, $x_t$ and $\widetilde{x}_t$ respectively.

First we decompose $\Delta_1(x, \theta, \omega)$ as

$$\Delta_1(x, \theta, \omega) = \underbrace{\Delta_1(x, \theta, \omega) - \Delta_1(x, \theta', \omega)}_{I_1} + \underbrace{\Delta_1(x, \theta', \omega) - \Delta_1(x, \theta', \omega')}_{I_2}$$
$$+ \underbrace{\Delta_1(x, \theta', \omega') - \Delta_1(\widetilde{x}, \theta', \omega')}_{I_3} + \underbrace{\Delta_1(\widetilde{x}, \theta', \omega')}_{I_4}. \tag{93}$$

We bound $I_1$ in (93) as

$$\Delta_1(x, \theta, \omega) - \Delta_1(x, \theta', \omega) = \left\langle \omega - \omega_\theta^*, g(x, \omega) - \overline{g}(\theta, \omega) \right\rangle - \left\langle \omega - \omega_{\theta'}^*, g(x, \omega) - \overline{g}(\theta', \omega) \right\rangle$$
$$\leq \left| \left\langle \omega - \omega_\theta^*, g(x, \omega) - \overline{g}(\theta, \omega) \right\rangle - \left\langle \omega - \omega_{\theta'}^*, g(x, \omega) - \overline{g}(\theta, \omega) \right\rangle \right|$$
$$+ \left| \left\langle \omega - \omega_{\theta'}^*, g(x, \omega) - \overline{g}(\theta, \omega) \right\rangle - \left\langle \omega - \omega_{\theta'}^*, g(x, \omega) - \overline{g}(\theta', \omega) \right\rangle \right|. \tag{94}$$

For the first term in (94), we have

$$\left| \left\langle \omega - \omega_\theta^*, g(x, \omega) - \overline{g}(\theta, \omega) \right\rangle - \left\langle \omega - \omega_{\theta'}^*, g(x, \omega) - \overline{g}(\theta, \omega) \right\rangle \right| = \left| \left\langle \omega_\theta^* - \omega_{\theta'}^*, g(x, \omega) - \overline{g}(\theta, \omega) \right\rangle \right|$$
$$\leq \left\| \omega_\theta^* - \omega_{\theta'}^* \right\|_2 \left\| g(x, \omega) - \overline{g}(\theta, \omega) \right\|$$
$$\leq 2C_\delta \left\| \omega_\theta^* - \omega_{\theta'}^* \right\|_2$$
$$\leq 2C_\delta L_\omega \left\| \theta - \theta' \right\|_2,$$

where the last inequality is due to Proposition 2.

We use $x \sim \theta'$ as shorthand notations to represent that $s \sim \mu_{\theta'}$, $a \sim \pi_{\theta'}$, $s' \sim \widetilde{\mathcal{P}}$. For the second term in (94), we have

$$
\begin{aligned}
&|\langle \omega - \omega_{\theta'}^*, g(x,\omega) - \overline{g}(\theta,\omega)\rangle - \langle \omega - \omega_{\theta'}^*, g(x,\omega) - \overline{g}(\theta',\omega)\rangle| \\
&= |\langle \omega - \omega_{\theta'}^*, \overline{g}(\theta',\omega) - \overline{g}(\theta,\omega)\rangle| \\
&\leq \|\omega - \omega_{\theta'}^*\|_2 \|\overline{g}(\theta',\omega) - \overline{g}(\theta,\omega)\|_2 \\
&\leq 2R_\omega \|\overline{g}(\theta',\omega) - \overline{g}(\theta,\omega)\|_2 \\
&= 2R_\omega \left\| \mathbb{E}_{x\sim\theta'}[g(x,\omega)] - \mathbb{E}_{x\sim\theta}[g(x,\omega)] \right\|_2 \\
&\leq 2R_\omega \sup_x \|g(x,\omega)\|_2 \|\mu_{\theta'} \otimes \pi_{\theta'} \otimes \widetilde{\mathcal{P}} - \mu_\theta \otimes \pi_\theta \otimes \widetilde{\mathcal{P}}\|_{TV} \\
&\leq 2R_\omega C_\delta \|\mu_{\theta'} \otimes \pi_{\theta'} \otimes \widetilde{\mathcal{P}} - \mu_\theta \otimes \pi_\theta \otimes \widetilde{\mathcal{P}}\|_{TV} \\
&= 4R_\omega C_\delta d_{TV}\left(\mu_{\theta'} \otimes \pi_{\theta'} \otimes \widetilde{\mathcal{P}}, \mu_\theta \otimes \pi_\theta \otimes \widetilde{\mathcal{P}}\right) \\
&\leq 4R_\omega C_\delta |\mathcal{A}| L_\pi (1 + \log_\rho \kappa^{-1} + (1-\rho)^{-1})\|\theta - \theta'\|_2,
\end{aligned}
$$

where the third inequality follows the definition of TV norm, the second last inequality follows (32), and the last inequality follows Lemma A.1. in [17].

Collecting the upper bounds of the two terms in (94) yields

$$
I_1 \leq \left[2C_\delta L_\omega + 4R_\omega C_\delta |\mathcal{A}| L_\pi (1 + \log_\rho \kappa^{-1} + (1-\rho)^{-1})\right] \|\theta - \theta'\|_2.
$$

Next we bound $\mathbb{E}[I_2]$ in (93) as

$$
\begin{aligned}
\mathbb{E}[I_2] &= \mathbb{E}[\Delta_1(x,\theta',\omega) - \Delta_1(x,\theta',\omega')] \\
&= \mathbb{E}\langle \omega - \omega_{\theta'}^*, g(x,\omega) - \overline{g}(\theta',\omega)\rangle - \langle \omega' - \omega_{\theta'}^*, g(x,\omega') - \overline{g}(\theta',\omega')\rangle \\
&\leq \mathbb{E}|\langle \omega - \omega_{\theta'}^*, g(x,\omega) - \overline{g}(\theta',\omega)\rangle - \langle \omega - \omega_{\theta'}^*, g(x,\omega') - \overline{g}(\theta',\omega')\rangle| \\
&\quad + \mathbb{E}|\langle \omega - \omega_{\theta'}^*, g(x,\omega') - \overline{g}(\theta',\omega')\rangle - \langle \omega' - \omega_{\theta'}^*, g(x,\omega') - \overline{g}(\theta',\omega')\rangle|. \quad (95)
\end{aligned}
$$

We bound the first term in (95) as

$$
\begin{aligned}
&\mathbb{E}|\langle \omega - \omega_{\theta'}^*, g(x,\omega) - \overline{g}(\theta',\omega)\rangle - \langle \omega - \omega_{\theta'}^*, g(x,\omega') - \overline{g}(\theta',\omega')\rangle| \\
&= \mathbb{E}|\langle \omega - \omega_{\theta'}^*, g(x,\omega) - g(x,\omega') + \overline{g}(\theta',\omega') - \overline{g}(\theta',\omega)\rangle| \\
&\leq 2R_\omega \left(\mathbb{E}\|g(x,\omega) - g(x,\omega')\|_2 + \mathbb{E}\|\overline{g}(\theta',\omega') - \overline{g}(\theta',\omega)\|_2\right) \\
&\leq 2R_\omega \left(\mathbb{E}\|g(x,\omega) - g(x,\omega')\|_2 + \mathbb{E}\left\|\mathbb{E}_{x\sim\theta'}[g(x,\omega')] - \mathbb{E}_{x\sim\theta'}[g(x,\omega)]\right\|_2\right) \\
&= 2R_\omega \left(\mathbb{E}\|(\gamma\phi(s') - \phi(s))^\top(\omega - \omega')\|_2 + \mathbb{E}\left\|\mathbb{E}_{x\sim\theta'}\left[(\gamma\phi(s') - \phi(s))^\top\right](\omega' - \omega)\right\|_2\right) \\
&\leq 2R_\omega \left((1+\gamma)\mathbb{E}\|\omega - \omega'\|_2 + (1+\gamma)\mathbb{E}\|\omega - \omega'\|_2\right) \\
&= 4R_\omega(1+\gamma)\mathbb{E}\|\omega - \omega'\|_2.
\end{aligned}
$$

We bound the second term in (95) as

$$
\begin{aligned}
&\mathbb{E}|\langle \omega - \omega_{\theta'}^*, g(x,\omega') - \overline{g}(\theta',\omega')\rangle - \langle \omega' - \omega_{\theta'}^*, g(x,\omega') - \overline{g}(\theta',\omega')\rangle| \\
&= \mathbb{E}|\langle \omega - \omega', g(x,\omega') - \overline{g}(\theta',\omega')\rangle| \\
&\leq 2C_\delta \mathbb{E}\|\omega - \omega'\|_2.
\end{aligned}
$$

Collecting the upper bounds of the two terms in (95) yields

$$
\mathbb{E}[I_2] \leq (4(1+\gamma)R_\omega + 2C_\delta)\mathbb{E}\|\omega - \omega'\|_2.
$$

We first bound $I_3$ as

$$
\begin{aligned}
\mathbb{E}[I_3|\theta',\omega',s_{t-m+1}] &= \mathbb{E}\left[\Delta_1(x,\theta',\omega') - \Delta_1(\widetilde{x},\theta',\omega')|\theta',\omega',s_{t-m+1}\right] \\
&\leq |\mathbb{E}\left[\Delta_1(x,\theta',\omega')|\theta',\omega',s_{t-m+1}\right] - \mathbb{E}\left[\Delta_1(\widetilde{x},\theta',\omega')|\theta',\omega',s_{t-m+1}\right]| \\
&\leq \sup_x |\Delta_1(x,\theta',\omega')|\, \|\mathbb{P}(x \in \cdot|\theta',\omega',s_{t-m+1}) - \mathbb{P}(\widetilde{x} \in \cdot|\theta',\omega',s_{t-m+1})\|_{TV} \\
&\leq 8R_\omega C_\delta d_{TV}\left(\mathbb{P}(x \in \cdot|\theta',s_{t-m+1}), \mathbb{P}(\widetilde{x} \in \cdot|\theta',s_{t-m+1})\right), \quad (96)
\end{aligned}
$$

where the second last inequality follows the definition of TV norm, and the last inequality follows the fact that

$$|\Delta_1(x, \theta', \omega')| \leq \|\omega' - \omega_{\theta'}^*\|_2 \|g(x, \omega') - \overline{g}(\theta', \omega')\|_2 \leq 4R_\omega C_\delta.$$

By following (22) in Lemma 2, we have

$$d_{TV}\left(\mathbb{P}(x \in \cdot|\theta', s_{t-m+1}), \mathbb{P}(\widetilde{x} \in \cdot|\theta', s_{t-m+1})\right) \leq \frac{1}{2}|\mathcal{A}|L_\pi \sum_{i=\tau_k}^{d_m} \mathbb{E}\left[\|\theta_{k-i} - \theta_{k-d_m}\|_2 \,|\, \theta', s_{t-m+1}\right].$$

Substituting the last inequality into (96), then taking total expectation on both sides yield

$$\mathbb{E}[I_3] \leq 4R_\omega C_\delta |\mathcal{A}|L_\pi \sum_{i=\tau_k}^{d_m} \mathbb{E}\|\theta_{k-i} - \theta_{k-d_m}\|_2.$$

Next we bound $I_4$. Define $\overline{x} := (\overline{s}, \overline{a}, \overline{s}')$ where $\overline{s} \sim \mu_{\theta'}, \overline{a} \sim \pi_{\theta'}$ and $\overline{s}' \sim \widetilde{\mathcal{P}}$. It is immediate that

$$\mathbb{E}[\Delta_1(\overline{x}, \theta', \omega')|\theta', \omega', s_{t-m+1}] = \langle \omega' - \omega_{\theta'}^*, \mathbb{E}[g(\overline{x}, \omega')|\theta', \omega', s_{t-m+1}] - \overline{g}(\theta', \omega')\rangle$$
$$= \langle \omega' - \omega_{\theta'}^*, \overline{g}(\theta', \omega') - \overline{g}(\theta', \omega')\rangle = 0. \tag{97}$$

Then we have

$$\mathbb{E}[I_4|\theta', \omega', s_{t-m+1}] = \mathbb{E}\left[\Delta_1(\widetilde{x}, \theta', \omega') - \Delta_1(\overline{x}, \theta', \omega')|\theta', \omega', s_{t-m+1}\right]$$
$$\leq |\mathbb{E}\left[\Delta_1(\widetilde{x}, \theta', \omega')|\theta', \omega', s_{t-m+1}\right] - \mathbb{E}\left[\Delta_1(\overline{x}, \theta', \omega')|\theta', \omega', s_{t-m+1}\right]|$$
$$\leq \sup_x |\Delta_1(x, \theta', \omega')| \|\mathbb{P}(\widetilde{x} \in \cdot|\theta', s_{t-m+1}) - \mathbb{P}(\overline{x} \in \cdot|\theta', s_{t-m+1})\|_{TV}$$
$$\leq 8R_\omega C_\delta d_{TV}\left(\mathbb{P}(\widetilde{x} \in \cdot|\theta', s_{t-m+1}), \mathbb{P}(\overline{x} \in \cdot|\theta', s_{t-m+1})\right)$$
$$= 8R_\omega C_\delta d_{TV}\left(\mathbb{P}(\widetilde{x} \in \cdot|\theta', s_{t-m+1}), \mu_{\theta'} \otimes \pi_{\theta'} \otimes \widetilde{\mathcal{P}}\right), \tag{98}$$

where the second inequality follows the definition of TV norm, and the third inequality follows (97).

The auxiliary Markov chain with policy $\pi_{\theta'}$ starts from initial state $s_{t-m+1}$, and $\widetilde{s}_t$ is the $(m-1)$th state on the chain. Following Lemma 1, we have:

$$d_{TV}\left(\mathbb{P}(\widetilde{x} \in \cdot|\theta', s_{t-m+1}), \mu_{\theta'} \otimes \pi_{\theta'} \otimes \widetilde{\mathcal{P}}\right)$$
$$= d_{TV}\left(\mathbb{P}\left((\widetilde{s}_t, \widetilde{a}_t, \widetilde{s}_{t+1}) \in \cdot|\theta', s_{t-m+1}\right), \mu_{\theta'} \otimes \pi_{\theta'} \otimes \widetilde{\mathcal{P}}\right) \leq \kappa \rho^{m-1}.$$

Substituting the last inequality into (98) and taking total expectation on both sides yield

$$\mathbb{E}[I_4] \leq 8R_\omega C_\delta \kappa \rho^{m-1}.$$

Taking total expectation on (93) and collecting bounds of $I_1, I_2, I_3, I_4$ yield

$$\mathbb{E}\left[\Delta_1(x, \theta, \omega)\right] \leq C_4 \,\mathbb{E}\|\theta_k - \theta_{k-d_m}\|_2 + C_5 \sum_{i=\tau_k}^{d_m} \mathbb{E}\|\theta_{k-i} - \theta_{k-d_m}\|_2$$
$$+ C_6 \,\mathbb{E}\|\omega_k - \omega_{k-d_m}\|_2 + C_7 \kappa \rho^{m-1},$$

where $C_4 := 2C_\delta L_\omega + 4R_\omega C_\delta |\mathcal{A}|L_\pi(1 + \log_\rho \kappa^{-1} + (1-\rho)^{-1})$, $C_5 := 4R_\omega C_\delta |\mathcal{A}|L_\pi$, $C_6 := 4(1+\gamma)R_\omega + 2C_\delta$ and $C_7 := 8R_\omega C_\delta$. □

## C.2 SUPPORTING LEMMAS FOR THEOREM 4

**Lemma 5.** *For any $m \geq 1$ and $k \geq (K_0 + 1)m + K_0 + 1$, we have*

$$\mathbb{E}\left\langle \nabla J(\theta_k), \left(\hat{\delta}(x_{(k)}, \omega_k^*) - \delta(x_{(k)}, \theta_k)\right) \psi_{\theta_{k-\tau_k}}(s_{(k)}, a_{(k)})\right\rangle \geq -D_2 \,\mathbb{E}\|\theta_{k-\tau_k} - \theta_{k-d_m}\|_2$$

$$- D_3 \,\mathbb{E}\|\theta_k - \theta_{k-d_m}\|_2 - D_4 \sum_{i=\tau_k}^{d_m} \mathbb{E}\|\theta_{k-i} - \theta_{k-d_m}\|_2 - D_5 \kappa \rho^{m-1} - 2C_\psi L_V \epsilon_{\text{fa}} - 2C_\psi \epsilon_{\text{sp}} \,\mathbb{E}\|\nabla J(\theta)\|_2,$$

*where $D_2 := 2L_V L_\psi C_\delta$, $D_3 := (2C_\delta C_\psi L_J + L_V C_\psi(L_\omega + L_V)(1+\gamma) + 2C_\psi L_J \epsilon_{\text{app}})$, $D_4 := 2L_V C_\psi C_\delta |\mathcal{A}|L_\pi$ and $D_5 := 4L_V C_\psi C_\delta$.*

*Proof.* For the worker that contributes to the $k$th update, we construct its Markov chain:

$$s_{t-m} \xrightarrow{\theta_{k-d_m}} a_{t-m} \xrightarrow{\widetilde{\mathcal{P}}} s_{t-m+1} \xrightarrow{\theta_{k-d_{m-1}}} a_{t-m+1} \cdots s_{t-1} \xrightarrow{\theta_{k-d_1}} a_{t-1} \xrightarrow{\widetilde{\mathcal{P}}} s_t \xrightarrow{\theta_{k-d_0}} a_t \xrightarrow{\widetilde{\mathcal{P}}} s_{t+1},$$

where $(s_t, a_t, s_{t+1}) = (s_{(k)}, a_{(k)}, s'_{(k)})$, and $\{d_j\}_{j=0}^m$ is some increasing sequence with $d_0 := \tau_k$. By (92) in Lemma 4, we have $d_m \leq (K_0 + 1)m + K_0$.

Given $(s_{t-m}, a_{t-m}, s_{t-m+1})$ and $\theta_{k-d_m}$, we construct an auxiliary Markov chain:

$$s_{t-m} \xrightarrow{\theta_{k-d_m}} a_{t-m} \xrightarrow{\widetilde{\mathcal{P}}} s_{t-m+1} \xrightarrow{\theta_{k-d_m}} \widetilde{a}_{t-m+1} \cdots \widetilde{s}_{t-1} \xrightarrow{\theta_{k-d_m}} \widetilde{a}_{t-1} \xrightarrow{\widetilde{\mathcal{P}}} \widetilde{s}_t \xrightarrow{\theta_{k-d_m}} \widetilde{a}_t \xrightarrow{\widetilde{\mathcal{P}}} \widetilde{s}_{t+1}.$$

First we have

$$\left\langle \nabla J(\theta_k), \left( \hat{\delta}(x_{(k)}, \omega_k^*) - \delta(x_{(k)}, \theta_k) \right) \psi_{\theta_{k-\tau_k}}(s_{(k)}, a_{(k)}) \right\rangle$$
$$= \left\langle \nabla J(\theta_k), \left( \hat{\delta}(x_{(k)}, \omega_k^*) - \delta(x_{(k)}, \theta_k) \right) \left( \psi_{\theta_{k-\tau_k}}(s_{(k)}, a_{(k)}) - \psi_{\theta_{k-d_m}}(s_{(k)}, a_{(k)}) \right) \right\rangle$$
$$+ \left\langle \nabla J(\theta_k), \left( \hat{\delta}(x_{(k)}, \omega_k^*) - \delta(x_{(k)}, \theta_k) \right) \psi_{\theta_{k-d_m}}(s_{(k)}, a_{(k)}) \right\rangle. \tag{99}$$

We first bound the fist term in (99) as

$$\left\langle \nabla J(\theta_k), \left( \hat{\delta}(x_{(k)}, \omega_k^*) - \delta(x_{(k)}, \theta_k) \right) \left( \psi_{\theta_{k-\tau_k}}(s_{(k)}, a_{(k)}) - \psi_{\theta_{k-d_m}}(s_{(k)}, a_{(k)}) \right) \right\rangle$$
$$\geq -\|J(\theta_k)\|_2 |\hat{\delta}(x_{(k)}, \omega_k^*) - \delta(x_{(k)}, \theta_k)| \|\psi_{\theta_{k-\tau_k}}(s_{(k)}, a_{(k)}) - \psi_{\theta_{k-d_m}}(s_{(k)}, a_{(k)})\|_2$$
$$\geq -\|J(\theta_k)\|_2 \left( |\hat{\delta}(x_{(k)}, \omega_k^*)| + |\delta(x_{(k)}, \theta_k)| \right) \|\psi_{\theta_{k-\tau_k}}(s_{(k)}, a_{(k)}) - \psi_{\theta_{k-d_m}}(s_{(k)}, a_{(k)})\|_2$$
$$\geq -L_V \left( |\hat{\delta}(x_{(k)}, \omega_k^*)| + |\delta(x_{(k)}, \theta_k)| \right) \|\psi_{\theta_{k-\tau_k}}(s_{(k)}, a_{(k)}) - \psi_{\theta_{k-d_m}}(s_{(k)}, a_{(k)})\|_2$$
$$\geq -2L_V C_\delta \|\psi_{\theta_{k-\tau_k}}(s_{(k)}, a_{(k)}) - \psi_{\theta_{k-d_m}}(s_{(k)}, a_{(k)})\|_2$$
$$\geq -2L_V L_\psi C_\delta \|\theta_{k-\tau_k} - \theta_{k-d_m}\|_2, \tag{100}$$

where the last inequality follows Assumption 3 and second last inequality follows

$$|\hat{\delta}(x, \omega_\theta^*)| \leq |r(x)| + \gamma \|\phi(s')\|_2 \|\omega_\theta^*\|_2 + \|\phi(s)\|_2 \|\omega_\theta^*\|_2 \leq r_{\max} + (1 + \gamma) R_\omega \leq C_\delta,$$
$$|\delta(x, \theta)| \leq |r(x)| + \gamma |V_{\pi_\theta}(s')| + |V_{\pi_\theta}(s)| \leq r_{\max} + (1 + \gamma) \frac{r_{\max}}{1 - \gamma} \leq C_\delta.$$

Substituting (100) into (99) gives

$$\left\langle \nabla J(\theta_k), \left( \hat{\delta}(x_{(k)}, \omega_k^*) - \delta(x_{(k)}, \theta_k) \right) \psi_{\theta_{k-\tau_k}}(s_{(k)}, a_{(k)}) \right\rangle$$
$$\geq -2L_V L_\psi C_\delta \|\theta_{k-\tau_k} - \theta_{k-d_m}\|_2 + \left\langle \nabla J(\theta_k), \left( \hat{\delta}(x_{(k)}, \omega_k^*) - \delta(x_{(k)}, \theta_k) \right) \psi_{\theta_{k-d_m}}(s_{(k)}, a_{(k)}) \right\rangle. \tag{101}$$

Then we start to bound the second term in (101). For brevity, we define

$$\Delta_2(x, \theta) := \left\langle \nabla J(\theta), \left( \hat{\delta}(x, \omega_\theta^*) - \delta(x, \theta) \right) \psi_{\theta_{k-d_m}}(s, a) \right\rangle.$$

In the following proof, we use $\theta$, $\theta'$, $\omega_\theta^*$, $\omega_{\theta'}^*$, $x$ and $\widetilde{x}$ as shorthand notations for $\theta_k$, $\theta_{k-d_m}$, $\omega_k^*$, $\omega_{k-d_m}^*$, $x_t$ and $\widetilde{x}_t$ respectively. We also define $\overline{x} := (\overline{s}, \overline{a}, \overline{s}')$, where $\overline{s} \sim \mu_{\theta'}$, $\overline{a} \sim \pi_{\theta'}$ and $\overline{s}' \sim \widetilde{\mathcal{P}}$.

We decompose the second term in (101) as

$$\Delta_2(x, \theta) = \underbrace{\Delta_2(x, \theta) - \Delta_2(x, \theta')}_{I_1} + \underbrace{\Delta_2(x, \theta') - \Delta_2(\widetilde{x}, \theta')}_{I_2} + \underbrace{\Delta_2(\widetilde{x}, \theta') - \Delta_2(\overline{x}, \theta')}_{I_3} + \underbrace{\Delta_2(\overline{x}, \theta')}_{I_4}.$$

We bound the term $I_1$ as

$$I_1 = \left\langle \nabla J(\theta), \left( \hat{\delta}(x, \omega_\theta^*) - \delta(x, \theta) \right) \psi_{\theta'}(s, a) \right\rangle - \left\langle \nabla J(\theta'), \left( \hat{\delta}(x, \omega_{\theta'}^*) - \delta(x, \theta') \right) \psi_{\theta'}(s, a) \right\rangle$$
$$= \left\langle \nabla J(\theta), \left( \hat{\delta}(x, \omega_\theta^*) - \delta(x, \theta) \right) \psi_{\theta'}(s, a) \right\rangle - \left\langle \nabla J(\theta'), \left( \hat{\delta}(x, \omega_\theta^*) - \delta(x, \theta) \right) \psi_{\theta'}(s, a) \right\rangle$$
$$+ \left\langle \nabla J(\theta'), \left( \hat{\delta}(x, \omega_\theta^*) - \delta(x, \theta) \right) \psi_{\theta'}(s, a) \right\rangle - \left\langle \nabla J(\theta'), \left( \hat{\delta}(x, \omega_{\theta'}^*) - \delta(x, \theta') \right) \psi_{\theta'}(s, a) \right\rangle.$$

For the first term in $I_1$, we have

$$\left\langle \nabla J(\theta), \left(\hat{\delta}(x, \omega_\theta^*) - \delta(x, \theta)\right) \psi_{\theta'}(s, a) \right\rangle - \left\langle \nabla J(\theta'), \left(\hat{\delta}(x, \omega_\theta^*) - \delta(x, \theta)\right) \psi_{\theta'}(s, a) \right\rangle$$
$$= \left\langle \nabla J(\theta) - \nabla J(\theta'), \left(\hat{\delta}(x, \omega_\theta^*) - \delta(x, \theta)\right) \psi_{\theta'}(s, a) \right\rangle$$
$$\geq -\|\nabla J(\theta) - \nabla J(\theta')\|_2 \|\hat{\delta}(x, \omega_\theta^*) - \delta(x, \theta)\|_2 \|\psi_{\theta'}(s, a)\|_2$$
$$\geq -2C_\delta C_\psi \|\nabla J(\theta) - \nabla J(\theta')\|_2$$
$$\geq -2C_\delta C_\psi L_J \|\theta - \theta'\|_2,$$

where the last inequality is due to the $L_J$-Lipschitz of policy gradient shown in Proposition 1.

For the second term in $I_1$, we have

$$\left\langle \nabla J(\theta'), \left(\hat{\delta}(x, \omega_\theta^*) - \delta(x, \theta)\right) \psi_{\theta'}(s, a) \right\rangle - \left\langle \nabla J(\theta'), \left(\hat{\delta}(x, \omega_{\theta'}^*) - \delta(x, \theta')\right) \psi_{\theta'}(s, a) \right\rangle$$
$$= \left\langle \nabla J(\theta'), \left(\hat{\delta}(x, \omega_\theta^*) - \hat{\delta}(x, \omega_{\theta'}^*) + \delta(x, \theta') - \delta(x, \theta)\right) \psi_{\theta'}(s, a) \right\rangle$$
$$\geq -L_V C_\psi \left|\hat{\delta}(x, \omega_\theta^*) - \hat{\delta}(x, \omega_{\theta'}^*) + \delta(x, \theta') - \delta(x, \theta)\right|$$
$$\geq -L_V C_\psi \left|\gamma \phi(s')^\top (\omega_\theta^* - \omega_{\theta'}^*) + \phi(s)^\top (\omega_{\theta'}^* - \omega_\theta^*) + \gamma V_{\pi_{\theta'}}(s') - \gamma V_{\pi_\theta}(s') + V_{\pi_\theta}(s) - V_{\pi_{\theta'}}(s)\right|$$
$$\geq -L_V C_\psi \left(\gamma \|\omega_\theta^* - \omega_{\theta'}^*\|_2 + \|\omega_{\theta'}^* - \omega_\theta^*\|_2 + \gamma|V_{\pi_{\theta'}}(s') - V_{\pi_\theta}(s')| + |V_{\pi_\theta}(s) - V_{\pi_{\theta'}}(s)|\right)$$
$$\geq -L_V C_\psi \left(\gamma L_\omega \|\theta - \theta'\|_2 + L_\omega \|\theta - \theta'\|_2 + \gamma L_V \|\theta - \theta'\|_2 + L_V \|\theta - \theta'\|_2\right)$$
$$= -L_V C_\psi (L_\omega + L_V)(1 + \gamma)\|\theta - \theta'\|_2,$$

where the last inequality is due to the $L_\omega$-Lipschitz continuity of $\omega_\theta^*$ shown in Proposition 2 and $L_V$-Lipschitz continuity of $V_{\pi_\theta}(s)$ shown in Lemma 3. Collecting the upper bounds of $I_1$ yields

$$I_1 \geq -\left(2C_\delta C_\psi L_J + L_V C_\psi (L_\omega + L_V)(1 + \gamma)\right)\|\theta - \theta'\|_2.$$

First we bound $I_2$ as

$$\mathbb{E}[I_2|\theta', s_{t-m+1}] = \mathbb{E}\left[\Delta_2(x, \theta') - \Delta_2(\tilde{x}, \theta')|\theta', s_{t-m+1}\right]$$
$$\geq -\left|\mathbb{E}\left[\Delta_2(x, \theta')|\theta', s_{t-m+1}\right] - \mathbb{E}\left[\Delta_2(\tilde{x}, \theta')|\theta', s_{t-m+1}\right]\right|$$
$$\geq -\sup_x |\Delta_2(x, \theta')| \|\mathbb{P}(x \in \cdot|\theta', s_{t-m+1}) - \mathbb{P}(\tilde{x} \in \cdot|\theta', s_{t-m+1})\|_{TV}$$
$$\geq -4L_V C_\psi C_\delta d_{TV}\left(\mathbb{P}(x \in \cdot|\theta', s_{t-m+1}), \mathbb{P}(\tilde{x} \in \cdot|\theta', s_{t-m+1})\right)$$
$$\geq -2L_V C_\psi C_\delta |\mathcal{A}| L_\pi \sum_{i=\tau_k}^{d_m} \mathbb{E}\left[\|\theta_{k-i} - \theta_{k-d_m}\|_2|\theta', s_{t-m+1}\right], \qquad (102)$$

where the second inequality is due to the definition of TV norm, the last inequality follows (22) in Lemma 2, and the second last inequality follows the fact that

$$|\Delta_2(x, \theta')| \leq \|\nabla J(\theta')\|_2 \|\hat{\delta}(x, \omega_{\theta'}^*) - \delta(x, \theta')\| \|\psi_{\theta'}(s, a)\|_2 \leq 2L_V C_\delta C_\psi. \qquad (103)$$

Taking total expectation on both sides of (102) yields

$$\mathbb{E}[I_2] \geq -2L_V C_\psi C_\delta |\mathcal{A}| L_\pi \sum_{i=\tau_k}^{d_m} \mathbb{E}\|\theta_{k-i} - \theta_{k-d_m}\|_2.$$

Next we bound $I_3$ as

$$\mathbb{E}[I_3|\theta', s_{t-m+1}] = \mathbb{E}\left[\Delta_2(\tilde{x}, \theta') - \Delta_2(\bar{x}, \theta')|\theta', s_{t-m+1}\right]$$
$$\geq -\left|\mathbb{E}\left[\Delta_2(\tilde{x}, \theta')|\theta', s_{t-m+1}\right] - \mathbb{E}\left[\Delta_2(\bar{x}, \theta')|\theta', s_{t-m+1}\right]\right|$$
$$\geq -\sup_x |\Delta_2(x, \theta')| \|\mathbb{P}(\tilde{x} \in \cdot|\theta', s_{t-m+1}) - \mathbb{P}(\bar{x} \in \cdot|\theta', s_{t-m+1})\|_{TV}$$
$$\geq -4L_V C_\psi C_\delta d_{TV}\left(\mathbb{P}(\tilde{x} \in \cdot|\theta', s_{t-m+1}), \mu_{\theta'} \otimes \pi_{\theta'} \otimes \widetilde{\mathcal{P}}\right), \qquad (104)$$

where the second inequality is due to the definition of TV norm, and the last inequality follows (103).

The auxiliary Markov chain with policy $\pi_{\theta'}$ starts from initial state $s_{t-m+1}$, and $\widetilde{s}_t$ is the $(m-1)$th state on the chain. Following Lemma 1, we have:

$$d_{TV}\left(\mathbb{P}(\widetilde{x}\in\cdot|\theta',s_{t-m+1}),\mu_{\theta'}\otimes\pi_{\theta'}\otimes\widetilde{\mathcal{P}}\right)$$
$$=d_{TV}\left(\mathbb{P}\left((\widetilde{s}_t,\widetilde{a}_t,\widetilde{s}_{t+1})\in\cdot|\theta',s_{t-m+1}\right),\mu_{\theta'}\otimes\pi_{\theta'}\otimes\widetilde{\mathcal{P}}\right)\leq\kappa\rho^{m-1}.$$

Substituting the last inequality into (104) and taking total expectation on both sides yield

$$\mathbb{E}[I_3]\geq-4L_VC_\psi C_\delta\kappa\rho^{m-1}$$

We bound $I_4$ as

$$\mathbb{E}[I_4|\theta']=\mathbb{E}\left[\left\langle\nabla J(\theta'),\left(\hat{\delta}(\overline{x},\omega_{\theta'}^*)-\delta(\overline{x},\theta')\right)\psi_{\theta'}(s,a)\right\rangle\Big|\theta'\right]$$
$$\geq-C_\psi\|\nabla J(\theta')\|_2\,\mathbb{E}\left[\left|\hat{\delta}(\overline{x},\omega_{\theta'}^*)-\delta(\overline{x},\theta')\right|\Big|\theta'\right]$$
$$=-C_\psi\|\nabla J(\theta')\|_2\,\mathbb{E}\left[\left|\gamma\left(\phi(\overline{s}')^\top\omega_{\theta'}^*-V_{\pi_{\theta'}}(\overline{s}')\right)+V_{\pi_{\theta'}}(\overline{s})-\phi(\overline{s})^\top\omega_{\theta'}^*\right|\Big|\theta'\right]$$
$$\geq-C_\psi\|\nabla J(\theta')\|_2\left(\gamma\,\mathbb{E}\left[|\phi(\overline{s}')^\top\omega_{\theta'}^*-V_{\pi_{\theta'}}(\overline{s}')|\Big|\theta'\right]+\mathbb{E}\left[|V_{\pi_{\theta'}}(\overline{s})-\phi(\overline{s})^\top\omega_{\theta'}^*|\Big|\theta'\right]\right)$$
$$\geq-C_\psi\|\nabla J(\theta')\|_2\left(\gamma\sqrt{\mathbb{E}\left[|\phi(\overline{s}')^\top\omega_{\theta'}^*-V_{\pi_{\theta'}}(\overline{s}')|^2\Big|\theta'\right]}+\sqrt{\mathbb{E}\left[|V_{\pi_{\theta'}}(\overline{s})-\phi(\overline{s})^\top\omega_{\theta'}^*|^2\Big|\theta'\right]}\right)$$
$$=-C_\psi\|\nabla J(\theta')\|_2\left(\gamma\sqrt{\mathop{\mathbb{E}}_{\overline{s}'\sim\mu_{\theta'}}|\phi(\overline{s}')^\top\omega_{\theta'}^*-V_{\pi_{\theta'}}(\overline{s}')|^2}+\sqrt{\mathop{\mathbb{E}}_{\overline{s}\sim\mu_{\theta'}}|V_{\pi_{\theta'}}(\overline{s})-\phi(\overline{s})^\top\omega_{\theta'}^*|^2}\right)$$
$$\geq-2C_\psi\|\nabla J(\theta')\|_2\epsilon_{\mathrm{app}},$$

where the second last inequality follows Jensen's inequality.

The last inequality further implies

$$\mathbb{E}[I_4]\geq-2C_\psi\,\mathbb{E}\,\|\nabla J(\theta')-\nabla J(\theta)+\nabla J(\theta)\|_2\epsilon_{\mathrm{app}}$$
$$\geq-2C_\psi\epsilon_{\mathrm{app}}\,\mathbb{E}\,\|\nabla J(\theta')-\nabla J(\theta)\|_2-2C_\psi\epsilon_{\mathrm{app}}\,\mathbb{E}\,\|\nabla J(\theta)\|_2$$
$$\geq-2C_\psi\epsilon_{\mathrm{app}}\,\mathbb{E}\,\|\nabla J(\theta')-\nabla J(\theta)\|_2-2C_\psi\epsilon_{\mathrm{fa}}\,\mathbb{E}\,\|\nabla J(\theta)\|_2-2C_\psi L_V\epsilon_{\mathrm{sp}}$$
$$\geq-2C_\psi L_J\epsilon_{\mathrm{app}}\,\mathbb{E}\,\|\theta-\theta'\|_2-2C_\psi\epsilon_{\mathrm{fa}}\,\mathbb{E}\,\|\nabla J(\theta)\|_2-2C_\psi L_V\epsilon_{\mathrm{sp}},$$

where the last inequality follows Proposition 1.

Taking total expectation on both sides of (101), and collecting lower bounds of $I_1$, $I_2$, $I_3$ and $I_4$ yield

$$\mathbb{E}\left\langle\nabla J(\theta_k),\left(\hat{\delta}(x_{(k)},\omega_k^*)-\delta(x_{(k)},\theta_k)\right)\psi_{\theta_{k-\tau_k}}(s_{(k)},a_{(k)})\right\rangle$$
$$\geq-D_2\,\mathbb{E}\,\|\theta_{k-\tau_k}-\theta_{k-d_m}\|_2-D_3\,\mathbb{E}\,\|\theta_k-\theta_{k-d_m}\|_2-D_4\sum_{i=\tau_k}^{d_m}\mathbb{E}\,\|\theta_{k-i}-\theta_{k-d_m}\|_2$$
$$-D_5\kappa\rho^{m-1}-2C_\psi L_V\epsilon_{\mathrm{fa}}-2C_\psi\epsilon_{\mathrm{sp}}\,\mathbb{E}\,\|\nabla J(\theta_k)\|_2,$$

where $D_2:=2L_VL_\psi C_\delta$, $D_3:=(2C_\delta C_\psi L_J+L_VC_\psi(L_\omega+L_V)(1+\gamma)+2C_\psi L_J\epsilon_{\mathrm{app}})$, $D_4:=2L_VC_\psi C_\delta|\mathcal{A}|L_\pi$ and $D_5:=4L_VC_\psi C_\delta$. $\qquad\square$

**Lemma 6.** *For any $m\geq1$ and $k\geq(K_0+1)m+K_0+1$, we have*

$$\mathbb{E}\left\langle\nabla J(\theta_k),\delta(x_{(k)},\theta_k)\psi_{\theta_{k-\tau_k}}(s_{(k)},a_{(k)})-\nabla J(\theta_k)\right\rangle\geq-D_6\,\mathbb{E}\,\|\theta_{k-\tau_k}-\theta_{k-d_m}\|_2$$

$$-D_7\,\mathbb{E}\,\|\theta_k-\theta_{k-d_m}\|_2-D_8\sum_{i=\tau_k}^{d_m}\mathbb{E}\,\|\theta_{k-i}-\theta_{k-d_m}\|_2-D_9\kappa\rho^{m-1}-8C_\psi r_{\max}(1-\gamma)\,\mathbb{E}\,\|\nabla J(\theta_k)\|_2,$$

*where $D_6:=L_VC_\delta L_\psi$, $D_7:=C_pL_J+(1+\gamma)L_V^2C_\psi+2L_VL_J+8C_\psi r_{\max}L_J(1-\gamma)$, $D_8:=L_V(C_p+L_V)|\mathcal{A}|L_\pi$, $D_9:=2L_V(C_p+L_V)$.*

*Proof.* For the worker that contributes to the $k$th update, we construct its Markov chain:

$$s_{t-m}\xrightarrow{\theta_{k-d_m}}a_{t-m}\xrightarrow{\widetilde{\mathcal{P}}}s_{t-m+1}\xrightarrow{\theta_{k-d_{m-1}}}a_{t-m+1}\cdots s_{t-1}\xrightarrow{\theta_{k-d_1}}a_{t-1}\xrightarrow{\widetilde{\mathcal{P}}}s_t\xrightarrow{\theta_{k-d_0}}a_t\xrightarrow{\widetilde{\mathcal{P}}}s_{t+1},$$

where $(s_t, a_t, s_{t+1}) = (s_{(k)}, a_{(k)}, s'_{(k)})$, and $\{d_j\}_{j=0}^m$ is some increasing sequence with $d_0 := \tau_k$. By (92) in Lemma 4, we have $d_m \leq (K_0 + 1)m + K_0$.

Given $(s_{t-m}, a_{t-m}, s_{t-m+1})$ and $\theta_{k-d_m}$, we construct an auxiliary Markov chain:

$$s_{t-m} \xrightarrow{\theta_{k-d_m}} a_{t-m} \xrightarrow{\widetilde{\mathcal{P}}} s_{t-m+1} \xrightarrow{\theta_{k-d_m}} \widetilde{a}_{t-m+1} \cdots \widetilde{s}_{t-1} \xrightarrow{\theta_{k-d_m}} \widetilde{a}_{t-1} \xrightarrow{\widetilde{\mathcal{P}}} \widetilde{s}_t \xrightarrow{\theta_{k-d_m}} \widetilde{a}_t \xrightarrow{\widetilde{\mathcal{P}}} \widetilde{s}_{t+1}.$$

First we have

$$\left\langle \nabla J(\theta_k), \delta(x_{(k)}, \theta_k)\psi_{\theta_{k-\tau_k}}(s_{(k)}, a_{(k)}) - \nabla J(\theta_k) \right\rangle$$
$$= \left\langle \nabla J(\theta_k), \delta(x_{(k)}, \theta_k)\left(\psi_{\theta_{k-\tau_k}}(s_{(k)}, a_{(k)}) - \psi_{\theta_{k-d_m}}(s_{(k)}, a_{(k)})\right) \right\rangle$$
$$+ \left\langle \nabla J(\theta_k), \delta(x_{(k)}, \theta_k)\psi_{\theta_{k-d_m}}(s_{(k)}, a_{(k)}) - \nabla J(\theta_k) \right\rangle. \tag{105}$$

We bound the first term in (105) as

$$\left\langle \nabla J(\theta_k), \delta(x_{(k)}, \theta_k)\left(\psi_{\theta_{k-\tau_k}}(s_{(k)}, a_{(k)}) - \psi_{\theta_{k-d_m}}(s_{(k)}, a_{(k)})\right) \right\rangle$$
$$\geq -\|\nabla J(\theta_k)\|_2 \|\delta(x_{(k)}, \theta_k)\|_2 \|\psi_{\theta_{k-\tau_k}}(s_{(k)}, a_{(k)}) - \psi_{\theta_{k-d_m}}(s_{(k)}, a_{(k)})\|_2$$
$$\geq -L_V \|\delta(x_{(k)}, \theta_k)\|_2 \|\psi_{\theta_{k-\tau_k}}(s_{(k)}, a_{(k)}) - \psi_{\theta_{k-d_m}}(s_{(k)}, a_{(k)})\|_2$$
$$\geq -L_V C_\delta \|\psi_{\theta_{k-\tau_k}}(s_{(k)}, a_{(k)}) - \psi_{\theta_{k-d_m}}(s_{(k)}, a_{(k)})\|_2$$
$$\geq -L_V C_\delta L_\psi \|\theta_{k-\tau_k} - \theta_{k-d_m}\|_2, \tag{106}$$

where the last inequality follows Assumption 3, and the second last inequality follows the fact that

$$|\delta(x, \theta)| \leq |r(x)| + \gamma|V_{\pi_\theta}(s')| + |V_{\pi_\theta}(s)| \leq r_{\max} + (1 + \gamma)\frac{r_{\max}}{1 - \gamma} \leq C_\delta.$$

Substituting (106) into (105) gives

$$\left\langle \nabla J(\theta_k), \delta(x_{(k)}, \theta_k)\psi_{\theta_{k-\tau_k}}(s_{(k)}, a_{(k)}) - \nabla J(\theta_k) \right\rangle$$
$$\geq -L_V C_\delta L_\psi \|\theta_{k-\tau_k} - \theta_{k-d_m}\|_2 + \left\langle \nabla J(\theta_k), \delta(x_{(k)}, \theta_k)\psi_{\theta_{k-d_m}}(s_{(k)}, a_{(k)}) - \nabla J(\theta_k) \right\rangle. \tag{107}$$

Then we start to bound the second term in (107). For brevity, we define

$$\Delta_3(x, \theta) := \left\langle \nabla J(\theta), \delta(x, \theta)\psi_{\theta_{k-d_m}}(s, a) - \nabla J(\theta) \right\rangle.$$

Throughout the following proof, we use $\theta$, $\theta'$, $x$ and $\widetilde{x}$ as shorthand notations of $\theta_k$, $\theta_{k-d_m}$, $x_t$ and $\widetilde{x}_t$ respectively.

We decompose $\Delta_3(x, \theta)$ as

$$\Delta_3(x, \theta) = \underbrace{\Delta_3(x, \theta) - \Delta_3(x, \theta')}_{I_1} + \underbrace{\Delta_3(x, \theta') - \Delta_3(\widetilde{x}, \theta')}_{I_2} + \underbrace{\Delta_3(\widetilde{x}, \theta')}_{I_3}.$$

We first bound $I_1$ as

$$|I_1| = |\Delta_3(x, \theta) - \Delta_3(x, \theta')|$$
$$= \left| \langle \nabla J(\theta), \delta(x, \theta)\psi_{\theta'}(s, a)\rangle - \|\nabla J(\theta)\|_2^2 - \langle \nabla J(\theta'), \delta(x, \theta')\psi_{\theta'}(s, a)\rangle + \|\nabla J(\theta')\|_2^2 \right|$$
$$\leq |\langle \nabla J(\theta), \delta(x, \theta)\psi_{\theta'}(s, a)\rangle - \langle \nabla J(\theta'), \delta(x, \theta')\psi_{\theta'}(s, a)\rangle| + \left| \|\nabla J(\theta')\|_2^2 - \|\nabla J(\theta)\|_2^2 \right|$$
$$\leq |\langle \nabla J(\theta), \delta(x, \theta)\psi_{\theta'}(s, a)\rangle - \langle \nabla J(\theta'), \delta(x, \theta')\psi_{\theta'}(s, a)\rangle| + \|\nabla J(\theta') + \nabla J(\theta)\|_2 \|\nabla J(\theta') - \nabla J(\theta)\|_2$$
$$\leq |\langle \nabla J(\theta), \delta(x, \theta)\psi_{\theta'}(s, a)\rangle - \langle \nabla J(\theta'), \delta(x, \theta')\psi_{\theta'}(s, a)\rangle| + 2L_V L_J \|\theta - \theta'\|_2, \tag{108}$$

where the last equality is due to $L_V$-Lipschitz of value function and $L_J$-Lipschitz of policy gradient. We bound the first term in (108) as

$$
\begin{aligned}
& \left|\langle \nabla J(\theta), \delta(x,\theta)\psi_{\theta'}(s,a)\rangle - \langle \nabla J(\theta'), \delta(x,\theta')\psi_{\theta'}(s,a)\rangle\right| \\
& \leq \left|\langle \nabla J(\theta), \delta(x,\theta)\psi_{\theta'}(s,a)\rangle - \langle \nabla J(\theta), \delta(x,\theta')\psi_{\theta'}(s,a)\rangle\right| \\
& \quad + \left|\langle \nabla J(\theta), \delta(x,\theta')\psi_{\theta'}(s,a)\rangle - \langle \nabla J(\theta'), \delta(x,\theta')\psi_{\theta'}(s,a)\rangle\right| \\
& = \left|\langle \nabla J(\theta), \left(\delta(x,\theta) - \delta(x,\theta')\right)\psi_{\theta'}(s,a)\rangle\right| + \left|\langle \nabla J(\theta) - \nabla J(\theta'), \delta(x,\theta')\psi_{\theta'}(s,a)\rangle\right| \\
& \leq L_V C_\psi \left|\delta(x,\theta) - \delta(x,\theta')\right| + C_p \|\nabla J(\theta) - \nabla J(\theta')\|_2 \\
& = L_V C_\psi \left|\gamma(V_{\pi_\theta}(s') - V_{\pi_{\theta'}}(s')) + V_{\pi_{\theta'}}(s) - V_{\pi_\theta}(s)\right| + C_p \|\nabla J(\theta) - \nabla J(\theta')\|_2 \\
& \leq L_V C_\psi \left(\gamma \left|V_{\pi_\theta}(s') - V_{\pi_{\theta'}}(s')\right| + \left|V_{\pi_{\theta'}}(s) - V_{\pi_\theta}(s)\right|\right) + C_p \|\nabla J(\theta) - \nabla J(\theta')\|_2 \\
& \leq L_V C_\psi \left(\gamma L_V \|\theta - \theta'\|_2 + L_V \|\theta' - \theta\|\right) + C_p L_J \|\theta - \theta'\|_2 \\
& = \left(C_p L_J + (1+\gamma)L_V^2 C_\psi\right) \|\theta - \theta'\|_2.
\end{aligned}
$$

Substituting the above inequality into (108) gives the lower bound of $I_1$:

$$
I_1 \geq - \left(C_p L_J + (1+\gamma)L_V^2 C_\psi + 2 L_V L_J\right) \|\theta - \theta'\|_2.
$$

First we bound $I_2$ as

$$
\begin{aligned}
\mathbb{E}[I_2|\theta', s_{t-m+1}] = \mathbb{E}\left[\Delta_3(x,\theta') - \Delta_3(\widetilde{x},\theta')|\theta', s_{t-m+1}\right] \\
\geq - \left|\mathbb{E}\left[\Delta_3(x,\theta')|\theta', s_{t-m+1}\right] - \mathbb{E}\left[\Delta_3(\widetilde{x},\theta')|\theta', s_{t-m+1}\right]\right| \\
\geq - \sup_x \left|\Delta_3(x,\theta')\right| \|\mathbb{P}(x \in \cdot|\theta', s_{t-m+1}) - \mathbb{P}(\widetilde{x} \in \cdot|\theta', s_{t-m+1})\|_{TV} \\
\geq -2 L_V (C_p + L_V) d_{TV}\left(\mathbb{P}(x \in \cdot|\theta', s_{t-m+1}), \mathbb{P}(\widetilde{x} \in \cdot|\theta', s_{t-m+1})\right) \\
\geq -L_V (C_p + L_V)|\mathcal{A}|L_\pi \sum_{i=\tau_k}^{d_m} \mathbb{E}\left[\|\theta_{k-i} - \theta_{k-d_m}\|_2|\theta', s_{t-m+1}\right], \qquad (109)
\end{aligned}
$$

where the second inequality is due to the definition of TV norm, the last inequality is due to (22) in Lemma 2, and thesecond last inequality follows the fact that

$$
|\Delta_3(x,\theta')| \leq \|\nabla J(\theta)\|_2 \left(\|\delta(x,\theta)\psi_{\theta_{k-d_m}}(s,a)\|_2 + \|\nabla J(\theta)\|_2\right) \leq L_V(C_p + L_V). \qquad (110)
$$

Taking total expectation on both sides of (109) yields

$$
\mathbb{E}[I_2] \geq -L_V(C_p + L_V)|\mathcal{A}|L_\pi \sum_{i=\tau_k}^{d_m} \mathbb{E}\|\theta_{k-i} - \theta_{k-d_m}\|_2.
$$

Define $\overline{x} := (\overline{s}, \overline{a}, \overline{s}')$, where $\overline{s} \sim d_{\theta'}, \overline{a} \sim \pi_{\theta'}$ and $\overline{s}' \sim \widetilde{\mathcal{P}}$. Then we have

$$
\mathbb{E}[I_3] = \mathbb{E}\left[\Delta_3(\widetilde{x},\theta') - \Delta_3(\overline{x},\theta')\right] + \mathbb{E}\left[\Delta_3(\overline{x},\theta')\right]. \qquad (111)
$$

We bound the first term in (111) as

$$
\begin{aligned}
& \mathbb{E}\left[\Delta_3(\widetilde{x},\theta') - \Delta_3(\overline{x},\theta')|\theta', s_{t-m+1}\right] \\
& \geq - \left|\mathbb{E}\left[\Delta_3(\widetilde{x},\theta')|\theta', s_{t-m+1}\right] - \mathbb{E}\left[\Delta_3(\overline{x},\theta')|\theta', s_{t-m+1}\right]\right| \\
& \geq - \sup_x |\Delta_3(x,\theta')| \|\mathbb{P}(\widetilde{x} \in \cdot|\theta', s_{t-m+1}) - \mathbb{P}(\overline{x} \in \cdot|\theta', s_{t-m+1})\|_{TV} \\
& \geq -2 L_V(C_p + L_V) d_{TV}\left(\mathbb{P}(\widetilde{x} \in \cdot|\theta', s_{t-m+1}), \mathbb{P}(\overline{x} \in \cdot|\theta', s_{t-m+1})\right) \\
& = -2 L_V(C_p + L_V) d_{TV}\left(\mathbb{P}(\widetilde{x} \in \cdot|\theta', s_{t-m+1}), d_{\theta'} \otimes \pi_{\theta'} \otimes \widetilde{\mathcal{P}}\right) \\
& = -2 L_V(C_p + L_V) d_{TV}\left(\mathbb{P}(\widetilde{x} \in \cdot|\theta', s_{t-m+1}), \mu_{\theta'} \otimes \pi_{\theta'} \otimes \widetilde{\mathcal{P}}\right) \qquad (112)
\end{aligned}
$$

where the second inequality follows the definition of total variation norm, and the third inequality follows (110). The last equality is due to the fact shown by [6] that $\mu_{\theta'}(\cdot) = d_{\theta'}(\cdot)$, where $\mu_{\theta'}$ is the stationary distribution of an artificial MDP with transition kernel $\widetilde{\mathcal{P}}(\cdot|s,a)$ and policy $\pi_{\theta'}$.

The auxiliary Markov chain with policy $\pi_{\theta'}$ starts from initial state $s_{t-m+1}$, and $\widetilde{s}_t$ is the $(m-1)$th state on the chain. Following Lemma 1, we have:

$$d_{TV}\left(\mathbb{P}(\widetilde{x} \in \cdot | \theta', s_{t-m+1}), \mu_{\theta'} \otimes \pi_{\theta'} \otimes \widetilde{\mathcal{P}}\right) = d_{TV}\left(\mathbb{P}\left((\widetilde{s}_t, \widetilde{a}_t, \widetilde{s}_{t+1}) \in \cdot | \theta', s_{t-m+1}\right), \mu_{\theta'} \otimes \pi_{\theta'} \otimes \widetilde{\mathcal{P}}\right)$$
$$\leq \kappa \rho^{m-1}.$$

Substituting the last inequality into (112) and taking total expectation on both sides yield

$$\mathbb{E}\left[\Delta_3(\widetilde{x}, \theta') - \Delta_3(\overline{x}, \theta')\right] \geq -2L_V(C_p + L_V)\kappa \rho^{m-1}.$$

Consider the second term in (111). Note its form is similar to (59), so by following the derivation of (63), we directly have

$$\mathbb{E}[\Delta_3(\overline{x}, \theta')] = \mathbb{E}\left\langle \nabla J(\theta'), \delta(\overline{x}, \theta')\psi_{\theta'}(\overline{s}, \overline{a}) - \nabla J(\theta')\right\rangle \geq -8C_\psi r_{\max}(1-\gamma)\mathbb{E}\left\|\nabla J(\theta')\right\|_2,$$

which further implies

$$\mathbb{E}[\Delta_3(\overline{x}, \theta')] \geq -8C_\psi r_{\max}(1-\gamma)\mathbb{E}\left\|\nabla J(\theta')\right\|_2$$
$$\geq -8C_\psi r_{\max}(1-\gamma)\mathbb{E}\left\|\nabla J(\theta') - \nabla J(\theta)\right\|_2 - 8C_\psi r_{\max}(1-\gamma)\mathbb{E}\left\|\nabla J(\theta)\right\|_2$$
$$\geq -8C_\psi r_{\max}L_J(1-\gamma)\mathbb{E}\left\|\theta' - \theta\right\|_2 - 8C_\psi r_{\max}(1-\gamma)\mathbb{E}\left\|\nabla J(\theta)\right\|_2$$

where the last inequality follows from Proposition 1.

Collecting the lower bounds gives

$$\mathbb{E}[I_3] \geq -2L_V(C_p + L_V)\kappa \rho^{m-1} - 8C_\psi r_{\max}(1-\gamma)\left(L_J \mathbb{E}\left\|\theta' - \theta\right\|_2 - \mathbb{E}\left\|\nabla J(\theta)\right\|_2\right).$$

Taking total expectation on $\Delta_3(x, \theta)$ and collecting lower bounds of $I_1$, $I_2$, $I_3$ yield

$$\mathbb{E}[\Delta_3(x, \theta)] \geq -\left(C_p L_J + (1+\gamma)L_V^2 C_\psi + 2L_V L_J + 8C_\psi r_{\max}L_J(1-\gamma)\right)\mathbb{E}\left\|\theta_k - \theta_{k-d_m}\right\|_2$$

$$- L_V(C_p + L_V)|\mathcal{A}|L_\pi \sum_{i=\tau_k}^{d_m} \mathbb{E}\left\|\theta_{k-i} - \theta_{k-d_m}\right\|_2 - 2L_V(C_p + L_V)\kappa \rho^{m-1} - 8C_\psi r_{\max}(1-\gamma)\mathbb{E}\left\|\nabla J(\theta_k)\right\|_2.$$

Taking total expectation on (107) and substituting the above inequality into it yield

$$\mathbb{E}\left\langle \nabla J(\theta_k), \delta(x_{(k)}, \theta_k)\psi_{\theta_{k-\tau_k}}(s_{(k)}, a_{(k)}) - \nabla J(\theta_k)\right\rangle \geq -D_6 \mathbb{E}\left\|\theta_{k-\tau_k} - \theta_{k-d_m}\right\|_2$$

$$- D_7 \mathbb{E}\left\|\theta_k - \theta_{k-d_m}\right\|_2 - D_8 \sum_{i=\tau_k}^{d_m} \mathbb{E}\left\|\theta_{k-i} - \theta_{k-d_m}\right\|_2 - D_9 \kappa \rho^{m-1} - 8C_\psi r_{\max}(1-\gamma)\mathbb{E}\left\|\nabla J(\theta_k)\right\|_2,$$

where $D_6 := L_V C_\delta L_\psi$, $D_7 := C_p L_J + (1+\gamma)L_V^2 C_\psi + 2L_V L_J + 8C_\psi r_{\max}L_J(1-\gamma)$, $D_8 := L_V(C_p + L_V)|\mathcal{A}|L_\pi$, $D_9 := 2L_V(C_p + L_V)$. □

## C.3 Explanation of the approximation error

In this section, we will provide a justification for the circumstances when the approximation error $\epsilon_{\text{app}}$ defined in (14) is small.

**Lemma 7.** *Suppose Assumption 2 and 4 hold. Then it holds that*

$$\epsilon_{\text{app}} \leq \max_{\theta \in \mathbb{R}^d} \sqrt{\mathbb{E}_{s \sim \mu_\theta} |V_{\pi_\theta}(s) - \hat{V}_{\overline{\omega}_\theta^*}(s)|^2} + 4r_{\max}(\lambda^{-1} + \lambda^{-2}r_{\max})\left(1 + \log_\rho \kappa^{-1} + \frac{1}{1-\rho}\right)(1-\gamma) \tag{113}$$

*where $\overline{\omega}_\theta^*$ the critic stationary point of original Markov chain with policy $\pi_\theta$ and transition kernel $\mathcal{P}$.*

In (113), the first term captures the quality of critic function parameterization method which also appears in previous works [14, 15, 17]. When using linear critic function approximation, it becomes zero when the value function $V_{\pi_\theta}$ belongs to the linear function space for any $\theta$. The second term corresponds to the error introduced by sampling from the artificial transition kernel $\widetilde{\mathcal{P}}(\cdot|s, a) = (1-\gamma)\mathcal{P}(\cdot|s, a) + \gamma\eta(\cdot)$. For a large $\gamma$ close to 1, the artificial Markov chain is close to the original

one. In this case, the second error term is therefore small. This fact also consists with practice where large $\gamma$ is commonly used in two time-scale actor critic algorithms [3].

Before going into the proof, we first define that:

$$\bar{A}_{\theta,\phi} := \mathop{\mathbb{E}}_{s\sim\bar{\mu}_\theta,s'\sim\mathcal{P}_{\pi_\theta}}[\phi(s)(\gamma\phi(s')-\phi(s))^\top], \qquad \bar{b}_{\theta,\phi} := \mathop{\mathbb{E}}_{s\sim\bar{\mu}_\theta,a\sim\pi_\theta,s'\sim\mathcal{P}}[r(s,a,s')\phi(s)],$$

where $\bar{\mu}_\theta$ as the stationary distribution of the original Markov chain with $\pi_\theta$ and transition kernel $\mathcal{P}$.

*Proof.* Recall the definition of the approximation error:

$$\epsilon_{\mathrm{app}} = \max_{\theta\in\mathbb{R}^d}\sqrt{\mathop{\mathbb{E}}_{s\sim\mu_\theta}|V_{\pi_\theta}(s)-\hat{V}_{\omega_\theta^*}(s)|^2},$$

where $\mu_\theta$ is the stationary distribution of the artificial Markov chain with $\pi_\theta$ and transition kernel $\widetilde{\mathcal{P}}$, and $\omega_\theta^*$ is the stationary point of critic update under the artificial Markov chain.

We decompose $\epsilon_{\mathrm{app}}$ as

$$\epsilon_{\mathrm{app}} = \max_{\theta\in\mathbb{R}^d}\sqrt{\mathop{\mathbb{E}}_{s\sim\mu_\theta}|V_{\pi_\theta}(s)-\hat{V}_{\bar{\omega}_\theta^*}(s)+\hat{V}_{\bar{\omega}_\theta^*}(s)-\hat{V}_{\omega_\theta^*}(s)|^2}$$

$$\leq \underbrace{\max_{\theta\in\mathbb{R}^d}\sqrt{\mathop{\mathbb{E}}_{s\sim\mu_\theta}|V_{\pi_\theta}(s)-\hat{V}_{\bar{\omega}_\theta^*}(s)|^2}}_{\epsilon_{\mathrm{fa}}} + \underbrace{\max_{\theta\in\mathbb{R}^d}\sqrt{\mathop{\mathbb{E}}_{s\sim\mu_\theta}|\hat{V}_{\bar{\omega}_\theta^*}(s)-\hat{V}_{\omega_\theta^*}(s)|^2}}_{\epsilon_{\mathrm{sp}}}, \tag{114}$$

where the first term corresponds to the function approximation error $\epsilon_{\mathrm{fa}}$, and second term corresponds to the sampling error $\epsilon_{\mathrm{sp}}$.

With $A$, $b$ and $\bar{A}$, $\bar{b}$ as shorthand notations for $A_{\theta,\psi}$, $b_{\theta,\psi}$ and $\bar{A}_{\theta,\psi}$, $\bar{b}_{\theta,\psi}$ respectively, we bound the second term in (114) as

$$\begin{aligned}
|\hat{V}_{\bar{\omega}_\theta^*}(s)-\hat{V}_{\omega_\theta^*}(s)| &= \left|\phi(s)^\top\omega_\theta^*-\phi(s)^\top\bar{\omega}_\theta^*\right| \\
&\leq \left\|A^{-1}b-\bar{A}^{-1}\bar{b}\right\|_2 \\
&= \left\|A^{-1}b-A^{-1}\bar{b}+A^{-1}\bar{b}-\bar{A}^{-1}\bar{b}\right\|_2 \\
&\leq \left\|A^{-1}(b-\bar{b})\right\|_2 + \left\|(A^{-1}-\bar{A}^{-1})\bar{b}\right\|_2 \\
&\leq \lambda^{-1}\|b-\bar{b}\|_2 + r_{\max}\left\|A^{-1}-\bar{A}^{-1}\right\|_2 \\
&= \lambda^{-1}\|b-\bar{b}\|_2 + r_{\max}\left\|A^{-1}(\bar{A}-A)\bar{A}^{-1}\right\|_2 \\
&\leq \lambda^{-1}\|b-\bar{b}\|_2 + \lambda^{-2}r_{\max}\left\|\bar{A}-A\right\|_2. \tag{115}
\end{aligned}$$

We bound the first term in last inequality as

$$\begin{aligned}
\|b-\bar{b}\|_2 &= \left\|\mathop{\mathbb{E}}_{s\sim\mu_\theta,a\sim\pi_\theta,s'\sim\widetilde{\mathcal{P}}}[r(s,a,s')\phi(s)] - \mathop{\mathbb{E}}_{s\sim\bar{\mu}_\theta,a\sim\pi_\theta,s'\sim\mathcal{P}}[r(s,a,s')\phi(s)]\right\| \\
&\leq \sup\|r(s,a,s')\phi(s)\|_2\|\mu_\theta\otimes\pi_\theta\otimes\widetilde{\mathcal{P}}-\bar{\mu}_\theta\otimes\pi_\theta\otimes\mathcal{P}\|_{TV} \\
&\leq 2r_{\max}d_{TV}(\mu_\theta\otimes\pi_\theta\otimes\widetilde{\mathcal{P}},\bar{\mu}_\theta\otimes\pi_\theta\otimes\mathcal{P}). \tag{116}
\end{aligned}$$

We now bound the divergence term in the last inequality as

$$\begin{aligned}
&d_{TV}(\mu_\theta\otimes\pi_\theta\otimes\widetilde{\mathcal{P}},\bar{\mu}_\theta\otimes\pi_\theta\otimes\mathcal{P}) \\
&= \int_{s\in\mathcal{S}}\sum_{a\in\mathcal{A}}\int_{s'\in\mathcal{S}}\left|\mu_\theta(s)\pi_\theta(a|s)\widetilde{\mathcal{P}}(s'|s,a)-\bar{\mu}_\theta(s)\pi_\theta(a|s)\mathcal{P}(s'|s,a)\right| \\
&= \int_{s\in\mathcal{S}}\sum_{a\in\mathcal{A}}\int_{s'\in\mathcal{S}}|\mu_\theta(s)\pi_\theta(a|s)\widetilde{\mathcal{P}}(s'|s,a)-\mu_\theta(s)\pi_\theta(a|s)\mathcal{P}(s'|s,a) \\
&\quad + \mu_\theta(s)\pi_\theta(a|s)\mathcal{P}(s'|s,a)-\bar{\mu}_\theta(s)\pi_\theta(a|s)\mathcal{P}(s'|s,a)| \\
&\leq \int_{s\in\mathcal{S}}\sum_{a\in\mathcal{A}}\mu_\theta(s)\pi_\theta(a|s)\int_{s'\in\mathcal{S}}\left|\widetilde{\mathcal{P}}(s'|s,a)-\mathcal{P}(s'|s,a)\right| + \int_{s\in\mathcal{S}}|\mu_\theta(s)-\bar{\mu}_\theta(s)|. \tag{117}
\end{aligned}$$

We bound the first term in (117) as

$$\int_{s' \in \mathcal{S}} \left| \widetilde{\mathcal{P}}(s'|s, a) - \mathcal{P}(s'|s, a) \right| = (1 - \gamma) \int_{s' \in \mathcal{S}} |\mathcal{P}(s'|s, a) - \eta(s')| \leq 2(1 - \gamma). \quad (118)$$

Following [39, Theorem 3.1], the second term in (117) can be bounded as

$$\int_{s \in \mathcal{S}} |\mu_\theta(s) - \bar{\mu}_\theta(s)| \leq \left( \log_\rho \kappa^{-1} + \frac{1}{1 - \rho} \right) \sup_s \int_{s' \in \mathcal{S}} \left| \sum_a \pi_\theta(a|s) \left( \widetilde{\mathcal{P}}(s'|s, a) - \mathcal{P}(s'|s, a) \right) \right|$$

$$\leq \left( \log_\rho \kappa^{-1} + \frac{1}{1 - \rho} \right) \sup_s \sum_a \pi_\theta(a|s) \int_{s' \in \mathcal{S}} \left| \widetilde{\mathcal{P}}(s'|s, a) - \mathcal{P}(s'|s, a) \right|$$

$$\leq 2 \left( \log_\rho \kappa^{-1} + \frac{1}{1 - \rho} \right) (1 - \gamma), \quad (119)$$

where the last inequality follows (118).

Substituting (118) and (119) into (117) gives

$$d_{TV}(\mu_\theta \otimes \pi_\theta \otimes \widetilde{\mathcal{P}}, \bar{\mu}_\theta \otimes \pi_\theta \otimes \mathcal{P}) \leq 2 \left( 1 + \log_\rho \kappa^{-1} + \frac{1}{1 - \rho} \right) (1 - \gamma).$$

Substituting the above inequality into (116) gives

$$\|b - \bar{b}\|_2 \leq 4 r_{\max} \left( 1 + \log_\rho \kappa^{-1} + \frac{1}{1 - \rho} \right) (1 - \gamma). \quad (120)$$

Similarly, we also have

$$\|A - \bar{A}\|_2 \leq 4 r_{\max} \left( 1 + \log_\rho \kappa^{-1} + \frac{1}{1 - \rho} \right) (1 - \gamma). \quad (121)$$

Substituting (120) and (121) into (115), then substituting (115) into (114) completes the proof. □

## D  EXPERIMENT DETAILS

**Hardware device.** The tests on synthetic environment and CartPole was performed in a 16-core CPU computer. The test on Atari game was run in a 4 GPU computer.

**Parameterization.** For the synthetic environment, we used linear value function approximation and tabular softmax policy [36]. For CartPole, we used a 3-layer MLP with 128 neurons and sigmoid activation function in each layer. The first two layers are shared for both actor and critic network. For the Atari seaquest game, we used a convolution-LSTM network. For network details, see [40].

| Hyper-parameters | Value |
|---|---|
| Number of workers | 16 |
| Optimizer | Adam |
| Step size | 0.00015 |
| Batch size | 20 |
| Discount factor | 0.99 |
| Entropy coefficient | 0.01 |
| Frame size | $80 \times 80$ |
| Frame skip rate | 4 |
| Grayscaling | Yes |
| Training reward clipping | [-1,1] |

Table 1: Hyper-parameters of A3C-TD(0) in the Atari seaquest game.

**Hyper-parameters.** For the synthetic environment tests, we run Algorithm 1 with actor step size $\alpha_k = \frac{0.05}{(1+k)^{0.6}}$ and critic step size $\beta_k = \frac{0.05}{(1+k)^{0.4}}$. In tests of CartPole, we run Algorithm 1 with a minibatch of 20 samples. We update the actor network with a step size of $\alpha_k = \frac{0.01}{(1+k)^{0.6}}$ and critic network with a step size of $\beta_k = \frac{0.01}{(1+k)^{0.4}}$. See Table 1 for hyper-parameters to generate the Atari game results in Figure 4.

