# OpenReview forum: "Asynchronous Advantage Actor Critic: Non-asymptotic Analysis and Linear Speedup"
_ICLR.cc/2021/Conference — Reject_

### Official Review · AnonReviewer4 · 2020-10-28
**The result is reasonable but the sampling has some issues**

**Rating:** 5
**Confidence:** 4

**Review:**

This paper studied the two time scale A3C in discounted MDP based on recent development in the finite sample analysis of A2C. The sample complexity result in this paper matches previous result in two time-scale A2C in terms of the dependence of \epsilon, and this paper further shows the benefit of "linear speed up" brough by the structure of A3C. Given the practical usefulness of A3C, the result established in this paper is meaninful.

Although most of the technical proof seem correct to me, the algorithm studied in this paper has the following issues.

(1) In the policy evaluation part (critic), samples are generated according to the the "hybrid" transition kernel \tilde{P}, which is not reasonable. The purpose of the policy evaluation is to evalue the value function, thus the samples used by the critic need to follow the transition kernel P not \tilde{P}. Otherwise, the value function learned by the critic would be different from the pure TD(0). I understand that the the author adopt this sampling method because the author want to make a "two time-scale" algorithm to study here. However, under such a sampling startegy only the actor can obtain appropriate samples. In fact, it is very difficult to design a two time-scale algorithm for a discount MDP, as the transition kernel required by actor and critic are different. I suggest the author to study "averaged MDP" instead. In the averaged MDP, both the actor and critic can share the same transition kernel, which make designing a two time-scale algorithm possible.

(2) Another problem is the projection used in the critic. The projection in linear SA is a practical problem, as we do not have prior knowledge of the radius before we run the algorithm. The projection issue has been criticized by many researchers' in the RL community in the last few years thus should not be ignored here. In fact, the projection issue can be avoid if the author adopts a nested-loop structure, and I think the sample complexity result can even be  improved in the nested-loop setting. Thus I suggest the author to used nested-loop structure to avoide this projection issue.

Overall I think the paper is well written. However, considering the two problems that I mentioned above, I think the current version is not ready to be published.

---

> ### Author Response · Authors · 2020-11-18
> **Thanks for valuable suggestions! We have clarified the sampling and projection questions.**
>
> We thank the reviewer for the careful reviews and insightful suggestions.
>
>
> (1) Sampling method
>
> As pointed out by the reviewer, it is *very difficult* to design a two time-scale AC algorithm for the discounted MDP, as the transition kernel required by the actor and critic updates are different. Aware of this sampling mismatch, we adopted the sampling method introduced in the seminal work [R4, R5] and many follow-up works, e.g., [R7, R8], which inevitably introduces extra bias because of sampling from the artificial transition $\widetilde{\cal P}$ instead of the original ${\cal P}$.  However, this extra bias is small when the discount factor $\gamma$ is close to 1. This choice of $\gamma$ is also common in practice. For example, $\gamma$ is chosen as 0.99 in A3C [R6]. In this case, the difference between the stationary distribution and visitation measure are both small [R4].
>
> To resolve the concerns of the reviewer, we have decomposed the approximation error defined in (14) into two errors: i) the *function approximation error*, which is common in TD with function approximation; and, ii) the *sampling error* induced by distribution mismatch. We have added Lemma 7 to analytically quantify this sampling error in supplementary material, which is indeed ${\cal O}(1-\gamma)$. We have also highlighted the corresponding changes we have made for the rest of proofs. This analysis also justifies the choice of $\gamma= 0.99$ in A3C [R6]. The result is also consistent with the intuition that the discrepancy between average reward setting and discounted reward setting will be small with a $\gamma$ close to 1, as one can see that our result reduces to that of [R3] which studies the average reward setting with a $\gamma$ approaching 1.
>
> Of course, one can bypass this subtle issue by studying A3C for an average-reward MDP. However, this will make our setting different from the original work [R6], and thus deviate from our original purpose --- provide a theoretical justification of A3C.
>
> (2) Projection of critic parameter
>
> The projection step is often used to control the norm of the gradient. In the context of AC, it prevents the actor or critic from going a too large step in the ‘wrong’ direction. Such control of gradient is even more important in a two time-scale algorithm where the tolerance for ‘bad’ direction is especially small compared to, say, nested-loop algorithms. Projection has also been adopted by many previous TD/AC works like [R1, R2, R3, R4, R5, R7, R8, R9]. To the best of our knowledge, no provably convergent two time-scale AC algorithm has been studied without projection. In practice, one can choose the projection radius as ${\cal O}(r_{\rm max}/\lambda)$ [R3, R8, R9], *without loss of optimality*.
>
> Thanks for the insightful suggestion! Indeed, one can bypass this subtle issue by studying the nested-loop AC that may even enjoy an improved sample complexity; see e.g., [R7]. However, this will also make the considered A3C algorithm different from the original one in [R6]. To mitigate the reviewer’s concerns, we have added some clarifications after Assumption 2.
>
> To resolve your technical concerns, we have added some clarifications and new derivations in the revised paper.
>
> Feel free to let us know if this resolves your concerns, and we are open to further discussions!
>
>
>
> [R1] J. Bhandari, D. Russo, and R. Singal, ``A Finite-Time Analysis of Temporal Difference Learning With Linear Function Approximation,’’ COLT 2018, arxiv.org/pdf/1806.02450.pdf
>
> [R2] S. Qiu, Z. Yang, J. Ye, and Z. Wang, ``On the Finite-Time Convergence of Actor-Critic Algorithm,’’ NeurIPS workshop 2019, optrl2019.github.io/assets/accepted_papers/43.pdf
>
> [R3] Y. Wu, W. Zhang, P. Xu, and Q. Gu, “A finite time analysis of two time-scale actor critic methods,” NeurIPS 2020, https://arxiv.org/pdf/2005.01350.pdf
>
> [R4] V. Konda, ``Actor-critic algorithms,’’ MIT PhD Thesis 2002,
> https://dspace.mit.edu/bitstream/handle/1721.1/8120/51552606-MIT.pdf;sequence=2
>
> [R5] V. Konda, J. Tsitsiklis, ``Actor-critic algorithms,’’ NIPS 2000
>
> [R6] V. Mnih, et al, ``Asynchronous methods for deep reinforcement learning,’’ ICML 2016,
> https://arxiv.org/pdf/1602.01783.pdf
>
> [R7] T. Xu, Z. Wang, and Y. Liang, ``Improving sample complexity bounds for (natural) actor-critic algorithms,” NeurIPS 2020, https://arxiv.org/pdf/2004.12956.pdf
>
> [R8] T. Xu, Z. Wang, and Y. Liang, ``Non-asymptotic Convergence Analysis of Two Time-scale (Natural) Actor-Critic Algorithms,” https://arxiv.org/pdf/2005.03557.pdf
>
> [R9] S. Zou, T. Xu, and Y. Liang, “Finite-sample analysis for SARSA with linear function approximation,” NeurIPS 2019, https://arxiv.org/pdf/1902.02234.pdf

---

### Official Review · AnonReviewer2 · 2020-10-29
**Theoretical analysis for A3C with 1-step TD**

**Rating:** 6
**Confidence:** 3

**Review:**

This paper revisits the A3C algorithm with TD(0) for the critic update to provide better theoretical analysis of A3C. A3C-TD(0) achieves linear speedup and it also matches our intuition. To show the empirical results, the authors provide convergence results of A3C-TD(0) with Markovian sampling in synthetic environments and speedup of A3C-TD(0) in CartPole and Seaquest.
In this paper, the theoretical and experimental results show that using multiple workers in parallel improves learning speed without loss of sample efficiency. I think it is a valuable research direction.
However, A3C-TD(0) is limited compared to A3C because A3C-TD(0) does not use multi-step TD or TD(\lambda). Moreover, the authors use only two gym environments, which seems insufficient.

---

> ### Author Response · Authors · 2020-11-18
> **Thanks for careful reviews! We have added explanation and new simulation.**
>
> We thank the reviewer for reviewing our paper and recognizing our theoretical contribution.
>
> (1) Extension to multi-step TD method
>
> The multi-step TD method is a hybrid combination of TD(0) method and the Monte-Carlo method that uses an almost unbiased stochastic gradient. For the asynchronous AC method based on Monte-Carlo sampling, its convergence follows directly from the delayed SGD. And our work has already proved the convergence of the asynchronous AC method based on TD(0).
> In order to extend our theory to multi-step TD, we will use the error-reduction property of n-step return which establishes that the expected value of all n -step returns is guaranteed to improve over the current value function approximator to the true value function (see (7.3) in page 144 of [R1]):
>
> $ \max_s  \Bigg|E_\pi\Big[\sum_{i=0}^{n-1}\gamma^i r_{t+i} + \gamma^n \hat{V}_\omega(s_n)\Big|s_t=s\Big] - V_\pi (s)\Bigg| \leq \gamma^n \max_s \Big| \hat{V}_\omega(s)- V_\pi (s)\Big|
>  $.
>
> This also implies that the n-step TD error will improve our 1-step TD error, which is a critical quantity in our convergence analysis. With the new expression and bound of TD error, we will need to slightly modify our analysis in places where we need to quantify the approximate TD error, e.g., (58) and Lemma 5. But it will only improve some constants. Therefore, we believe that our results can be easily extended to the convergence of async AC with multi-step TD.
>
> We have added new discussion at the end of Section III. Due to the space limitation, we will pursue this extension in our future work.
>
>
> (2) More experiments
>
> We viewed our work as a theory-oriented one, with main focus on establishing the convergence rate and sample complexity for a well-known algorithm (A3C), following the recent line of works on the finite-sample analysis of AC [12]-[18]. In fact, except some basic tests (not gym tests) reported in [13], none of these works have provided any numerical results. We include some empirical validations as an add-on to our theoretical contribution.
>
> To mitigate the concerns of the reviewer, we have added a new gym Atari game, and we are also running more numerical tests which will be included in the final version.
>
> [R1] R. S. Sutton, A. G. Barto, ``Reinforcement Learning: An Introduction,’’ MIT Press 2018, http://incompleteideas.net/book/RLbook2020.pdf

---

### Official Review · AnonReviewer1 · 2020-10-30
**Well written paper with sound results**

**Rating:** 6
**Confidence:** 2

**Review:**

This paper studied the convergence rate of the asynchronous actor-critic algorithm (with linear value function approximation)  for RL. This paper showed A3C-TD(0) has O(\epsilon^{-2.5}/N)  sample complexity per worker where N is the number of workers., which means the per-worker sample complexity can be reduced linearly with respect to # of worker.

In general, this paper is well written and easy to follow. The sample complexity results are insightful.  Could the authors address my following comments/questions?

(1)$w_\theta^\ast$ or $w_{\theta_k}^\ast$ are never formally defined. Please clarify the notations properly.
(2)Unlike "linear-speedup" in setting such as convergence rates for SGD,  the linear speedup is not so "idea". The gradient norm has a constant error floor term $O(\epsilon_{app})$, which eventually dominates the error as K increases.   To my understanding, an error term is unavoidable because the valuation is approximated by a linear function.  I wonder if the results in the current paper can be easily extended to cover the situation where no linear approximation is used, in which case I expect the error terms disappears totally.
(3)  This is also related to (2). It is reasonable that the critic-convergence (Theorem 1) uses  a drifting $\theta_k$. But if we look at the actor and critic together, our goal is to converge to a fixed policy, am I right? If so, is it necessary to define $\epsilon_{app}$ in (13) as the error regarding to the worst \theta? (In other words, can we remove $max_{\theta\in R^d}$ in (13) and define $\epsilon_{app}$ using a single $\theta$? By doing that, $\epsilon_{app}$ can be much smaller)

---

> ### Author Response · Authors · 2020-11-18
> **We thank Reviewer 1 for detailed comments!**
>
> Thanks for your favorable rating. We appreciate your careful review. Our response to your comments follows.
>
> (1) Formal definition of $w_\theta^\ast$
>
> The variables $w_\theta^\ast$ and $w_{\theta_k}^\ast$ denote the stationary points of TD update under policies $\pi_\theta$ and $\pi (\theta_k)$, respectively. Now it is defined explicitly in (11). Thanks for the careful reading.
>
> (2) Extension to no value function approximation
>
> Indeed, our convergence result can be easily extended to cover the situation where no function approximation is used, e.g., tabular case, and the corresponding function approximation error will be 0. Specifically, for the tabular case, we can simply treat the state feature $\phi(s)$ as a |S|-dimensional standard basis vector, where only s-th entry is 1 and other entries are 0. In this case, the |S|-dimensional critic parameter concatenates the true value functions. In general, if the linear combination of the feature mapping can express the true value function, then the value function approximation error will be zero (see Lemma 2 of [R1]).
>
> (3) Relaxation of the definition of $\epsilon_{app}$
>
> In the proof, we use $\epsilon_{app}$ to bound the function approximation error at each iteration. Since at each iteration we have different policy parameter $\theta_k$, $\epsilon_{app}$ cannot be defined over a fixed policy parameter such as the stationary point of actor update. We define $\epsilon_{app}$ in this way just for notational brevity. Similar definition of $\epsilon_{app}$ was also widely used in prior works such as [R2], [R3]. An alternative way is to define an iteration-dependent $\epsilon_{app}$, which will complicate the already heavy notations.
>
> [R1] J. Bhandari, D. Russo, and R. Singal, "A Finite-Time Analysis of Temporal Difference Learning With Linear Function Approximation,’’ COLT 2018, arxiv.org/pdf/1806.02450.pdf
>
> [R2] S. Qiu, Z. Yang, J. Ye, and Z. Wang, ``"On the Finite-Time Convergence of Actor-Critic Algorithm,’’ NeurIPS workshop 2019, optrl2019.github.io/assets/accepted_papers/43.pdf
>
> [R3] Y. Wu, W. Zhang, P. Xu, and Q. Gu, “A finite time analysis of two time-scale actor critic methods,” NeurIPS 2020, https://arxiv.org/pdf/2005.01350.pdf

---

### Author Response · Authors · 2020-11-20
**General response**

We first thank all the reviewers for their valuable comments!

Most of the reviews are quite positive, finding our work “insightful”, “theoretical results solid”, and “well-written." We really appreciate those positive comments.

Negative comments mainly came from our insufficient explanation on the algorithm implementation and the theoretical results.
Following your suggestions, we have updated the submission by adding more explanations and quantifying approximation errors.

We hope that our response and revision can resolve your remaining concerns. More suggestions on further improving the paper are also always welcomed! We are looking forward to the discussions.

---

### Author Response · Authors · 2020-11-24
**Summary of contributions and revisions**

Thanks again for all the careful reviews!

Below we summarize our **main contributions**.

(1)	We revisit the A3C method with TD(0) for the critic update, termed A3C-TD(0), and provides non-asymptotic guarantee with both i.i.d. and Markovian sampling. To the best of our knowledge, this is the first non-asymptotic result for *asynchronous* parallel AC algorithms.

(2)	We further show that in the i.i.d. setting, A3C-TD(0) achieves a sample complexity of $\mathcal{O}(\epsilon^{-2.5}/N)$ per worker, where $N$ is the number of workers. Compared to the best-known complexity of $\mathcal{O}(\epsilon^{-2.5})$ for i.i.d. two-timescale AC, A3C-TD(0) achieves *linear speedup*, thanks to the *parallelism* and *asynchrony*. In the Markovian setting, the sample complexity of A3C-TD(0) matches the order of the non-parallel AC algorithm.

(3)	We test A3C-TD(0) on the synthetically generated environment as well as the OpenAI Gym environment (classic control and Atari Games). We make our code publicly available to enhance reproducibility.

Below we summarize the **major improvements** we have made during the rebuttal period.

(1)	As suggested by Reviewer 1, we have added explicit definitions of some variables and discussed the reduction to the tabular setting without value function approximation.

(2)	As suggested by Reviewer 2, we have added some discussion on how to generalize our results to A3C with n-step TD. Due to space limitation, we will pursue this extension in the future work.

(3)	As suggested by Reviewer 2, we have added new simulations on Atari game, and we are running more numerical tests which will be included in the final version.

(4)	Reviewer 4 pointed out the difficulty of sampling in two-time scale AC for discounted MDP and kindly suggested that we should analyze the average-reward setting. However, analyzing the average MDP will make our setting different from the *original A3C work*, and thus deviate from our original purpose --- to provide some theoretical justification of the use of asynchronous update in AC algorithms. Therefore, we still consider the discounted setting (as in the *original A3C paper*) and follow the update rule as in the *classical AC paper* (Konda, MIT PhD thesis, 2002).

However, we have managed to provide a *fine-grained* analysis to explicitly decompose the approximation error into: i) the function approximation error, which comes from the TD with function approximation; and, ii) the sampling error which seems inevitable in two-timescale AC for discounted MDP due to the distribution mismatch. We have added Lemma 7 to quantify this sampling error, which is indeed ${\cal O}(1-\gamma)$. We have highlighted the corresponding changes we have made for the rest of the proofs due to this new analysis. This analysis also justifies the choice of $\gamma= 0.99$ in the A3C work.

(5)	To further resolve Reviewer 4’s concern on the projection used in the critic update, we have added more discussion in the paper on its *necessity* and the *practical choice* of the projection radius, without loss of optimality. We have also added several key references that range from the original AC paper to some exciting recent results to justify that the projection is common in both implementing and analyzing two timescale AC.

We hope that our response and revision can resolve all your concerns. **If you have any further suggestions and comments, we are more than happy to implement them in our final version.**

---

### Decision · Program_Chairs · 2021-01-07
**Final Decision**

**Decision:**

Reject

**Comment:**

Although the reviewers found the paper well-written that analyzes a relatively popular algorithm (TD(0) version of A3C), there are concerns regarding the novelty of the convergence results given those for A2C, the comparison of the results with those for A2C, and the sufficiency of the experiments. Although the authors addressed some of these issues/comments during the rebuttals, it seems none of the reviewers is excited about the paper and there still exist concerns regarding the novelty of the results and how they are compared with those in the literature. I would suggest that the authors take the reviewers' comments into account, have a more comprehensive discussion about the relation of their results with those in the literature (two-time scale algorithms), and prepare their work for future conferences.